# Automatic Combination of Sample Selection Strategies for Few-Shot Learning

## Abstract

In few-shot learning, such as meta-learning, few-shot fine-tuning or in-context learning, the selection of samples has a significant impact on the performance of the trained model. Although many sample selection strategies are employed and evaluated in typical supervised settings, their impact on the performance of few-shot learning is largely unknown. In this paper, we investigate the impact of 20 sample selection strategies on the performance of 5 representative few-shot learning approaches over 8 image and 6 text datasets. We propose a new method for Automatic Combination of SamplE Selection Strategies (ACSESS), to leverage the strengths and complementarity of the individual strategies in order to select more impactful samples. The experimental results show that our method consistently outperforms all individual selection strategies. We also show that the majority of existing strategies strongly depend on modality, dataset characteristics and few-shot learning approach, while improving performance especially on imbalanced and noisy datasets. Lastly, we show that sample selection strategies work well even on smaller datasets and provide larger benefit when selecting a lower number of shots, while frequently regressing to random selection with higher numbers of shots.

## 1 Introduction

Many domains are characterised by a labelled data scarcity due to data collection and annotation costs and privacy, making the training of typical deep learning models unfeasible. To address these problems, few-shot learning aims to train or adapt machine learning models to new tasks using only a few labelled samples per class (Song et al., 2023). There are two main categories of few-shot learning (more details regarding their differences are included in Appendix K): 1) *gradient-based*, which includes meta-learning and fine-tuning, where the knowledge from relevant datasets is transferred to a new problem using gradient-based training, similar to supervised learning (Song et al., 2023; Chen et al., 2019; Vanschoren, 2018; Hospedales et al., 2021); and 2) few-shot learning via *in-context learning*, a new paradigm, where a pretrained large language model is conditioned on few examples to generate the prediction without any parameter updates (Dong et al., 2022; Liu et al., 2023).

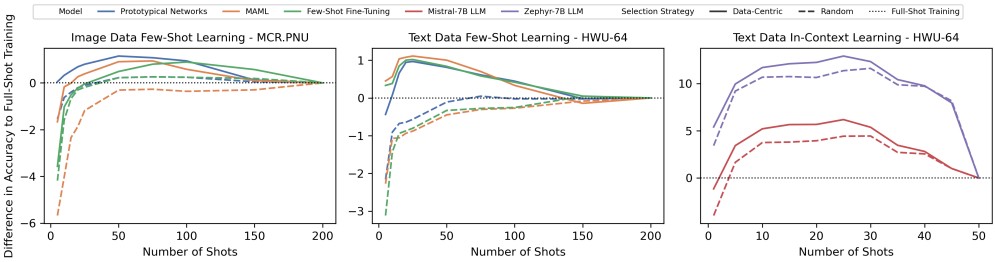

Figure 1: Curating a smaller subset of informative and high quality samples frequently leads to better performance than training on all available samples on specific datasets (e.g., with noisy samples).

As the number of utilised labelled samples is small in few-shot learning, the quality of individual samples is crucial. Existing studies (in different settings such as aligning large language models (Zhou et al., 2024)) have shown that **curating a smaller subset of the most informative and high-quality**

**samples can lead to better performance than training on the full set of available samples**. This holds for few-shot learning as well, especially for noisy or unbalanced datasets, as illustrated in Figure 1. Moreover, a strong sensitivity to the sample selection causes performance variance between state-of-the-art and random predictions (Pecher et al., 2023; Agarwal et al., 2021; Köksal et al., 2022).

Despite a lot of research dedicated to sample selection, such as active learning (Ren et al., 2021) or core-set selection (Guo et al., 2022), **the impact of sample selection for few-shot learning remains under-explored**, where the most common approach is random selection. The exception is in-context learning, where a large number of specific sample selection strategies have been proposed recently. They select samples according to sample similarity, diversity, informativeness or quality (Li & Qiu, 2023; Zhang et al., 2022; Chang & Jia, 2023). However, the use of more general approaches (e.g., active learning) for in-context learning is still largely unknown. Therefore, this paper takes the data-centric perspective of few-shot learning to **investigate the impact of the sample selection strategies on few-shot image and text classification tasks across a wide range of datasets**.

The various existing strategies mostly focus on selecting samples based on a single property (e.g., how often the samples are forgotten (Toneva et al., 2018), how easy they are to learn (Swayamdipta et al., 2020; Zhang & Plank, 2021) or how informative they are (Ye et al., 2023; Liu & Wang, 2023)). In this paper, we ask **whether we can automatically and optimally combine sample selection strategies to identify samples with multiple complementary properties**. To answer this question and to leverage the strengths of different sample selection strategies, we propose **ACSESS**, an effective method for **A**utomatic **C**ombination of **S**ampl**E** **S**election **S**trategies. The proposed method first identifies a subset of relevant strategies that can improve the overall success of few-shot learning. Afterwards, the identified strategies are combined together, using weighting based on their expected contribution, in order to identify the most informative and high-quality samples that can provide the most benefit. We summarise our contributions and findings as follows:[1]

- We evaluate the impact of 20 sample selection strategies on 5 few-shot learning approaches, including meta-learning, few-shot fine-tuning and in-context learning, across 8 image and 6 text datasets. Results show that the selection strategies often lead to non-negligible improvements in performance (up to 2 and 3 percentage points for gradient few-shot learning and in-context learning respectively), with a strong approach and dataset dependence. Overall, we find that curating a set of high-quality samples is especially important for imbalanced datasets with noisy samples, while the performance increase is limited for datasets with high-quality samples.

- We propose a method for automatic combination of sample selection strategies to leverage their strengths and identify samples with complementary properties required for the success of few-shot learning (i.e., informativeness, representativeness and learnability). Experimental results show that the method consistently leads to performance improvement (up to 3 and 5 percentage points for gradient few-shot learning and in-context learning respectively).

- We further analyse how the number of shots and the size of dataset affect the impact of sample selection. We find following key insights: 1) sample selection has higher impact when the number of shots is low or when using noisy datasets; 2) at higher number of shots (30-40 on average), the impact of sample selection is negligible as all strategies regress to random selection; 3) after a certain point, the boost in performance from using more shots becomes negligible (50-shot for gradient few-shot learning; 20-shot for in-context learning); 4) on well-balanced datasets without noisy samples, reducing the number of samples often leads only to reduction in computation costs, while keeping the same performance as with all samples; and 5) sample selection is beneficial even for small dataset sizes (achieving similar performance when selecting from only 10% of the dataset for gradient few-shot learning, or 25% for in-context learning).

## 2 RELATED WORK

A lot of research is dedicated to sample selection: 1) to reduce the computation costs without the loss of performance, by distilling dataset to lower number of samples (Yu et al., 2023); 2) to select a core

---

[1]To support replicability and extension of our results, we openly publish the source code of our experiments at https://anonymous.4open.science/r/ACSESS

set of a dataset with similar properties and same performance as the full set (Guo et al., 2022); or 3) to reduce annotation costs using active learning (Ren et al., 2021). A specific research focus is dedicated to selecting good performing samples for in-context learning, where the overall performance was found to be sensitive to this choice, leading to large variability in results (Pecher et al., 2023; Köksal et al., 2022; Zhang et al., 2022). Particularly in few-shot learning, the adoption of sample selection is limited to active learning strategies and in-context learning.

The sample selection strategies can be divided into two groups. The first, more prevalent, **uses heuristics and unsupervised metrics** to rate and select samples according to specific property that serves as estimate of their potential to increase the model's performance. The most popular approach is selecting samples based on the **similarity** to the test sample for in-context learning. The methods differ in employed: 1) similarity measure, such as cosine/Euclidean distance (Liu et al., 2022; An et al., 2023; Zemlyanskiy et al., 2022; Gupta et al., 2022; Pasupat et al., 2021), BM-25 (Wang et al., 2022; Nashid et al., 2023) or their combination (Agrawal et al., 2023; Gao et al., 2021); and 2) representation, such as simple TF-IDF (Agrawal et al., 2023; Wang et al., 2022; Zemlyanskiy et al., 2022), word embeddings from basic or further fine-tuned transformers (Liu et al., 2022; Gao et al., 2021; Gupta et al., 2022; Pasupat et al., 2021), or representation generated by prompting a large language model (An et al., 2023; Nashid et al., 2023).

Besides similarity, the samples are often selected based on their **informativeness** or the **uncertainty**. The active learning strategies are most common (Köksal et al., 2022; Schröder et al., 2022; Margatina et al., 2021; Park et al., 2022; Mavromatis et al., 2023), but also other approaches are utilised, such as using the bias of the samples (Ma et al., 2023) or adversarial training (Agarwal et al., 2021). Other methods define a notion of **quality** for each sample, either by prompting a large language model to rate the samples (Shin et al., 2021) or observing how the inclusion or removal of the sample affects the performance (Maronikolakis et al., 2023; Nguyen & Wong, 2023). When considering **representativeness**, the samples are selected using core-set methods that are designed to find a subset of samples with properties that are representative of the full training set (Guo et al., 2022; Killamsetty et al., 2021; Mirzasoleiman et al., 2020; Paul et al., 2021; Coleman et al., 2020). **Learnability** metric either determines how easy it is to learn the specific sample (Swayamdipta et al., 2020; Zhang & Plank, 2021), or how often the sample is forgotten after being learned (Toneva et al., 2018).

Finally, some approaches **balance multiple properties** at the same time. Often, the similarity of samples is balanced with their diversity (Wu et al., 2023; Qin et al., 2023) or the informativeness of the samples is balanced with their representativeness (Su et al., 2022; Levy et al., 2023; Ye et al., 2023; Liu & Wang, 2023). In specific cases, a two step sample search is done, such as finding set of informative samples and then using diversity guided search to improve this set (Li & Qiu, 2023).

The second set of approaches **leverages training to select the samples**. One possibility is to train a sample retriever, such as a separate scoring model (Rubin et al., 2022; Luo et al., 2023; Li et al., 2023; Wang et al., 2023; Aimen et al., 2023). Another option is to leverage reinforcement learning to select good samples (Zhang et al., 2022; Shum et al., 2023; Scarlatos & Lan, 2023). Finally, the Datamodels approach trains a linear model to predict the performance gain of a set of samples to select the subset that would lead to the highest possible performance increase (Ilyas et al., 2022; Chang & Jia, 2023; Jundi & Lapesa, 2022; Vilar et al., 2023).

In this work, we investigate the impact of the existing strategies on few-shot setting and complement work by Agarwal et al. (2021) by exploring which sample characteristics are important for few-shot learning performance. In contrast to Li & Qiu (2023), which represents a work closest to ours, we explore multiple few-shot learning approaches not only in-context learning. Our proposed method is inspired by the Datamodels approach (Ilyas et al., 2022), but performs the selection at the level of strategies instead of samples.

# 3 SAMPLE SELECTION STRATEGIES

## 3.1 SINGLE-PROPERTY STRATEGIES

In this work, we thoroughly investigate the impact of strategies that consider 3 groups of properties: 1) *informativeness*, or how informative the samples are for the model; 2) *representativeness*, or how well the subset of samples represents the full dataset; and 3) *learnability*, or how easy it is to obtain or

retain the information contained in the samples. We specifically focus on strategies that assign a score to each sample that is used to sort the samples and choose the best. This includes active learning strategies that measure the uncertainty of the model, core-set selection strategies that find subset representative of the full-set and other strategies that use training dynamics. The high-level overview of the selection strategies we use is presented in Table 1, with their further details in Appendix D.

Table 1: The few-shot scenario sample selection strategies evaluated in this paper, categorised based on the sample property they consider and the group of selection strategies it belongs to.

|  | Group | Strategy | Description | Reference |
|---|---|---|---|---|
| **Informativeness** | Similarity | Similarity | Most similar samples (popular for in-context learning). | — |
|  | Similarity | Diversity | Most diverse/dissimilar samples (popular for in-context learning). | — |
|  | Active Learn. | Entropy | Highest entropy over class probabilities. | (Park et al., 2022) |
|  | Active Learn. | Margin | Lowest difference in class probability between the 2 most probable classes. | (Park et al., 2022) |
|  | Active Learn. | Least Confidence | Class with lowest assigned probability. | (Park et al., 2022) |
|  | Active Learn. | Loss | Highest loss. | (Park et al., 2022) |
|  | Core-Set | Contrastive Active Learning (CAL) | Predictive likelihood that diverges most from the neighbourhood as determined by KL divergence. | (Margatina et al., 2021) |
|  | Core-Set | DeepFool | Smallest amount of perturbation to change class. | (Ducoffe & Precioso, 2018) |
|  | Core-Set | GraNd | Highest contribution to decline of training loss. | (Paul et al., 2021) |
|  | Core-Set | Graph-Cut | Submodularity function that measures diversity and informativeness. | (Iyer & Bilmes, 2013; Iyer et al., 2021) |
| **Representativeness** | Core-Set | Herding | Minimising distance of subset centre to full dataset. | (Welling, 2009; Chen et al., 2010) |
|  | Core-Set | KCenter | Minimising distance of every sample in subset to full dataset. | (Sener & Savarese, 2018; Agarwal et al., 2020) |
|  | Core-Set | CRAIG | Gradients representative of full dataset. | (Mirzasoleiman et al., 2020) |
|  | Core-Set | Glister | Bi-level optimisation maximising log-likelihood. | (Killamsetty et al., 2021) |
| **Learnability** | Core-Set | Forgetting | How often the samples are forgotten. | (Toneva et al., 2018) |
|  | Cartography | Cartography | How easy it is to learn the samples. Due to no obvious consensus, all of easy, ambiguous, hard and combination of easy and ambiguous samples are considered. | (Swayamdipta et al., 2020; Zhang & Plank, 2021) |

## 3.2 ACSESS: Automatic Combination of Strategies

Even though the different selection strategies can identify samples with a specific single property, samples characterised by only one property may not necessarily have a significant impact on the performance in a few-shot setting. For example, the most informative sample that is hard to learn or is often forgotten may not contribute as much, leading to low impact on the performance. Instead, we propose to select examples that are characterised by a balance of complementary properties, even when they do not represent the best sample of any given property (e.g., a sample that may be less informative, but is more easily learned or retained). To accomplish this, we propose **ACSESS**, an effective method for **A**utomatic **C**ombination of **S**ampl**E** **S**election **S**trategies, composed of two stages: 1) *identifying a subset of well-performing and relevant strategies* that can improve the overall success of few-shot learning; and then 2) *combining the identified strategies* using a weighted scheme based on their expected contribution. Similar to previous strategies, our method assigns a score to each sample (based on the combination of the scores from the identified strategies) and selects the samples with the highest score. The steps of the method are summarised in Algorithm 1, with further supplementary details, including steps of forward, backward and datamodels selection in Appendix C.

---

**Algorithm 1** Automatic Combination of Sample Selection Strategies

---
**Require:** $\mathbf{S}$ - set of sample selection strategies
**Require:** $scores$ - empty vector of scores for each sample
**Require:** $N$ - number of samples to choose
1: Identify strategies using forward selection, $\mathbf{S_F} = forward(\mathbf{S})$
2: Identify strategies using backward selection, $\mathbf{S_B} = backward(\mathbf{S})$
3: Identify strategies and weights using datamodels selection $\mathbf{S_D}, weights_D = datamodels(\mathbf{S})$
4: Select final set of samples $\mathbf{S_{final}} = \mathbf{S_F} \cap \mathbf{S_B} \cap \mathbf{S_D}$
5: **for all** $s_i$ in $\mathbf{S_{final}}$ **do**
6:     Assign score to each sample using strategy $s_i$, $scores_i = apply\_strategy(s_i)$
7: **end for**
8: $weights_{final} = weights_D$ if weighted combination else $1/|\mathbf{S_{final}}|$
9: Calculate final set of scores $scores_{final} = \sum_{i=0}^{|\mathbf{S_{final}}|} scores^i \cdot weights_{final}^i$
10: Select top $N$ samples based on $scores_{final}$

---

**Identifying the Subset of Relevant Strategies** In the first phase, the aim is to identify relevant strategies that, when combined, can select high quality samples based on their different properties. Even though the most optimal combination would be achieved by searching through all the combinations of strategies, such a search is infeasible as it requires extensive computation. Instead, we draw inspiration from strategies for feature selection in traditional machine learning – namely the *forward* and *backward* selection strategies – and for the first time apply them to the problem of sample selection. In addition, we adapt a strategy for individual sample selection, *Datamodels* strategy (introduced by Ilyas et al. (2022)), to work on the level of strategies instead. Such strategy allows us to explore a more diverse set of possible strategy combination as compared to forward/backward selection. All three of the methods are run independently of each other, each producing a separate set of strategies. The final set of strategies used by our method is constructed as an intersection between these sets to identify the most impactful strategies (i.e., strategies identified by all of the methods) and minimise the potential drawbacks of the individual methods (i.e., exploring also diverse combinations).

**Forward Selection** iteratively adds to the subset the selection strategy that results in the highest performance increase (similarly to forward feature selection). Starting with an empty subset, the method alternates between two steps. First, for each strategy not included in the subset we determine the increase in performance the strategy would yield if added to the subset. We determine the increase by evaluating the performance of the samples selected by the uniform combination of the already selected strategies and the potential new strategy, and then compare it with the performance of the old subset. Afterwards, the strategy that provides the highest positive increase in performance is added to the subset and the first step is repeated. If no added strategy leads to increase of the overall performance, the process ends, resulting in the subset of relevant selection strategies.

**Backward Selection** iteratively removes strategies from the subset until the performance no longer increases (similarly to backward feature selection). The method works the same way as the forward selection, but starting with all selection strategies and removing them from the subset based on the performance increase such removal leads to.

**Datamodels Selection** draws inspiration from Datamodels (Ilyas et al., 2022; Chang & Jia, 2023), but works on the level of the strategies instead of samples. First, a set of 150 random combinations of strategies is created and evaluated (using uniform combination). The random search is constrained to include each strategy at least 5 times to ensure sufficient coverage of each strategy. Afterwards, these combinations are used to create a simple regression dataset, with the difference in performance to the baseline selection (the classic few-shot learning selection) as target, and the multi-label presence vector of strategies as features. The dataset is used to train a regularised linear regression, LASSO (Tibshirani, 1996), which zeros out the weights for as many strategies as possible. The selection strategies with positive weights in the model represent the subset of relevant strategies.

**Combining the Identified Strategies** After identifying the relevant selection strategies, the strategies are combined to perform the selection. Every sample is assigned an importance score, normalised to zero-one interval, by each of the identified strategies. The final sample score is obtained as a weighted average of its assigned scores using a weighting scheme (i.e., weighted sum of the scores).

In this paper, we evaluate three separate weighting schemes. First, **uniform** combination, assigns the same weight to each of the identified strategies. This represents the simplified version of the ACSESS method that potentially leads to lower performance impact, but also lower computation costs and better transferability as no further optimisation is required. This weighting is also used for the combination of strategies in the previous step. Second, **weighted** combination, assigns a different weight to each strategy, prioritising specific sample properties. We use the final trained weights of the *Datamodels* linear model as the weights for the strategies. As this weighting scheme is dataset and model specific, it should lead to the highest performance impact, at the cost of requiring more extensive computation. Finally, the **weighted combination with random selection** extends the *weighted* combination by introducing a random element to it, as the random selection was identified as a strong baseline that can improve active learning performance (Park et al., 2022). For this weighting scheme, each sample is assigned additional uniformly randomised score. To determine the weight of this score, the random selection is included as an additional strategy to the *Datamodels* method.

## 4 EXPERIMENTS

**Datasets.** The experiments are conducted on representative image and text few-shot classification datasets composed of different tasks with different number of classes and with different characteristics, such as the class imbalance or amount of noisy samples or labels. For the image datasets, we use 8 datasets from the **Meta-Album** (Ullah et al., 2022), specifically **LR_AM.DOG** and **LR_AM.AWA** for animal classification, **VCL.APL** for plane classification, **HUM_ACT.ACT_410** for classification of human actions, **MNF.TEX_DTD** for classification of texture, **MCR.PRT** and **MCR.PNU** for classification of microscope images, and **PLT.FLW** for flower classification. In addition, we use 6 widely-adopted text datasets: **20 News Group** (Lang, 1995) and **News Category** (Misra, 2022) for news category classification and **ATIS** (Hemphill et al., 1990), **Facebook** (Schuster et al., 2019), **HWU-64** (Liu et al., 2021) and **SNIPS** (Coucke et al., 2018) for intent classification. For each dataset, we select a subset of up to 200 labelled samples per class to simulate the limited data setting if needed (relevant mostly for the text classification datasets).

**Few-shot Learning Approaches.** The evaluation is done in a **5-way 5-shot setting** using the gradient few-shot learning on both the image and text datasets and the in-context learning on the text datasets. For gradient few-shot learning, we use representative and widely-used few-shot learning approaches, in order to limit the possible confounding factors. Specifically, we focus on meta-learning approaches, **Prototypical Networks** (Snell et al., 2017) and **MAML** (Finn et al., 2017), and **Few-Shot Fine-Tuning** (Chen et al., 2019). For in-context learning, we use the **Mistral-7B** (Jiang et al., 2023) and **Zephyr-7B** (Tunstall et al., 2023) large language models with 4 bit quantisation.

**Baselines.** The selection strategies are compared with 2 baselines: 1) **Classic selection**, where a new set of samples is randomly selected for each new task, potentially covering the whole datasets when evaluating on enough tasks; and 2) **Random selection**, where only a single set of 5 samples per class is randomly selected at the start and used for every single task, representing a more realistic setting with limited budget. In-context learning is also compared with the recently proposed in-context learning specific strategy **LENS** (Li & Qiu, 2023). Each selection strategy is run 10 times with different random seeds to determine their sensitivity to the effects of randomness (e.g., random initialisation or sample order). In addition, each experiment is repeated 5 times for different data splits (and the choice of the 200 labelled samples) and an average accuracy over these runs is reported (if not specified otherwise). Further experiment setup details are in Appendix E.

### 4.1 IMPACT OF SINGLE-PROPERTY SELECTION STRATEGIES

In this section, our goal is to answer the following research question: ***RQ1: What is the impact of single-property selection strategies on the few-shot learning?*** The objective is to analyse the performance increase of the different selection strategies across various datasets and few-shot learning approaches. The results are shown in Figure 2 and Table 2, aggregated over all the datasets (the results for individual datasets are included in Appendix L). We present the results separately for image and text datasets, as well as for the gradient few-shot learning and in-context learning, as we observed a strong dependence of the strategy benefit on the modality and few-shot learning approach used. Additional ablation studies for the selection strategies are included in Appendix F.

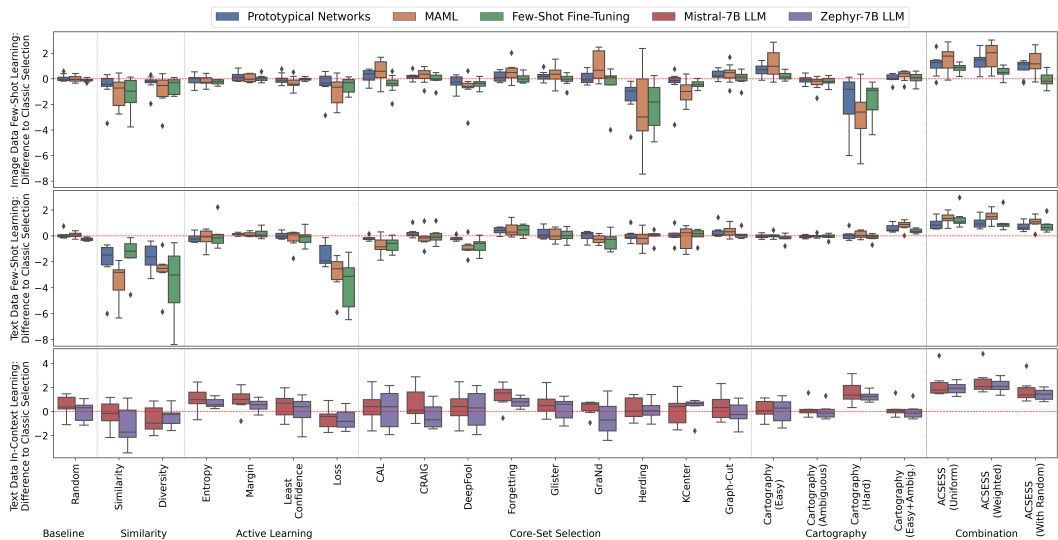

Figure 2: Benefit of the different sample selection strategies, calculated as the difference in accuracy between the specific strategy and the classic few-shot selection, aggregated over the image and text datasets (box plots show the distribution of results across the datasets). The performance of the classic selection is represented as the red dashed line (zero value). The consistently beneficial selection strategies depend on the data modality (image vs. text) and the approach (gradient few-shot learning using meta-learning and fine-tuning vs. in-context learning using large language models). Our proposed method ACSESS consistently leads to improved performance.

**The majority of selection strategies fail to outperform the *Classic selection* in case of gradient few-shot learning**. In most cases, the strategies lead to performance on par or even lower than the *Classic selection* (for example hard to learn samples on image datasets or diverse samples on text datasets). Only specific strategies consistently lead to better performance (e.g., easy to learn samples on image datasets, or *Forgetting* on text datasets). As such, the **random sample selection represents a strong baseline** for the gradient few-shot learning.

**Sample selection is more beneficial for in-context learning.** Many of the selection strategies lead to increase in performance for in-context learning, while only few strategies, such as the popular selection based on *similarity* or *diversity*, under-perform the *Classic selection*. At the same time, **the increase in performance is higher than with gradient few-shot learning**, although with a larger variance in results. In contrast to gradient few-shot learning, the hard to learn samples provide the most benefit for in-context learning. The reason may be that hard to learn samples provide a lot of information beneficial for the classification that cannot be easily leveraged in the gradient few-shot learning due to the samples being hard to learn. However, as there is no explicit "learning" in the in-context learning use of large language models, such samples can be better leveraged.

**The overall benefit of sample selection is higher for noisy datasets and limited for high-quality datasets.** Sample selection leads to a considerable performance increase over the *Classic selection*, on average around 2 percentage points, for the datasets with many noisy samples (as determined by the *Cartography*) or for datasets with large class imbalance (e.g., difference between lowest and median number of samples per class in few hundred samples), such as HUM_ACT.ACT_410, MCR.PRT or MCR.PNU. At the same time, performance increase is quite limited (on average around 0.1 percentage points) for the datasets with few noisy samples or no extensive class imbalance, such as the VCL.APL, PLT.FLW or NewsCategory.

**Learnability property is a strong indicator of quality for few-shot learning**. The sample selection based on how easy it is to learn the samples (*Cartography*) or how often they are forgotten (*Forgetting*) consistently leads to overall the highest increase in performance. The observed improvement is on average as high as 1 percentage point for gradient few-shot learning and 1.5 percentage points for in-context learning over the *Classic selection*.

Table 2: Benefit of the different selection strategies calculated as the difference in accuracy to the *classic* few-shot selection strategy, aggregated over the image and text datasets. The subscript represents the standard deviation of the difference over the aggregated datasets.

| STRATEGY | IMAGE DATA - FSL | | | TEXT DATA - FSL | | | TEXT DATA - ICL | |
|---|---|---|---|---|---|---|---|---|
| | PROTONET | MAML | FT | PROTONET | MAML | FT | MISTRAL | ZEPHYR |
| Random | $+0.06_{0.26}$ | $-0.00_{0.27}$ | $-0.10_{0.13}$ | $+0.10_{0.30}$ | $+0.07_{0.21}$ | $-0.26_{0.11}$ | $+0.46_{0.86}$ | $+0.01_{0.82}$ |
| LENS | N/A | N/A | N/A | N/A | N/A | N/A | $+1.73_{0.70}$ | $+1.42_{0.32}$ |
| Similarity | $-0.64_{1.14}$ | $-1.05_{1.15}$ | $-1.21_{1.24}$ | $-2.15_{1.82}$ | $-3.51_{1.50}$ | $-1.56_{1.45}$ | $-0.24_{1.11}$ | $-1.22_{1.61}$ |
| Diversity | $-0.35_{0.65}$ | $-0.93_{1.22}$ | $-0.57_{0.58}$ | $-1.65_{1.02}$ | $-2.77_{1.55}$ | $-3.65_{2.69}$ | $-0.66_{1.08}$ | $-0.42_{0.80}$ |
| Entropy | $-0.12_{0.43}$ | $-0.17_{0.37}$ | $-0.26_{0.18}$ | $-0.17_{0.33}$ | $-0.20_{0.68}$ | $+0.07_{1.03}$ | $+1.02_{0.98}$ | $+0.70_{0.38}$ |
| Margin | $+0.15_{0.34}$ | $+0.05_{0.29}$ | $+0.02_{0.26}$ | $+0.12_{0.10}$ | $+0.11_{0.17}$ | $+0.18_{0.35}$ | $+0.93_{0.94}$ | $+0.51_{0.50}$ |
| Least Confidence | $-0.05_{0.41}$ | $-0.31_{0.46}$ | $-0.03_{0.12}$ | $+0.02_{0.25}$ | $-0.25_{0.72}$ | $-0.14_{0.59}$ | $+0.49_{1.03}$ | $+0.04_{1.17}$ |
| Loss | $-0.44_{1.01}$ | $-1.02_{1.07}$ | $-0.53_{0.57}$ | $-1.49_{0.89}$ | $-3.00_{1.46}$ | $-3.75_{1.92}$ | $-0.56_{0.88}$ | $-0.65_{0.83}$ |
| CAL | $+0.19_{0.54}$ | $+0.54_{0.88}$ | $-0.43_{0.72}$ | $-0.18_{0.18}$ | $-0.76_{0.69}$ | $-0.69_{0.54}$ | $+0.39_{1.27}$ | $+0.18_{1.56}$ |
| CRAIG | $+0.18_{0.29}$ | $+0.26_{0.60}$ | $-0.04_{0.45}$ | $+0.22_{0.40}$ | $-0.15_{0.69}$ | $-0.00_{0.62}$ | $+0.66_{1.36}$ | $-0.38_{1.03}$ |
| DeepFool | $-0.29_{0.51}$ | $-0.77_{1.11}$ | $-0.41_{0.36}$ | $-0.18_{0.18}$ | $-0.90_{0.65}$ | $-0.76_{0.59}$ | $+0.40_{1.29}$ | $+0.18_{1.52}$ |
| Forgetting | $+0.16_{0.37}$ | $+0.53_{0.72}$ | $+0.03_{0.34}$ | $+0.39_{0.26}$ | $+0.49_{0.54}$ | $+0.41_{0.42}$ | $+1.27_{0.96}$ | $+0.77_{0.41}$ |
| Glister | $+0.20_{0.37}$ | $+0.33_{0.73}$ | $-0.10_{0.45}$ | $+0.19_{0.43}$ | $+0.06_{0.49}$ | $+0.03_{0.47}$ | $+0.64_{0.98}$ | $+0.21_{0.92}$ |
| GraNd | $+0.09_{0.49}$ | $+0.95_{1.11}$ | $-0.42_{1.41}$ | $-0.01_{0.38}$ | $-0.27_{0.34}$ | $-0.40_{0.67}$ | $+0.29_{0.62}$ | $-0.54_{1.41}$ |
| Herding | $-1.43_{1.31}$ | $-2.36_{2.94}$ | $-2.14_{1.78}$ | $+0.03_{0.51}$ | $-0.24_{0.69}$ | $-0.04_{0.45}$ | $+0.24_{0.91}$ | $+0.11_{0.78}$ |
| KCenter | $-0.45_{1.24}$ | $-1.03_{0.87}$ | $-0.42_{0.31}$ | $+0.04_{0.58}$ | $-0.15_{0.91}$ | $+0.03_{0.49}$ | $+0.13_{1.24}$ | $+0.31_{0.88}$ |
| Graph-Cut | $+0.32_{0.36}$ | $+0.40_{0.75}$ | $+0.05_{0.53}$ | $+0.34_{0.50}$ | $+0.35_{0.44}$ | $+0.10_{0.33}$ | $+0.42_{1.11}$ | $-0.17_{0.92}$ |
| Cartography$_{Easy}$ | $+0.65_{0.50}$ | $+1.17_{1.11}$ | $+0.21_{0.28}$ | $-0.04_{0.17}$ | $+0.02_{0.21}$ | $-0.19_{0.31}$ | $+0.16_{0.76}$ | $+0.06_{0.98}$ |
| Cartography$_{Ambiguous}$ | $-0.08_{0.36}$ | $-0.36_{0.50}$ | $-0.21_{0.32}$ | $-0.06_{0.12}$ | $-0.01_{0.13}$ | $-0.06_{0.21}$ | $+0.21_{0.64}$ | $+0.02_{0.64}$ |
| Cartography$_{Hard}$ | $-2.42_{3.35}$ | $-3.03_{2.26}$ | $-1.71_{1.47}$ | $+0.05_{0.37}$ | $+0.08_{0.29}$ | $-0.14_{0.28}$ | $+1.60_{0.93}$ | $+1.26_{0.39}$ |
| Cartography$_{Easy+Ambig.}$ | $+0.05_{0.33}$ | $+0.24_{0.40}$ | $+0.04_{0.40}$ | $+0.59_{0.30}$ | $+0.78_{0.39}$ | $+0.37_{0.16}$ | $+0.21_{0.64}$ | $+0.02_{0.64}$ |
| ACSESS$_{Uniform}$ | $+1.12_{0.80}$ | $+1.59_{0.97}$ | $\mathbf{+0.81_{0.35}}$ | $+0.92_{0.41}$ | $+1.34_{0.45}$ | $\mathbf{+1.36_{0.75}}$ | $+2.30_{1.11}$ | $+1.93_{0.48}$ |
| ACSESS$_{Weighted}$ | $\mathbf{+1.31_{0.75}}$ | $\mathbf{+1.83_{0.97}}$ | $+0.49_{0.41}$ | $\mathbf{+1.02_{0.44}}$ | $\mathbf{+1.49_{0.47}}$ | $+1.07_{0.69}$ | $\mathbf{+2.55_{1.08}}$ | $\mathbf{+2.15_{0.53}}$ |
| ACSESS$_{With Random}$ | $+0.85_{0.65}$ | $+1.32_{0.91}$ | $-0.06_{0.56}$ | $+0.72_{0.33}$ | $+1.03_{0.48}$ | $+0.80_{0.54}$ | $+1.78_{0.98}$ | $+1.42_{0.45}$ |

## 4.2 ACSESS RESULTS AND CHARACTERISTICS

In this section, we answer the following research question: **RQ2: *How does the ACSESS method perform in comparison to the single-property strategies?*** Besides analysing and comparing the benefit with other strategies, we analyse the different configurations of our method (methods for strategy identification and weighting schemes).

**The ACSESS method consistently leads to performance increases across all few-shot learning approaches.** Our proposed method outperforms all the investigated single-property strategies, leading to an average performance increase of up to $1.8$ percentage points for gradient few-shot learning and $2.5$ percentage points for in-context learning. Comparing with the in-context learning specific strategy LENS (Li & Qiu, 2023) (see Table 2 and Figure 6 in Appendix G for comparison), which also outperforms all single-property strategies, we observe the increase of ACSESS is statistically more significant (p-value of $0.0002$ using the Wilcoxon signed-rank test).

**The ACSESS method identifies samples with complementary properties, with higher priority on learnability.** For the different data modalities (image vs. text) and the type of few-shot learning (gradient vs. in-context learning), our method identifies and combines different set of selection strategies. For the image datasets, the combination of easy to learn samples (*Cartography*) that are located on the decision boundary (*Margin*) provides the most benefit. For the gradient few-shot learning approaches on text datasets, the combination of easy and ambiguous samples (*Cartography*) that are least often forgotten (*Forgetting*) on the decision boundary (*Margin*) that provide additional informativeness from the *Graph-Cut* strategy provides the most benefit. In case of in-context learning, the combination of hard to learn samples (*Cartography*) that are least often forgotten (*Forgetting*) on the decision boundary (*Margin*) and with highest *Entropy* provide the most benefit. Further analysing the identified strategies and their weights, we find the **ACSESS method gives higher importance to samples that are *learnable*, lower importance to *informative* samples** (but with multiple sources for both), **while the strategies that focus on *representativeness* are not included at all**. This further reinforces the finding that learnability is a strong indicator of quality for few-shot learning.

**The dataset and approach specific weighting leads to the best performance**. The only exception is the Few-Shot Fine-Tuning approach, where the uniform weighting shows the most benefit. At

the same time, the weighting strategy that incorporates the random selection often leads to worse performance and higher variance in the results. Even though the uniform combination is computationally less expensive than the weighted combination, its performance increase is only slightly lower (average difference of $0.10 - 0.25$). As such, the **uniform weighting represents a good trade-off** between the performance increase and the computation costs, making it in general a better choice.

### 4.3 ABLATION STUDY: EFFECTS OF VARIABLE NUMBER OF SHOTS AND DATASET SIZE

In this section, we answer the following research question: ***RQ3:*** *How do the number of selected shots and the size of the dataset affect the benefit of sample selection?* We perform two ablation studies to investigate how the benefits of sample selection strategies change as we: 1) increase or decrease the number of selected shots; and 2) decrease the number of available labelled samples (i.e., the size of the dataset). For the first ablation study, we apply the *ACSESS* method and *Random* selection to select a different number of shots and compare their benefit with training on all available samples (results aggregated over all datasets are in Figure 3 and for individual datasets in Appendix I). For the second ablation study, we apply the *ACSESS* method and *Random* selection to select 5 samples per class from dataset subsets of different sizes and compare their benefit to the selection from the full dataset (results aggregated over all datasets are in Figure 4 and for individual datasets in Appendix J).

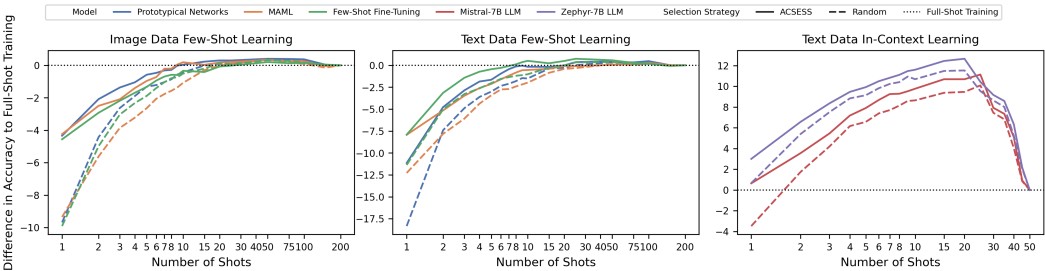

Figure 3: The change in benefit of selection strategies based on the number of shots used, aggregated over datasets. The change is calculated as a difference in accuracy to a setting using all available samples for training (full-shot training with 200 shots or 50 shots for in-context learning). When using a lower number of shots, the benefit of selection strategies over random selection is more significant. When using a larger number of shots, the benefit becomes negligible. In addition, the performance of full-shot training is matched, or in many cases exceeded, using smaller number of shots.

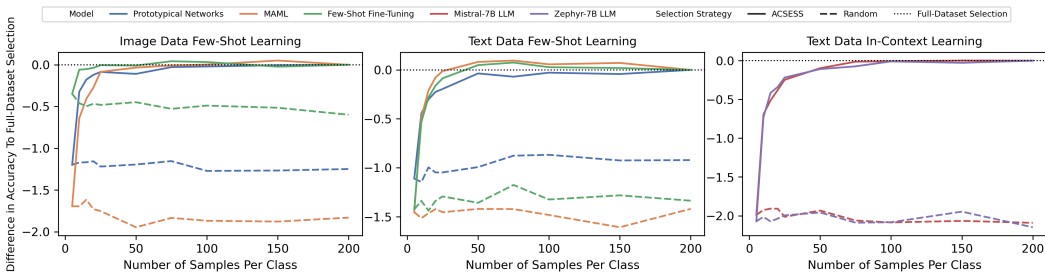

Figure 4: The difference in accuracy of selection strategies when using dataset subsets of different sizes, aggregated over the datasets. The difference is calculated as a comparison of the 5-shot selection using the dataset subset and the full dataset. Using only a fraction of dataset, the sample selection outperforms random baseline and quickly achieves similar performance to full dataset selection.

**The selection strategies provide more benefit on lower number of shots.** On the 1-shot setting, the difference in accuracy between the *Random* selection and the *ACSESS* strategy can be as high as $5$, $7$ or $4$ percentage points respectively for the few-shot learning on image data and text data, and

in-context learning on text data. **Increasing the number of selected shots leads to diminished impact**, as the *ACSESS* strategy regresses to random selection when selecting more than 30-40 shots. For more detailed analysis we compare all the investigated strategies in the 5-shot and 10-shot setting (distribution of results is shown in Figure 7 in Appendix H). This comparison further reinforces our findings, as we find that the difference to random selection is statistically significantly higher in the 5-shot setting (p-value of $1.93e - 07$ using the Wilcoxon signed-rank test).

**Impact of selection strategies and further increase of the number of shots is negligible after certain point.** For the gradient few-shot learning, the highest performance is achieved at lower number of shots (20) and then remains more or less constant (with slight decrease in some cases). For the in-context learning, the highest performance is achieved at 20-25 shots, and afterwards we observe a significant decrease, achieving similar performance to the 1-shot setting when using 50-shots. The main reason for this drop-off may be the limited context size of the large language models, making the model degenerate to random guessing as the cross-over is text length dependent. As such, the focus on **selecting high quality samples is paramount when selecting only a few labelled samples per class**, while the random selection can be safely used with higher number of shots. However, **labelling more samples and using them as additional shots may not always lead to better performance**.

However, **the benefit of curating smaller set of high-quality samples over training on all available samples is higher for noisy and imbalanced datasets.** On these datasets, such as HUM_ACT.ACT_410, MCR.PRT, MCR.PNU, ATIS, HWU-64 or Facebook, the highest performance is achieved at 20 samples and decreases as we select more shots. On the other hand, **on the datasets with low number of noisy samples or more balanced classes, the sample selection leads only to reduction in computation cost, with negligible performance improvements.** On these datasets, such as LR_AM.DOG, LR_AM.AWA, VCL.APL or NewsCategory, the sample selection achieves performance on par with full-shot training, while using only around $50 - 100$ shots.

**Sample selection is beneficial even when using dataset with low number of samples.** Using only 10% of the dataset (20 samples per class) for gradient few-shot learning or 25% (50 samples per class), sample selection can still identify samples that lead to similar performance as is achieved when using the full dataset. Decreasing the size further, the benefit of sample selection starts to decrease as well. Using only 10 samples per class, the benefit decreases by approximately $20 - 40\%$ (e.g., from performance increase of 2 percentage points to only 1.3), and then quickly deteriorates to the *Random* selection (as the number of samples to choose from is too low).

## 5 CONCLUSION

In this paper, we focus on a data-centric view of few-shot learning and investigate the impact of curating a small set of informative and high-quality samples on the success of few-shot learning. To this end, we investigate the performance benefit of a larger number of different sample selection strategies. To leverage the strengths of these strategies, we propose the *ACSESS* method for the automatic combination of the sample selection strategies to select samples with complementary properties. The experimental results show that although the individual sample selection strategies may not always be beneficial, their combination using our proposed method consistently leads to better performance. However, we observe that the overall benefit of sample selection is higher for imbalanced datasets with noisy samples and limited for datasets containing only high-quality samples. Curiously, we observe that the *learnability* of the samples is more important for few-shot learning than their *informativeness* or *representativeness*. Based on these results, the uniform combination of *Cartography*, *Forgetting* and *Margin* selection strategies is a good alternative transferable across all settings if running the full method is not feasible. Analysing the behaviour of the sample selection when changing the number of selected shots and the size of the dataset, we find that the sample selection provides the most benefit with a low number of shots, while with a higher number of shots all strategies regress to random guessing. In addition, after a certain point, increasing the number of shots used for training no longer leads to improvement in performance of few-shot learning, but may lead to a decrease in performance in some cases, especially for in-context learning. Overall, the benefit of selecting a subset of samples over full-shot training is in reduction of computation cost, while providing performance increase on the imbalanced and noisy datasets. Finally, even when the datasets contain only a few available samples per class, curating the samples further can often still lead to benefit in terms of performance and computation costs regardless of dataset characteristics.

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

## A  ETHICAL CONSIDERATIONS AND IMPACT STATEMENT

The experiments in this paper work with publicly available datasets, citing the original authors of each dataset (with the exception of the datasets contained in MetaAlbum, where we cite the creators of the aggregated dataset). All the datasets are used in accordance to their license and terms and conditions of use. We do not work with any personally identifiable information or offensive content and perform no crowdsourcing nor further data annotation. In addition, we are not aware of any potential ethical harms or negative societal impacts of our work, apart from the ones related to the advancement of the field of Machine Learning and Few-Shot learning (including large language models and in-context learning), such as the use of large amount of computation resources. Finally, we follow the license terms for all the models we use – all models and datasets allow their use as part of research. The large language models (Mistral and Zephyr), which we use for the experiments, may contain biases and may generate potentially offensive or harmful content. However, the authors of the models already reduce this potential bias as much as possible when training the models, while at the same time we limit the output to few tokens and do not release any output of these models, which should further reduce the potential bias and negative impact.

**Impact Statement: CO2 Emissions Related to Experiments**    The experiments presented in this paper used larger amount of compute resources in order to evaluate all the sample selection strategies for all the different few-shot learning approaches. Overall, all the experiments (including preliminary experiments whose results are not reported in this paper) were conducted using a private infrastructure, which has a carbon efficiency of 0.432 $kgCO_2eq$/kWh. A cumulative of approximately 2000 hours of computation was performed on hardware of type A100 PCIe 40GB (TDP of 250W). Total emissions are estimated to be 216 $kgCO_2eq$ of which 0 percent were directly offset. These estimations were conducted using the MachineLearning Impact calculator presented in Lacoste et al. (2019). Whenever possible, we tried to reduce the compute resources used as much as possible. The most compute resources were used by the large language models – Mistral-7B and Zephyr-7B and their further tuning. To reduce the computation costs, we use the 4-bit quantised versions of these models and in cases when the further tuning is required, we fine-tune only the last classification layer (keeping the rest frozen). In addition, for gradient few-shot learning approaches, we opt for smaller models that do not require such an extensive computation costs, while still providing relevant results. Finally, whenever possible, we reuse the already performed computations for the selection strategies as well as using the compute optimised version of our proposed method (more information included in Appendix C), such as generating the feature representation for the different samples only a single time and re-using it whenever possible (i.e., using pre-trained models without further tuning for these representations).

## B   LIMITATIONS

In the experiments for evaluating the impact of different sample selection strategies, we focused on such settings, where the impact of sample selection may be lower – within-domain few-shot learning with balanced-shot selection – despite that we observed significant differences in the achieved performance. First of all, in within-domain few-shot learning, the samples are selected from the same domain, thus the difference in impact and role of the individual samples may not be as high. We may expect that the impact of sample selection in cross-domain few-shot learning may be higher and lead to even larger performance deviations. In addition, we focus on a balanced-shot setting, where we select the same number of shots for each class, leading to lower impact of strategies for different classes. Investigating the impact of sample selection across these additional settings is out of scope of this paper and may be a subject of future research.

To diminish undesirable randomness effects, we report the results as an average over 5 repeats of data splitting, 10 repeated runs of the selection strategies (resulting in 50 overall evaluation runs), with each run evaluated on 600 (or 300 for in-context learning) tasks. Such setting limits the variance in results, but may also obscure the benefit of selection strategies to a certain extent. On the other hand, although we focus only on a 5-way few-shot learning setting (using only 5 classes per task), we do not expect significant change in result for most of the few-shot learning approaches when using higher number of classes (such as 10-way or 20-way classification). The only exception is the in-context learning, where we observed the dependence on input size and context width of the models. With higher number of classes, the observed performance of large language models would start to deteriorate to the performance that can be achieved with a low number of shots.

Similarly to other research papers focusing on sample selection and few-shot learning via in-context learning, we assume an availability of large-enough annotated dataset from which we select the sample using different strategies – we use up to 200 samples per class for each dataset for the main experiments. As such, we also have larger validation or held-out set available for evaluating the selection strategies and hyperparameters for our proposed methods or for the approaches themselves. If we considered the true few-shot learning setting, where we have only unlabelled data available, only the active learning strategies (i.e., Entropy, Margin and Least Confidence) could be used, as they work with unlabelled data only, or a different approach for sample selection should be employed, such as using weak-labels to select the best samples, or assign every single class to all the labels and perform the sample selection that way to select both best performing samples as well as the class for them (as it is done by Chang & Jia (2023)). **We address this (potentially most significant) limitation by running ablation studies to observe the benefit of sample selection** (using our proposed method and random selection) **when using smaller subsets of data** (i.e., selecting samples using only fraction of the dataset) **or using different number of shots per class.**

Before the experiments, we run hyperparameter optimisation for the gradient few-shot learning models and perform basic prompt-engineering for the in-context learning approaches. We follow the recommendations for good prompt engineering, using the meta-tags defined for each model (which lead to better performance of the models), while also not using overly-simple or overly-complicated prompts. The prompts we use are created based on dataset description, the prompts used in related works and the formats recommended for different models (e.g., taking inspiration from Sun et al. (2023)). As we do not perform any extensive or automatic prompt-engineering, these prompts may lead to lowered performance and in some cases also lower impact of the sample selection for the few-shot learning setting. However, running an extensive prompt-engineering would introduce another controllable parameter which could potentially introduce further bias into the results (e.g., different selection strategies showing different impact for different prompt formats).

Finally, we also identify factors that we consider to be sufficiently covered, but where the further extension could lead to more generalisable results (nevertheless, also requiring a considerable amount of additional computational resources), such as including more selection strategies, evaluating on more datasets covering other classification tasks (mainly in regards to text datasets) or using larger number of base models for meta-learning and few-shot fine-tuning, or large language models for in-context learning. First of all, although we focus on the major selection strategies and cover majority of the popular sample selection strategies, which are general for any few-shot learning approach, the work can be extended to include additional in-context learning specific strategies (that perform sample selection for each test sample separately) or that train a dataset-specific sample retriever

for selection. However, we consider these approaches to be out-of-scope for this paper, as they are not general for any few-shot learning approach (i.e., performing sample selection separately for each test sample is not really feasible or beneficial for gradient few-shot learning) or would require significantly larger number of labelled samples to work correctly. Second, even though we run the experiments on 8 image datasets and 6 text datasets, covering different classification tasks with different number of classes, the work could be potentially extended to even additional datasets and classification tasks, mainly in case of text modality where we are limited by the number of classes the datasets contain (to sufficiently explore few-shot learning). In addition, we explicitly do not evaluate the experiments on the image datasets commonly used for few-shot learning (such as miniImagenet), as these datasets were used for pre-training the classification model and contain large number of samples. Instead we focus on datasets more representative of the few-shot setting, with different dataset characteristics (inlcuding class imbalance and containing noisy samples). Finally, although we cover all the main groups of few-shot learning approaches and focus on the representative and well-understood models, the work could be extended to cover additional meta-learning approaches (where we focus on two of them), additional base models for meta-learning and few-shot fine-tuning (we focus on one model for image data and one model for text data) and additional large language models for in-context learning (we focus on two recent large language models that show consistent performance, Mistral and Zephyr). However, we do not expect significantly different results and findings, only an improvement of their generalisability.

## C  ACSESS METHOD SUPPLEMENTARY DETAILS

In this appendix, we provide and discuss supplementary details of our proposed strategy. This includes a detailed steps of the forward selection in Algorithm 2, backward selection in Algorithm 3 and datamodels selection in Algorithm 4. In addition, we provide low-level details how the method works, suggestions for using the method in order to consider its generalisability and reducing its model/dataset dependence, the computational costs of the method and how we reduce them to provide compute optimised version of our method, and, finally, suggestions for selecting the best configuration of the ACSESS method based on the compute-performance trade-off of these configurations. Some parts of this section are based on the observations from our experiments.

---

**Algorithm 2** Forward Selection

---
**Require:** $\mathbf{S}$ - set of sample selection strategies
 1: $\mathbf{S_{final}} \leftarrow \{\}$
 2: $best\_score$ = -1
 3: $strategy$ = First strategy from $\mathbf{S}$
 4: **while** $strategy$ is None **or** no new strategy was added to the list **or** $\mathbf{S}$ is empty **do**
 5:    $strategy$ = None
 6:    **for all** $s_i$ in $\mathbf{S}$ **do**
 7:      Create $\mathbf{S_{temp}}$ by uniformly combining $\mathbf{S_{final}}$ and $s_i$
 8:      Select samples using the combination of strategies $\mathbf{S_{temp}}$
 9:      Calculate $score_i$ by evaluating the selected samples on few-shot model
10:      **if** $score_i > best\_score$ **then**
11:        $best\_score = score_i$
12:        $strategy = s_i$
13:      **end if**
14:    **end for**
15:    **if** New strategy that improves performance was found **then**
16:      Add found strategy to the final set of strategies $\mathbf{S_{final}}$
17:      Remove found strategy from the overall list of strategies $\mathbf{S}$
18:    **end if**
19: **end while**
20: **return** $\mathbf{S_{final}}$

---

**Considerations for using the ACSESS method and reducing its model/dataset dependence**
Similarly to hyperparameter optimisation, we suggest to identify and combine the different strategies using the validation set to prevent any data leakage (i.e., evaluating the performance of different strategies using the validation set). At its core, the method is designed to be run separately for each model and dataset. Although such setting leads to the most benefit and highest increase in

---

**Algorithm 3** Backward Selection

---

**Require:** **S** - set of sample selection strategies
1: $\mathbf{S_{final}}$ = uniform combination of strategies from **S**
2: Evaluate the uniform combination of all strategies $\mathbf{S_{final}}$ to obtain $best\_score$
3: $strategy$ = First strategy from **S**
4: **while** $strategy$ is None **or** no new strategy was removed from the list **do**
5:     $strategy$ = None
6:     **for all** $s_i$ in **S do**
7:         Create $\mathbf{S_{temp}}$ by removing strategy $s_i$ from the uniform combination of $\mathbf{S_{final}}$
8:         Select samples using the combination of strategies $\mathbf{S_{temp}}$
9:         Calculate $score_i$ by evaluating the selected samples on few-shot model
10:        **if** $score_i > best\_score$ **then**
11:            $best\_score = score_i$
12:            $strategy$ = $s_i$
13:        **end if**
14:    **end for**
15:    **if** New strategy that improves performance was found **then**
16:        Remove found strategy from the final set of strategies $\mathbf{S_{final}}$
17:    **end if**
18: **end while**
19: **return** $\mathbf{S_{final}}$

---

**Algorithm 4** Datamodels Selection

---

**Require:** **S** - set of sample selection strategies
**Require:** $N$ - number of random combination to create
**Require:** $score$ - performance of the few-shot model on the baseline setting
1: $\mathbf{S_{final}} \leftarrow \{\}$
2: Create empty list of differences to baselines $Diff$
3: Create empty list of multi-label presence vector of strategies $Strategies$
4: Create a set **C** containing $N$ unique, random combinations of strategies from **S**, ensuring each strategy is contained at least 5 times
5: **for all** $c_i$ in **C do**
6:     Select samples using the combination of strategies $c_i$
7:     Calculate $score_i$ by evaluating the selected samples on few-shot model
8:     Calculate $diff_i = score_i$ - $score$
9:     Add $diff_i$ to the list of differences to baseline $Diff$
10:    Create a new multi-label presence vector using strategies from $c_i$ and add it to the list $Strategies$
11: **end for**
12: Train a $LASSO$ linear regression using $Strategies$ as features and $Diff$ as targets
13: Obtain vector of weights $\vec{W}$, each corresponding to one strategy in the multi-label presence vector
14: Select strategies with positive weights in $\vec{W}$ and add them to $\mathbf{S_{final}}$
15: Keep only weights of the selected strategies in $\vec{W}$ to obtain $\vec{W}_{final}$
16: **return** $\mathbf{S_{final}}, \vec{W}_{final}$

---

performance, this makes it significantly model/dataset dependent and requires significantly more computation resources. In order to reduce the model/dataset dependence, the different steps of the method be run across multiple models or datasets. This may be best illustrated on the identification of strategies and further weighting using the Datamodels method. Instead of having a separate set of combinations of strategies for each dataset a model, we can concatenate them together to create a single dataset. After training the LASSO method on this concatenated dataset, we obtain the relevant strategies and their weights for all the dataset and models. Similar modification can be applied to the forward and backward selection as well, where we evaluate the added/removed strategy on all the models/datasets, aggregate the score and select the strategies on this aggregated score. However, as we observed a strong dependence on the few-shot approach and the data modality, we suggest to run the ACSESS method separately for each few-shot learning approach and modality. In our experiments we also follow this suggestion and aggregate only over the datasets, as we have noticed only negligible changes when running the ACSESS method separately for each dataset (i.e., in some cases, the set of identified strategies we report in the results included 1 more strategy that performed really well on that specific dataset). Running the ACSESS strategy on even higher aggregation (over

all the approaches, models and modalities) is not recommended as our preliminary experiments indicate that it would not lead to significant performance benefit and the method could fail to identify any relevant selection strategy in some cases. Finally, foregoing the specific model/dataset weighted combination and instead using the uniform combination (or the weights provided by the Datamodels approach across multiple datasets) can increase the transferability and generalisability of the ACSESS method further.

**Considerations for the computation costs of the ACSESS method**     Even though the steps of ACSESS method (mainly the identification of relevant strategies) may require a significant computation resources, we provide suggestions on how to use the method in a compute-efficient way to reduce these costs as much as possible. First, as we need to identify which strategies to combine, each of them needs to be run and evaluated. The main computation cost for majority of the selection strategies is the training of the model that is then either directly used for sample selection or is used to provide features for each sample that are used for the selection. As such, majority of the selection strategies can be run at the same time, reusing the same trained model (i.e., we need to train the model only a single time and use it across majority of the strategies). The exception are the iterative strategies (active learning) that need to be run independently, as they train the model only on a fraction of samples, which changes for each run of the strategy instead of using all the available samples for training. Second, to identify the most relevant combination of strategies, the different strategies are combined, this combination is used to select the samples which are then used to evaluate the combination. However, as we work with strategies that assign scores to each sample, we need to run each selection strategy only a single time and then use the sample scores for combination. As such, each combination requires only the evaluation on the few-shot learning approaches, which is computationally quite cheap as there is only a quick adaptation with few samples (or no adaptation for in-context learning) followed by inference. However, the cost of this evaluation step is still dependent on the number of samples and the model used (i.e., training on 25 samples vs. full dataset; inference on smaller vs. larger models). Finally, the computation cost can be further reduced by foregoing the model and dataset specific tuning (e.g., using the ACSESS method on multiple datasets at the same time).

**Choosing the best configuration for the ACSESS method**     Based on the results of our experiments, to get the best performance benefit of the ACSESS method, running the full strategy identification, either separately for each model and dataset or aggregated across datasets, and using the weighted combination of these strategies should be used. This configuration provides the best results across all datasets, approaches and models, with the exception of Few-Shot Fine-Tuning. However, this may require extensive computation resources. If such resources are not available, or if we want to quickly determine the potential benefit of the sample selection, using a uniform combination of the *Cartography* and *Margin* (and in some cases *Forgetting*) selection strategies provides a good approximation. This configuration does not incur any additional computation costs, while providing consistent improvement on all the datasets and models, which is only slightly lower than when using the model/dataset specific configuration. Although, running the full method should always provide the most benefit, this configuration (uniform combination of the 3 best performing strategies identified in our experiments) is a good alternative that is transferable across all settings.

# D    SAMPLE SELECTION STRATEGIES DETAILS

In this appendix, we provide further details about the different single-property sample selection strategies that we evaluate in this paper, as well as for the baselines. In addition, we specify the hyperparameters of these methods and how we use them for selecting the set of samples.

The idea behind the **uncertainty-based** and **active learning** selection strategies is that the most informative samples with higher impact are the ones that the model is least certain about (Coleman et al., 2020; Park et al., 2022). These strategies select samples iteratively over multiple steps. In our experiments, we select 1 sample for every class in individual step. In addition, the first sample for every class is chosen randomly, as such setting was observed to lead to better active learning results (Park et al., 2022). When using these methods for combination, the samples that are selected after running all the required steps (e.g., 4 steps when choosing 5-shots) use the score assigned to them at their respective step when they were chosen, while the remaining samples are assigned the

average score over all the selection steps. The **Entropy** strategy measures the entropy for each of the samples over the predicted class probabilities (which is used as the sample score) and selects the sample with highest entropy. The **Margin (Breaking Ties)** strategy assigns to each sample the difference between the probability of the two most probable classes. The samples with lowest such difference are selected as those are the ones the model is most uncertain about (and which can be considered on the decision boundary of the model). The **Least Confidence** strategy simply selects the samples where the most likely class has the lowest probability assigned to it. Finally, the **Loss** strategy selects samples with the highest loss.

The idea behind the core-set selection strategies is to select subset of samples that are representative of the whole dataset (i.e., the subset has similar properties to the full set). The included strategies select samples based on different criteria, such as error/loss, gradient or submodularity. These strategies select the full set of samples at the same time according to the assigned scores. As such, when using these strategies for combination, the full sample scores are used from this single iteration.

The **Decision Boundary** based methods select samples the are close to the decision boundary, as these represent the samples that are hard to separate and should provide the most information to the models. The **Contrastive Active Learning (CAL)** (Margatina et al., 2021) selects the samples whose predictive likelihood diverges the most from their neighbourhood as determined by the KL divergence. The **DeepFool** strategy (Ducoffe & Precioso, 2018) approximates the samples on boundary by perturbing the samples until the predicted class changes and selects the samples that require the smallest perturbation.

The idea behind **Error or Loss** based methods is that the samples that contribute more loss or error to the training of model have higher importance and should contribute more to the training when selected. The **GraNd** strategy (Paul et al., 2021) calculates the average contribution of each sample to the decline of the training loss at early stage of training across several independent training runs (calculating gradient norm expectation of the sample). The **Forgetting** strategy (Toneva et al., 2018) simply calculates how often the specific sample is incorrectly classified after being classified correctly in the previous epoch. The number of such forgetting events is used as the sample score and samples that are forgotten the most often (or least often) are selected. In our evaluation of single-property strategies, we consider only the setting of most often forgotten samples (as it has shown the best performance in the preliminary experiments). However, when using the strategy for combination, we also consider the setting of selecting the least forgotten ones.

The **Submodularity** based methods use submodular functions (Iyer & Bilmes, 2013; Iyer et al., 2021), which naturally measure the informativeness and diversity of a subset of samples. We evaluate the selection based on **Graph-Cut**, as the method performed the best from all the different submodular functions in the preliminary experiments.

The **Geometry** based methods assume that samples that are close in the feature space have similar properties. As such, these methods try to remove samples that are close to each other, as they provide redundant information. Removing enough of such samples then leads to a small subset of sample that cover the properties of the whole dataset. The **Herding** strategy (Welling, 2009; Chen et al., 2010) iteratively and greedily adds one sample to the subset which minimises the distance between the centre of the subset and the centre of the whole dataset. On the other hand, the **KCenter** strategy (Sener & Savarese, 2018; Agarwal et al., 2020) selects sample that minimises the largest distance between each sample already in the subset and its closest sample not yet included in subset.

The **Gradient Matching** based methods try to find a subset of samples whose gradient can imitate the whole dataset. The **CRAIG** strategy (Mirzasoleiman et al., 2020) converts the problem into maximisation of a monotone submodular function which is then solved greedily.

Finally, the **Bilevel Optimisation** based methods convert the problem into a bilevel optimisation, with different objectives in the outer and inner level. The **Glister** strategy (Killamsetty et al., 2021) uses a validation set on the outer optimisation, where the subset selection is used as objective, and log-likelihood in the inner optimisation, which handles model parameter optimisation.

In addition, we also evaluate and use sample selection based on the **Cartography** (Swayamdipta et al., 2020; Zhang & Plank, 2021), which measures how easy or hard it is to learn different samples. This ease of learning is determined by observing the training dynamics of all the samples across few epochs, looking at the average confidence/probability of the correct class and the variance of this

confidence. The samples with high confidence and low variance are considered to be the *easy* to learn samples. At the same time the samples with small confidence and small to medium variance are considered the *hard* to learn ones. The remaining samples are considered to be *ambiguous* (medium confidence or samples with high variance). We explore four different settings in our experiments and choose the samples accordingly: 1) **Easy** samples, where we sort the samples based on confidence and choose the top-K samples with highest confidence for each class; 2) **Hard** samples, where we sort the samples based on confidence and choose the bottom-K samples for each class (i.e., the lowest confidence samples); 3) **Ambiguous**, where we first calculate average confidence and standard deviation of all the samples, select the samples whose confidence is close around the average confidence (in the interval defined by 1 standard deviation around the average confidence), and then randomly sample from them; and 4) **Easy + Ambiguous**, where we choose half of the samples from the *easy* set and the other half from *ambiguous* set.

All the approaches mentioned so far perform the selection using a model that is trained for some epochs. The number of epochs the training is done for significantly affects the outcome of different strategies. For this reason, we search for an optimal number of epochs by evaluating the selection strategies and the samples they select on a single run. The optimal number of epochs determined by this search is 10% of the full epochs used for the few-shot fine-tuning models. This number represents a good trade-off between the computational cost of the methods and the quality of samples. Running the training for full number of epochs may result in better selection of samples, but we consider such setting to not be representative. In addition, we use the same number of epochs for each strategy to keep the comparison as representative as possible. In addition, we also search for an optimal set of hyperparameters for the selection strategies (such as the learning rate, the optimiser used or batch size). The optimal set of hyperparameters that are used throughout the experiments is the use of the Adam optimiser with Cosine Annealing learning rate scheduler, with learning rate of 0.01 for image data and 0.0001 for text data and a batch size of 64.

To keep the results as comparable as possible, each selection strategy uses the same base model as the one used in the main experiments, i.e., ResNet-18 (He et al., 2016) pretrained on Imagenet dataset (Deng et al., 2009) for image data and pretrained BERT-base (Devlin et al., 2019) for text data in case of gradient few-shot learning, and the same large language models (Mistral (Jiang et al., 2023) and Zephyr (Tunstall et al., 2023)), but with only their last classification layer being learned, for in-context learning. In case the selection strategy requires sample features, we use the output of last fully-connected layer (the input of classification layer) as the feature vector similar to Guo et al. (2022).

We also evaluate the sample selection based on **Similarity** and **Diversity** as it represents popular approach for in-context learning. To select the samples, we use the cosine similarity between the feature representation (obtained from the penultimate layer) of the samples. To select the samples for gradient few-shot learning, we first randomly select one sample for each class. Afterwards, we expand the set of selected samples by iteratively adding the most similar (for Similarity) or most dissimilar (for Diversity) sample to the average feature representation of the set of selected samples for each class (i.e., if we already have 5 selected samples in the specific class, we calculate the similarity to the representation of each of these 5 samples and take an average over these similarities as the final score). For in-context learning, we instead the most similar (for Similarity) or the most dissimilar (for Diversity) samples for each test sample as is common for this approach (Sun et al., 2023).

Finally, we also use the in-context learning specific LENS method (Li & Qiu, 2023) as a baseline. We use the same models (Mistral and Zephyr) for the selection of samples and run a hyperparameter search informed by the recommendations of the authors to identify the optimal setup for the search. In addition, we use the modified version of their source code for running the method.

For the core-set selection strategies, we use the modified implementation of the methods released by Guo et al. (2022) and its extension provided by Park et al. (2022) for active learning methods. For the Cartography strategy, we use the modified implementation released by the original authors Swayamdipta et al. (2020).

# E EXPERIMENTS SETUP: FURTHER DETAILS

The experiments and evaluation of different sample selection strategies are done on the within-domain few-shot learning, where the classes in training, validation and testing datasets all come from the same domain (e.g., from the same dataset). Each evaluation run is done on 600 randomly selected tasks for the gradient few-shot learning approaches and 300 tasks for the in-context learning approaches (due to high computation costs of LLMs). Each task is a 5-way 5-shot (or $K$-shot for experiments where we change the number of shots) classification task evaluated on 16 test samples per class, following the standard few-shot learning methodology and the evaluation from Ullah et al. (2022). The performance of a single evaluation run is reported as the mean accuracy across these tasks and the different splits of data into labelled/unlabelled and train-validation-test sets (we use 5 splits of data, where in each we split the available data into train-validation-test sets and select 200 samples per class for training, as mentioned in the main part of the paper). These scores are then used to report the findings in the main experiments, either as mean and standard deviation over multiple repeated runs of the specific sample selection strategy or over the results from different models and datasets to produce the aggregated figures. As such, we do not explicitly report the standard deviation caused by the different data splits or different sampled tasks. However, the observed deviation from these two factors is quite small (on average up to 0.1 in terms of accuracy), as it is calculated over large number of repeats. This is also the reason why we can observe zero standard deviation on some of the sample selection strategies.

For the gradient few-shot learning approaches we use the ResNet-18 (He et al., 2016) for image data, pretrained on Imagenet dataset (Deng et al., 2009), and a pretrained BERT-base (Devlin et al., 2019) model for the text data as base models. All the models are further trained on each dataset using the different few-shot learning approaches, namely the Prototypical Networks (Snell et al., 2017), Model-Agnostic Meta-Learning (MAML) (Finn et al., 2017) and Few-Shot Fine-Tuning (Chen et al., 2019). In case of Few-Shot Fine-Tuning, the further training is done on the concatenation of all the meta-training classes and their corresponding samples, while at the meta-test time, only the last layer is replaced and fine-tuned on the specific task with 5 classes. The Prototypical Networks and MAML follow the typical few-shot training setting, training on large number of 5-way few-shot tasks. The meta-learning approaches on image data are trained on 15 000 tasks and the Few-Shot Fine-Tuning approach is trained on 15 000 randomly sampled batches of size 16. On text data, the number of sampled tasks and batches is increased to 25 000. In addition, the MAML approach uses 5 training iteration and 10 testing iterations on image data and 10 training and 15 testing on text data, while Prototypical Networks use only 1 in both cases. Finally, all approaches use the Adam optimiser with 0.001 (meta) learning rate and 0.01 base learning rate. The full training process is repeated 3 times and the best performing model, selected using a separate validation dataset, is selected for use throughout the reported experiments.

For the in-context learning approaches, we use the instruction tuned Mistral-7B (Jiang et al., 2023) (instruct v0.1) and Zephyr-7B (Tunstall et al., 2023) (alpha version) large language models in the 4-bit quantised mode. This significantly reduces the computation costs required, while having low impact on the performance and almost no impact on the benefit of sample selection. We perform no training for these models (except in case of sample selection, which is covered in Appendix D). Instead we run a basic prompt engineering to find the best performing template for each model. In the end, we opted for the single template that performed the best on average for each model and each dataset, which is illustrated in Table 3, along with the verbaliser (i.e., the possible outputs of the model that are mapped to the classes in the dataset). Finally, the in-context learning models are set to provide deterministic outputs (controlling the temperature and disabling sampling) and set to generate maximum of 10 tokens. In case where multiple words that can be mapped to a dataset class are generated, the output is treated as incorrect as we consider the model to be just hallucinating and not working as intended (this behaviour was mainly observed when using the models in zero-shot or 50-shot setting, where the model either does not understand the task or the context is not sufficient for it to follow the instructions and often either repeats all the classes or generates further sentences).

All the hyperparameters for all models were determined using a hyperparameter search on each data modality and model using a single validation split. For the experiments, we use a modified implementation provided by the MetaAlbum (Ullah et al., 2022), extended to further approaches, models and datasets. In addition, we also use the hyperparameters used in the paper as a starting point for the hyperparameter search.

| Datasets | Prompt Format |
|---|---|
| News classification | Determine category of the sentence using following options: 1) *[Option 1]* 2) *[Option 2]* 3) *[Option 3]* 4) *[Option 4]* 5) *[Option 5]*. *[Input]* *[Output]* |
| Intent classification | Determine intent of the sentence using following options: 1) *[Option 1]* 2) *[Option 2]* 3) *[Option 3]* 4) *[Option 4]* 5) *[Option 5]*. *[Input]* *[Output]* |

| Dataset | Verbaliser |
|---|---|
| 20 News Group | {IBM, Middle East Politics, Windows XP, Motorcycles, Medicine, For Sale, Religion, MS Windows, Baseball, Auto, Hockey, Mac, Graphics, Christianity, Guns, Electronics, Space, Crypto, Atheism, Politics} |
| News Category | {Politics, World News, Parenting, Money, Wellness, Business, Weddings, Entertainment, Impact, Black Voices, Queer Voices, Crime, Divorce, Food and Drink, Worldpost, Parents, Travel, The Worldpost, Healthy Living, Taste, Media, Culture and Arts, Style, Weird News, Style and Beauty, Comedy, Home and Living, Sports, Environment, Education, Good News, Fifty, Women, Science, Arts and Culture, US News, Green, Tech, Religion, Latino Voices, College, Arts} |
| ATIS | {Flight, Abbreviation, City, Airfare, Ground Service, Ground Fare, Airline, Flight Number, Aircraft, Distance, Capacity, Flight Time, Quantity, Airport} |
| Facebook | {Road Condition, Departure, Duration, Event, Arrival, Directions, Distance, Traffic, Update, Combine, Route, Location} |
| HWU-64 | {Set Alarm, Definition, Play Music, Set Calendar, Play Radio, Confirm, Quirky, Send Email, Currency, Like, Social Query, News, Neutral, Stop, Calendar Query, Alarm Query, Recommend Movie, Play Audiobook, Weather Query, Praise, Social Post, Play Podcast, Negate, Recipe, Affirm, Remove Calendar, Recommend Location, Remove List, Explain, Email Query, Maths, Stock, Transport Query, Order Takeaway, Datetime Query, Repeat, Factoid, Taxi, Add List, Add Contact, Recommend Event, Mute Volume, List Query, Ticket, Convert Datetime, Joke, Remove Alarm, Traffic, Volume Up, Takeaway Query, Play Game, Contact Query, Music Query, Volume Other, Volume Down, Music Setting, Greet, Dislike} |
| SNIPS | {Playlist, Weather, Event, Musing, Creative Work, Rate Book, Book Restaurant} |

Table 3: Prompt formats and verbalisers used for different datasets in the paper. The News classification datasets include **20 News Group** and **News Category** datasets and Intent classification includes **ATIS**, **Facebook**, **HWU-64** and **SNIPS**. The *[Option 1-5]* are replaced with the names of the classes as defined by the verbaliser. The *[Input]* is replaced by the sentence of the samples and the *[Output]* is replaced with the name of class as defined by the verbaliser. The *[Input]* and *[Output]* are repeated for each in-context sample, while the final *[Output]* is used to determine the predicted class. The same format is used for both the Mistral-7B and Zephyr-7B large language models as it was the best performing one from the ones tried during basic prompt engineering.

# F    ADDITIONAL EXPERIMENTS: DATASET AND MODEL DEPENDENCE OF SAMPLE SELECTION STRATEGIES AND THEIR SENSITIVITY TO RANDOM COMPONENTS

In this appendix, we provide additional insights from ablation studies. We are mainly focus on: 1) how the impact of selection strategies is affected by the used models and datasets, i.e., look at the model and dataset dependence of different selection strategies and configurations of our proposed method (based on dataset specific results from Appendix L); and 2) how sensitive the strategies are to their random components, i.e., look at how the set of selected samples changes when the strategies are run multiple times and how this affects the overall performance (based on the variance from repeated strategy runs in Figure 5).

**Specific strategies show large sensitivity to the effects of random seed.** Changing the random seed for specific strategies often leads to different set of selected samples and different performance impact. The core set selection strategies are less affected by this randomness. Specific core set strategies

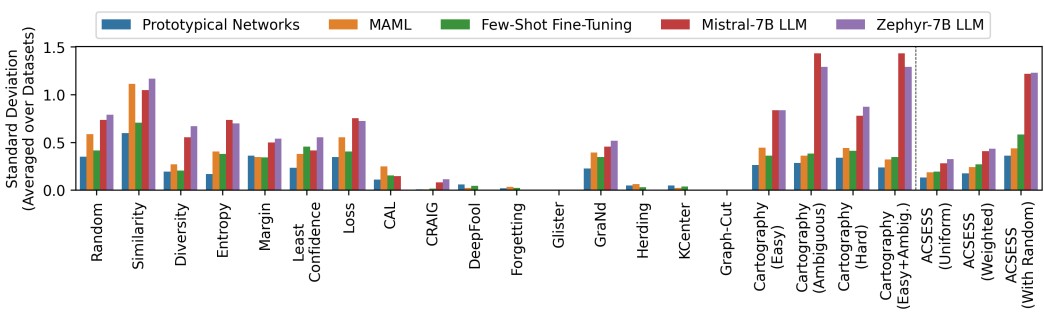

Figure 5: Standard deviation introduced by multiple runs of the different selection strategies. The results for Prototypical Networks, MAML and Few-Shot Fine-Tuning are aggregated over both the image and text datasets, while the results for Mistral and Zephyr are only from text datasets. The ACSESS method shows lower sensitivity to repeated runs compared to the majority of the strategies included in the combination, but still higher sensitivity than specific core set strategies that always lead to the same set of samples.

(such as *Glister* or *Graph-Cut*) consistently select the same set of samples and show no variance in results. Other strategies core set strategies (e.g., *Forgetting*, *DeepFool* or *Herding*) show only small variance. On the other hand, strategies with random initial condition, such as the similarity selection or some of the active learning strategies, are more sensitive to the effects of the random seed. **The ACSESS method shows lower sensitivity to repeated runs** (with the exception of weighting with random element), **but still may lead to different sets of samples based on the random seed** (as the standard deviation is higher than in core set strategies, i.e., up to 0.5 for the ACSESS method as opposed to 0 or 0.1 for majority of the core set strategies).

**Many selection strategies show strong dataset dependence.** In many cases, the strategies that perform well on one dataset (such as *DeepFool* on the HUM_ACT.ACT_410 dataset) may perform quite poorly on other datasets (such as MCR.PRT or MCR.PNU). Due to this dependence, such strategies cannot be used on any dataset without first evaluating them how well they perform on it. **Only few strategies perform consistently on the majority of the datasets**, such as *Herding* showing poor performance on all image datasets, or *Cartography* showing good performance across all datasets and models (while the most beneficial configuration of this strategy is dependent on data modality). Finally, we **do not observe a significant dataset dependence of the ACSESS method**, as it identifies the well-performing strategies and leverages their strengths for the specific dataset. This observation holds through for the weighting schemes as well (i.e., both *uniform* and *weighted* combinations show consistent performance across datasets).

**Many selection strategies show strong model dependence and sensitivity to selected samples.** The strategies performing well for one model (*GraNd* on MCR.PNU for MAML, or *KCenter* on 20 News for Zephyr) may result in a decrease of performance for other models (*GraNd* for Few-Shot Fine-Tuning, or *KCenter* for Mistral on the same setting). We do not observe such strong dependence for our proposed method ACSESS, with the exception for Few-Shot Fine-Tuning, where the uniform combination is always better than the weighted combination. Overall, having a different set of selected samples, either as a result from using different strategy or from repeated runs of the same strategy, **affects the Few-Shot Fine-Tuning less**, often leading to smaller benefit. At the same time, the **MAML** method and the **in-context learning** models (Mistral and Zephyr) are **more affected by the selected samples**, leading to more varied results and often higher performance increase. Such sensitivity to sample selection was observed in previous works as well (Pecher et al., 2023; Köksal et al., 2022; Zhang et al., 2022; Li & Qiu, 2023). Curiously, Prototypical Networks gain quite large performance increase from sample selection, but show low sensitivity to the selected samples and variance from repeated runs of sample selection. As such, **sample selection is more important for MAML and In-Context Learning and less important for Few-Shot Fine-Tuning.**

# G  ADDITIONAL EXPERIMENTS: COMPARISON OF THE ACSESS AND LENS METHODS

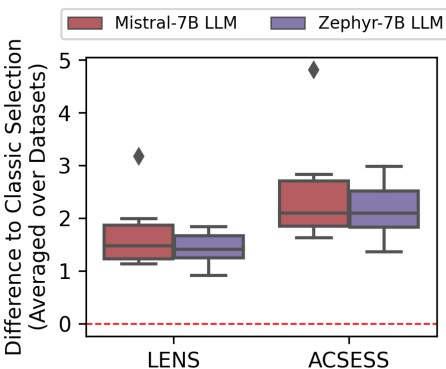

Figure 6: Comparison of the LENS method (Li & Qiu, 2023) to our proposed automatic combination of selection strategies (ACSESS).

In this appendix, we provide more information regarding the comparison of our proposed ACSESS method and the LENS method. As the LENS method is specifically designed for selecting samples for in-context, we perform this comparison only on text data using the Mistral-7B and Zephyr-7B models. These models are also used to select the samples in the LENS method, using the validation split of the dataset.

The LENS method represents a strong baseline for comparison for the in-context learning approach, as it is able to outperform all the remaining single-property selection strategies (only the *Cartography* sample selection strategy is able to achieve benefit comparable to this method). Based on the results in Figure 6, the ACSESS method is able to outperform the LENS method as well, achieving on average performance increase of $0.88$ percentage points for Mistral and $0.73$ percentage points for the Zephyr model. This increase is statistically more significant, with a p-value of $0.0002$ using the Wilcoxon signed-rank test.

# H  ADDITIONAL EXPERIMENTS: COMPARISON OF SAMPLE SELECTION IMPACT BETWEEN 5-SHOT AND 10-SHOT SETTING

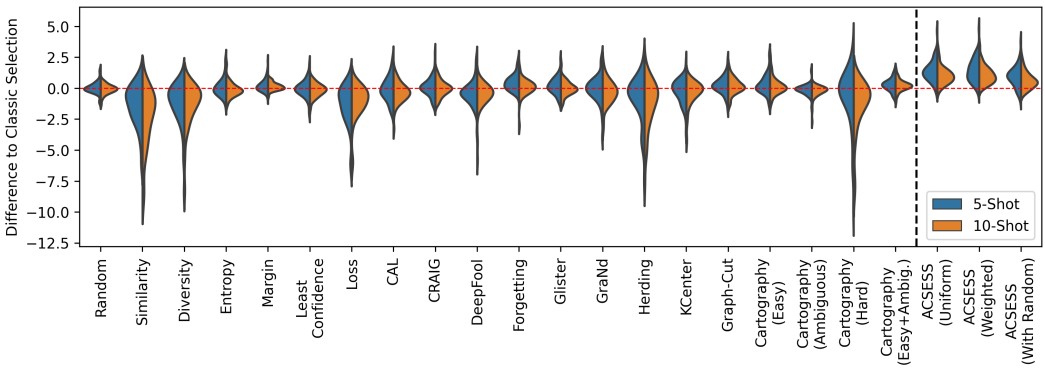

Figure 7: Distribution of the selection strategies benefit (calculated as difference to the classic few-shot selection) for the 5-shot and 10-shot setting. The benefit of the different selection strategies is more significant at lower number of shots.

To better explore the effect of increasing number of selected shots on the sample selection strategies, we repeat the evaluation of all the selection strategies in the 10-shot setting (using the same setup and methodology) and compare the results to 5-shot setting. For both 5-shot and 10-shot setting, we provide a distribution of the results as a difference to the *Classic* selection and illustrate the comparison of these distributions in Figure 7.

Overall, when moving from 5-shot to 10-shots, we observe an increase in the overall performance of the models for all selection strategies (with the exception of some bad performing ones) as well as the *Classic* selection. At the same time, the increase in the performance of the *Classic* selection is higher, leading to lower difference and lower impact of the different evaluated selection strategies. This finding holds for all strategies across all models and few-shot learning approaches.

In addition, we also compare the results using Wilcoxon signed-rank test, between the results of 5-shot and 10-shot setting. The resulting p-value ($1.93e - 07$) indicates that the different to Classic selection is more significant in case of 5-shot setting.

All in all, these experiments further reinforce the findings from comparison of ACSESS and *Random* selection, where we observe that the the sample selection progressively leads to lower impact as we increase the number of shots we select.

## I ADDITIONAL EXPERIMENTS: IMPACT OF INCREASING NUMBER OF SHOTS FOR INDIVIDUAL DATASETS

In this appendix, we provide results from investigating the impact of increasing the number of shots on the benefit of sample selection separately for the individual dataset. For each dataset, we increase the number of selected shots, starting from 1-shot up to the full dataset. The datasets HUM_ACT.ACT_410 and MCR.PRT contain only up to 50 samples per class when all samples are used and the MNF.TEX_DTD only up to 100. All the remaining datasets either contain at maximum 200 samples, or are subsampled to 200 samples per class if necessary (this is especially done for the text datasets, which are often significantly larger). For in-context learning, we increase the number of shots only up to 50 samples per class, as increasing the number further resulted in significantly lower performance (mainly due to limited context size of the large language models). The results for the image datasets are included in Figure 8, for the text datasets using the gradient few-shot learning in Figure 9 and for text datasets using the in-context learning in Figure 10.

For the majority of the datasets, we observe that the highest performance of the sample selection using the ACSESS method is achieved on lower number of shots and then either remains the same until the full dataset, or gradually decreases as we use more samples. Only for specific datasets (LR_AM.DOG and LR_AM.AWA datasets for all approaches; PLT.FLW and SNIPS for MAML; and MNF.TEX_DTD for Prototypical Networks and MAML) we observe continual increase of performance when using sample selection as we increase the number of shots, with the highest performance being achieved only at the full dataset (or slightly beforehand). In addition, on specific datasets, not all models behave the same, for example, on the MNF.TEX_DTD dataset, only the Few-Shot Fine-Tuning approach achieves highest performance on lower number of shots (which then monotonically decrease as we increase the number of used samples), while the MAML and Prototypical Networks meta-learning approaches achieve the highest performance only when using almost the full dataset.

On the other hand, using the Random selection, we often observe monotonic increase in performance as the number of the selected shots increases, with the highest performance being achieved on the full dataset. This behaviour can be observed across all datasets and few-shot learning approaches (with exception of in-context learning). As such, this further reinforces our findings and main motivation that the data-centric approach can be beneficial for the few-shot learning. Curating a subset of the most informative and high-quality samples can often lead to performance on par or better than the one when using all the samples for training.

Finally, we also observe that the benefit of sample selection is more significant at lower number of shots. The difference in performance between the ACSESS method and the Random selection on the lower number of shots is significantly higher and slowly decreases as we use more shots for training. The exception is MAML on the PLT.FLW and SNIPS datasets, or Few-Shot Fine-Tuning on PLT.FLW and MCR.PNU datasets, where the performance of the sample selection is similar to the

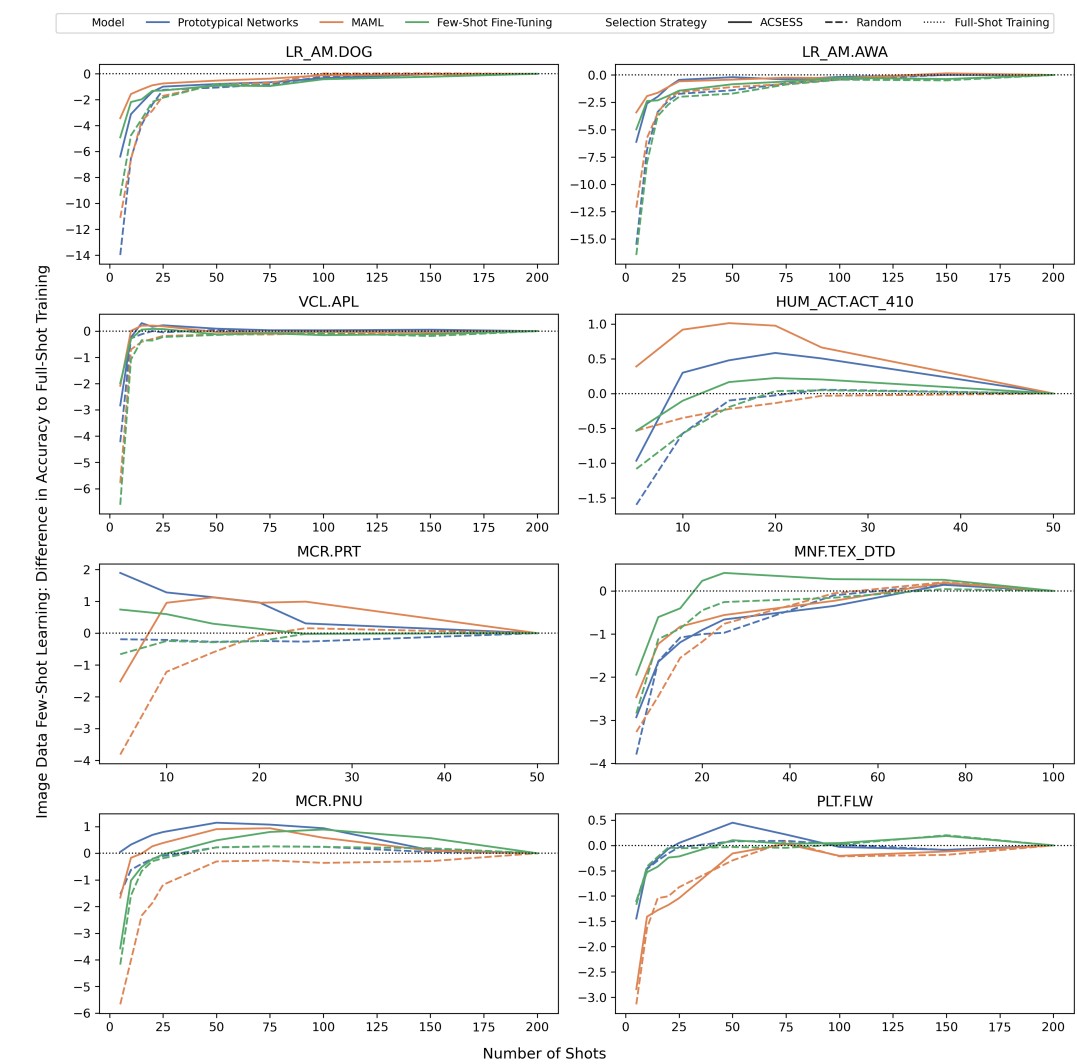

Figure 8: The different benefit of the selection strategies based on the number of selected shots for the image datasets using gradient few-shot learning. The reported benefit is calculated as a difference in performance to the one when using the full dataset, for both the ACSESS method and the Random selection.

one using random selection even when using small number of shots. After a certain point, the sample selection using the ACSESS method degrades to Random selection. As such, the focus on selecting high quality samples is paramount when selecting only few labelled samples per class, while random selection can be safely used when using higher number of shots.

# J ADDITIONAL EXPERIMENTS: SELECTION FROM DATASETS OF DIFFERENT SIZES FOR INDIVIDUAL DATASETS

In this appendix, we provide the results from investigating the impact of performing sample selection from datasets of different sizes for the individual datasets. Each dataset is subsampled into multiple sizes, starting with 5 samples per class up to the full dataset. Each subsampling is repeated 10 times and an average result over these repeats are reported. The full-dataset setting represents the selection from 200 samples per class for each datasets (for datasets with a lower number of samples per class, such as HUM_ACT.ACT_410, MCR.PRT or MNF.TEX_DTD, we observe a constant line after the

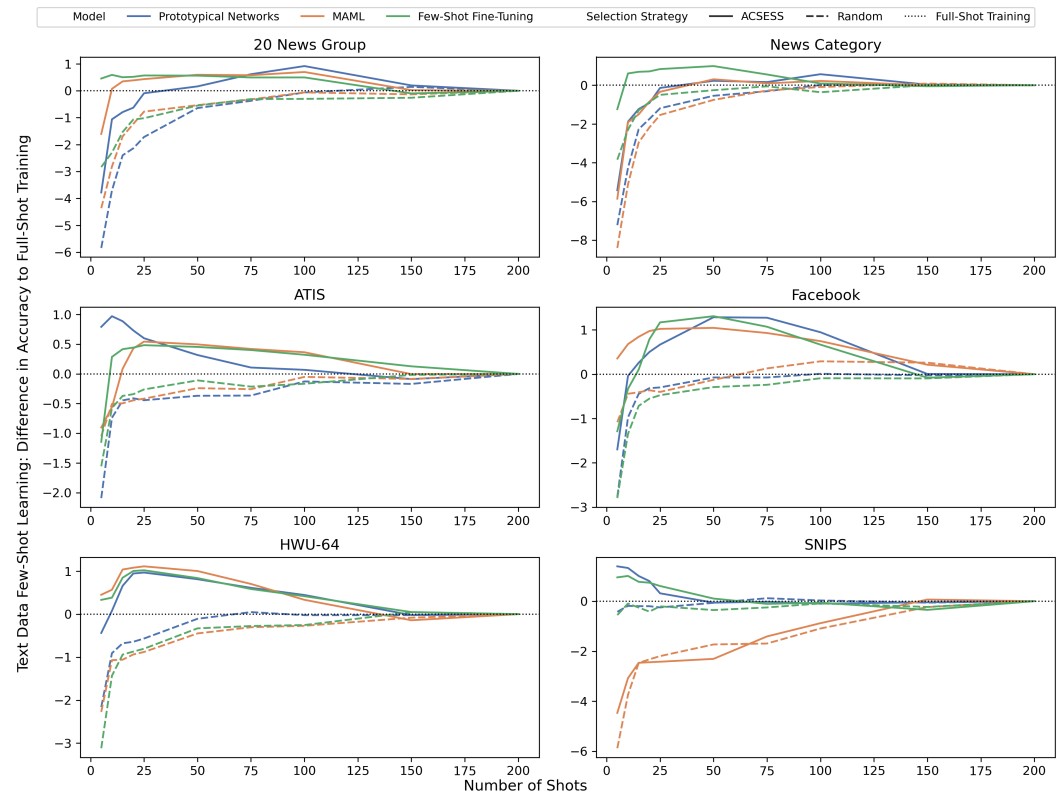

Figure 9: The different benefit of the selection strategies based on the number of selected shots for the text datasets using gradient few-shot learning. The reported benefit is calculated as a difference in performance to the one when using the full dataset, for both the ACSESS method and the Random selection.

full dataset was achieved). In all cases we perform selection of 5 shots per class using the ACSESS method and Random selection. The results for the image datasets are included in Figure 11, for the text datasets using the gradient few-shot learning in Figure 12 and for text datasets using the in-context learning in Figure 13.

For majority of the combinations of datasets and approaches, we can observe that sample selection can achieve similar performance using only a fraction of the dataset. For the gradient few-shot learning, this performance is achieved using only 10% of the dataset (20 sample per class), while for in-context learning, this performance is achieved using 25% of the dataset (50 samples per class). However, we observe that this strongly depends on the dataset and the model/approach used. For example on the ATIS dataset and in-context learning, using only 10 samples per class is enough to achieve the same performance as when selecting from the full dataset, while on the SNIPS dataset the number of samples per class increase up to 100. Similarly, on the ATIS dataset with gradient few-shot learning, the MAML a Few-Shot Fine-Tuning approaches achieve the similar performance using only 15-20 samples per class, while Prototypical Networks require the full dataset. Curiously, in specific cases, we can observe that running sample selection on subset of the dataset can lead to better performance than when performing selection from the full dataset (such as on the News Category dataset and gradient few-shot learning).

At the same time, we observe no change in impact of random selection from datasets of different sizes. In this case, the performance remains more or less constant (and often significantly below selection using the ACSESS method) throughout all the sizes – only difference in performance stems from the random subsampling. Finally, using only 10 samples per class, we can observer that sample selection using the ACSESS method can lead to quite big performance improvement over the random selection. Therefore we can conclude that the sample selection is beneficial even when using a small

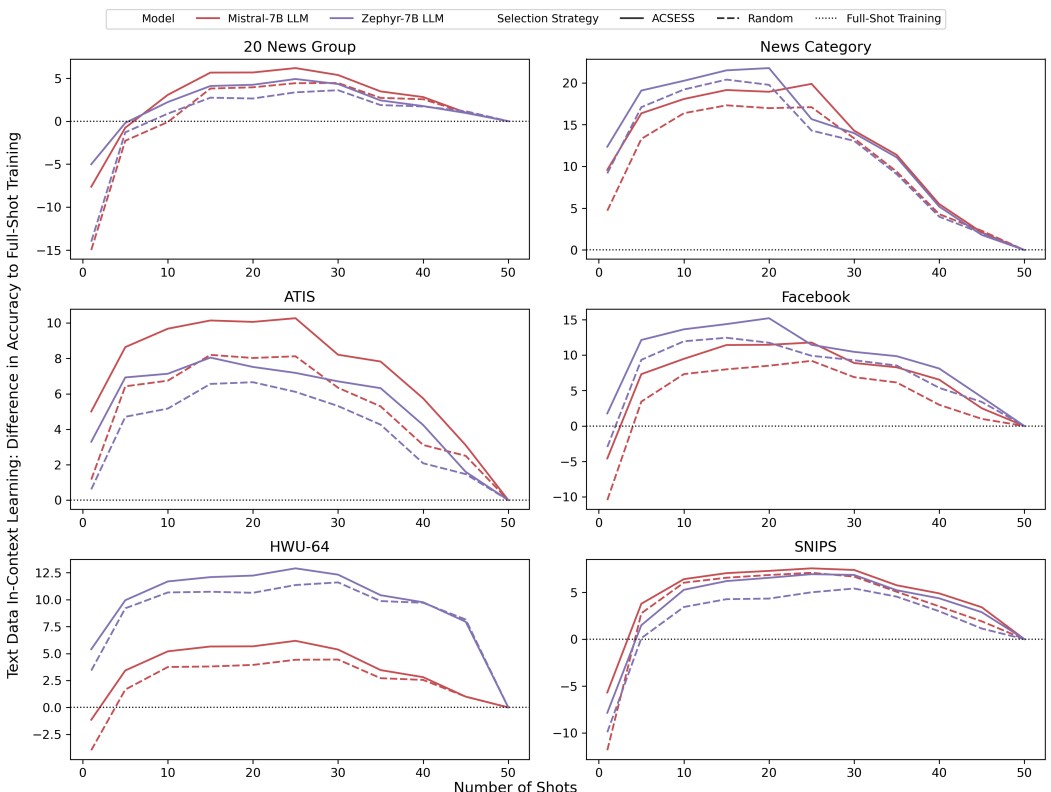

Figure 10: The different benefit of the selection strategies based on the number of selected shots for the text datasets using in-context learning. The reported benefit is calculated as a difference in performance to the one when using the full dataset, for both the ACSESS method and the Random selection.

fraction of the dataset size as the one used in the main experiments, although it will lead to a lowered benefit of the sample selection and its impact on the overall performance (which is still higher than when using random selection).

## K  FEW-SHOT LEARNING PROBLEM FORMULATION

In this section, we introduce the definition of the few-shot learning problem and the difference between gradient-based few-shot learning and few-shot learning via in-context learning. Note that in-context learning can be applied only to text datasets as it utilises pretrained large language models.

For the standard few-shot learning, we assume a larger collection of instances $D = (x_i, y_i)$, with larger number of classes. From this dataset, a separate task $D_T \subset D$ is created by randomly sampling few classes $N$. The task is characterised by a training set (also called support set, $S$) and test set (also called query set, $Q$). To create these sets, few samples ($k$) are sampled for each class. The few-shot learning algorithms then use the few samples from the support set $S_T$ to learn to solve the task $D_T$ quickly and are evaluated on this ability using the query set $Q_T$. This represents the $N$-way $k$-shot classification. The few-shot learning approaches mostly consist of two phases, training phase and testing phase. Often, these two phases work with separate sets of $D_T$, i.e., the classes appearing as part of tasks in the testing phase do not appear during training. As each task works only with subset of classes, the evaluation is often done on many such tasks and the average performance from these tasks is reported.

The gradient-based few-shot learning approaches follow episodic training similar to typical supervised training to gather knowledge and experience and learn to quickly adapt when encountering new task.

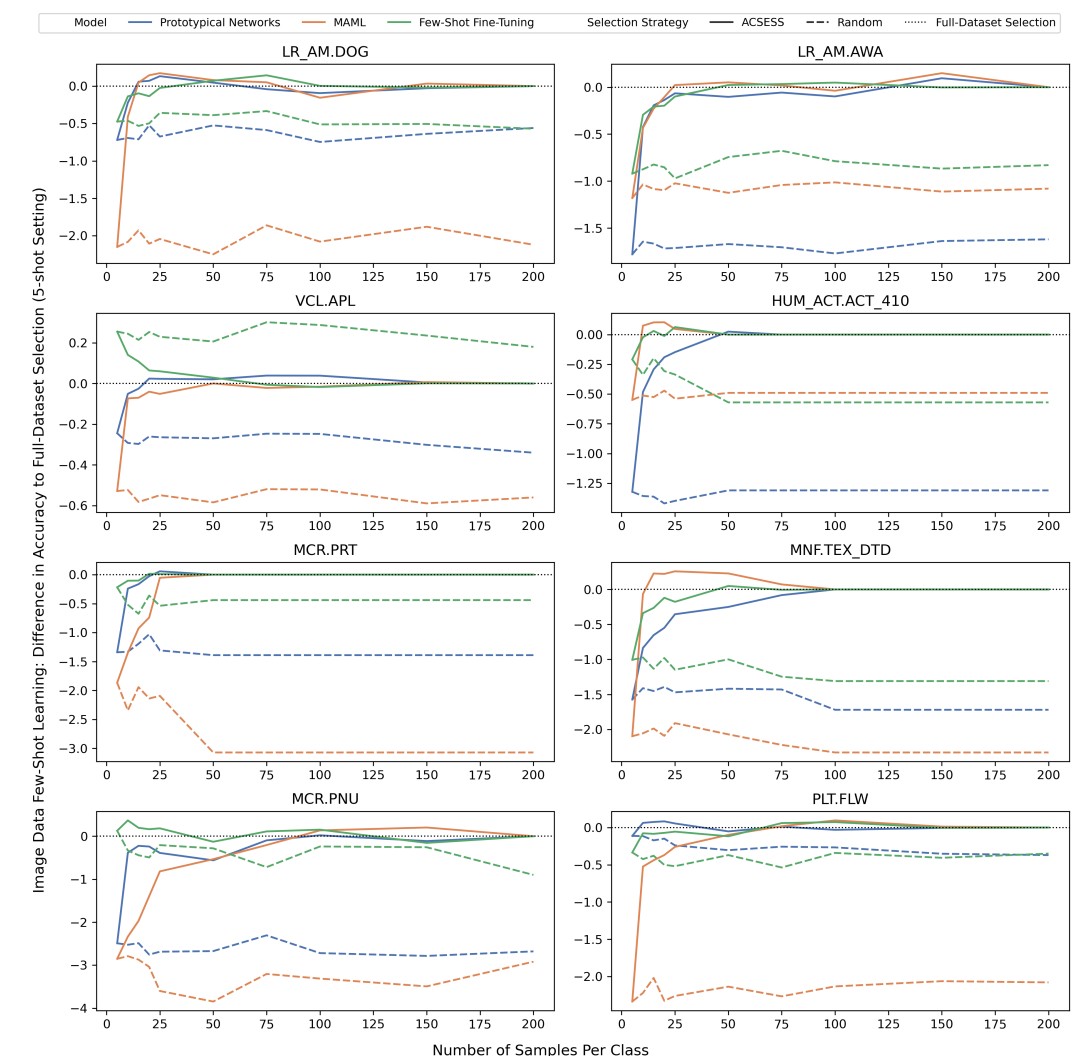

Figure 11: The difference in performance of sample selection strategies on dataset subsets of different sizes for the image datasets using gradient few-shot learning. The reported difference is in comparison to using sample selection from the full dataset.

Using the support set, the model is adapted to the task using gradient descent based approaches and then evaluated on the query set. Based on the evaluation, some kind of knowledge on how to better adapt is saved in part of the model, while the rest of the model is reset. This is done for a large number of episodes (often called meta-training), with the model accumulating the knowledge. After enough episodes are run, the model is evaluated (in a meta-testing phase) using the exact same setup.

On the other hand, no training phase is present in few-shot learning via in-context learning, but a pretrained large language model is used without any additional parameter updates instead. For each new task, a task-specific text prompt is formed, by concatenating task-specific instructions, the samples from the support set along with the their labels, and a specific test sample. After being given this prompt, the large language model generate a prediction in a form of word (or multiple words). These generated words are then mapped to the task labels and a class is assigned. The process is repeated for each sample in the query set. For example in case of binary classification, the task-specific instruction may be "Determine sentiment of following sentence", the concatenated samples from support set may be formatted as "sentence, sentiment: positive/negative", with the word "positive" mapping to class representing positive sentiment and "negative" to class representing

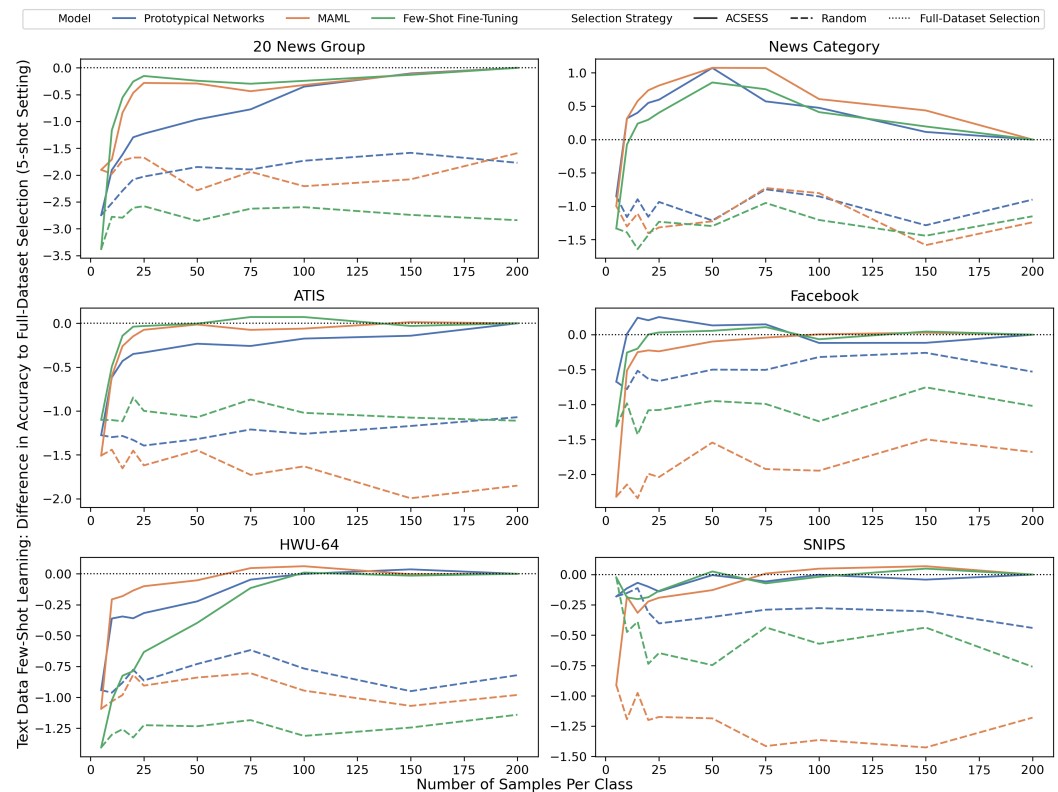

Figure 12: The difference in performance of sample selection strategies on dataset subsets of different sizes for the text datasets using gradient few-shot learning. The reported difference is in comparison to using sample selection from the full dataset.

negative sentiment. As such, the only supervision for the large language model is from the instruction and the concatenation of support samples. Therefore, the model is limited by its max input length. At the same time, this approach was found to be sensitive to the selected samples in the support set and their order (Zhao et al., 2021; Zhang et al., 2022; Pecher et al., 2023; Chang & Jia, 2023).

## L   DATASET SPECIFIC RESULTS AND VISUALISATIONS

In this appendix, we present the full results for each combination of dataset and model we use in the experiments. The results are presented using both figures, showing the difference to Classic selection, as well as tables, showing the overall performance of the model on the specific dataset, the standard deviation from multiple runs of the selection strategy and the difference to the Classic selection (while also showing the results for Classic selection). Following is the mapping for different datasets, each presenting results for all the models:

- **LR_AM.DOG** dataset in Figure 14 and Table 4
- **LR_AM.AWA** dataset in Figure 15 and Table 5
- **VCL.APL** dataset in Figure 16 and Table 6
- **HUM_ACT.ACT_410** dataset in Figure 17 and Table 7
- **MNF.TEX_DTD** dataset in Figure 18 and Table 8
- **MCR.PRT** dataset in Figure 19 and Table 9
- **MCR.PNU** dataset in Figure 20 and Table 10
- **PLT.FLW** dataset in Figure 21 and Table 11

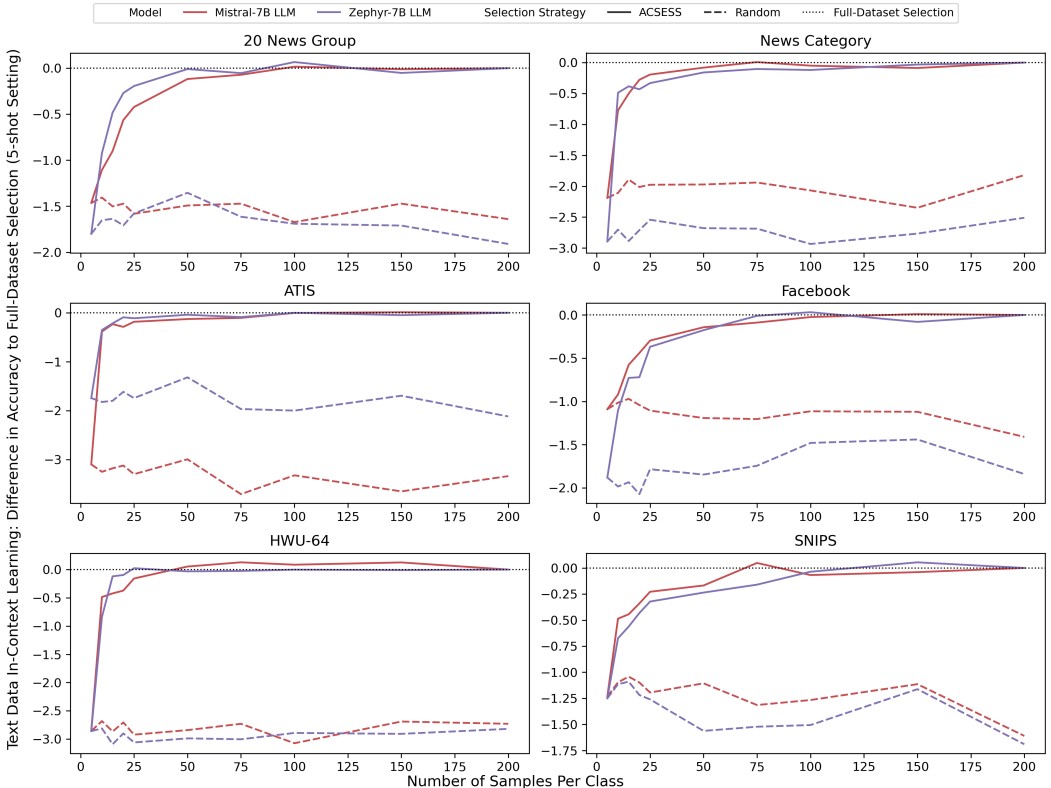

Figure 13: The difference in performance of sample selection strategies on dataset subsets of different sizes for the text datasets using in-context learning. The reported difference is in comparison to using sample selection from the full dataset.

- **20 News Group** dataset in Figure 22 and Table 12
- **News Category** dataset in Figure 23 and Table 13
- **ATIS** dataset in Figure 24 and Table 14
- **Facebook** dataset in Figure 25 and Table 15
- **HWU-64** dataset in Figure 26 and Table 16
- **SNIPS** dataset in Figure 27 and Table 17

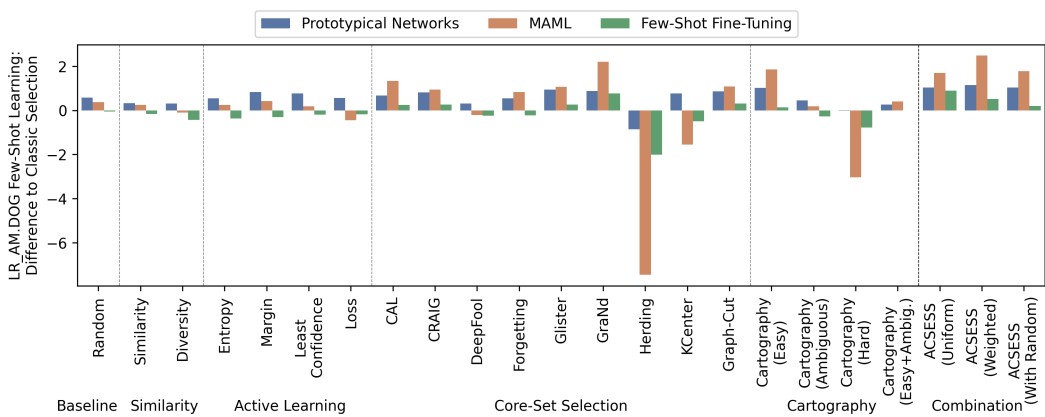

Figure 14: Benefit of the different selection strategies calculated as the difference to the classic few-shot selection strategy for the LR_AM.DOG dataset. The performance of the classic selection is represented as the zero value.

Table 4: Benefit of the different selection strategies calculated as the difference in accuracy to the classic few-shot selection strategy for the LR_AM.DOG dataset. The subscript represents the standard deviation in the accuracy from running the selection strategy multiple times. The difference to the classic selection baseline (first row) is also included.

| LR_AM.DOG | PROTONET | | MAML | | FINE-TUNING | |
|---|---|---|---|---|---|---|
| | Acc. + Std | Diff. | Acc. + Std | Diff. | Acc. + Std | Diff. |
| Classic | $70.01_{0.30}$ | $+0.00$ | $57.11_{0.31}$ | $+0.00$ | $65.36_{0.32}$ | $+0.00$ |
| Random | $70.59_{0.21}$ | $+0.58$ | $57.48_{0.21}$ | $+0.37$ | $65.30_{0.18}$ | $-0.05$ |
| Similarity | $70.33_{0.20}$ | $+0.32$ | $57.36_{0.35}$ | $+0.25$ | $65.19_{0.25}$ | $-0.16$ |
| Diversity | $70.31_{0.12}$ | $+0.30$ | $57.00_{0.15}$ | $-0.11$ | $64.92_{0.09}$ | $-0.43$ |
| Entropy | $70.56_{0.11}$ | $+0.55$ | $57.36_{0.09}$ | $+0.25$ | $64.98_{0.23}$ | $-0.38$ |
| Margin | $70.84_{0.31}$ | $+0.84$ | $57.53_{0.38}$ | $+0.42$ | $65.04_{0.33}$ | $-0.31$ |
| Least Confidence | $70.77_{0.14}$ | $+0.76$ | $57.29_{0.04}$ | $+0.18$ | $65.15_{0.12}$ | $-0.20$ |
| Loss | $70.58_{0.20}$ | $+0.57$ | $56.67_{0.13}$ | $-0.44$ | $65.17_{0.20}$ | $-0.18$ |
| CAL | $70.68_{0.04}$ | $+0.67$ | $58.45_{0.17}$ | $+1.33$ | $65.60_{0.09}$ | $+0.25$ |
| CRAIG | $70.82_{0.00}$ | $+0.81$ | $58.05_{0.00}$ | $+0.94$ | $65.62_{0.00}$ | $+0.26$ |
| DeepFool | $70.32_{0.00}$ | $+0.31$ | $56.90_{0.00}$ | $-0.21$ | $65.10_{0.00}$ | $-0.25$ |
| Forgetting | $70.55_{0.00}$ | $+0.54$ | $57.93_{0.00}$ | $+0.82$ | $65.12_{0.00}$ | $-0.23$ |
| Glister | $70.96_{0.00}$ | $+0.95$ | $58.18_{0.00}$ | $+1.07$ | $65.62_{0.00}$ | $+0.27$ |
| GraNd | $70.89_{0.03}$ | $+0.88$ | $59.31_{0.08}$ | $+2.20$ | $66.12_{0.20}$ | $+0.77$ |
| Herding | $69.15_{0.00}$ | $-0.86$ | $49.66_{0.00}$ | $-7.45$ | $63.34_{0.00}$ | $-2.01$ |
| KCenter | $70.77_{0.00}$ | $+0.76$ | $55.56_{0.00}$ | $-1.55$ | $64.86_{0.00}$ | $-0.50$ |
| Graph-Cut | $70.87_{0.00}$ | $+0.86$ | $58.20_{0.00}$ | $+1.09$ | $65.66_{0.00}$ | $+0.31$ |
| Cartography$_{Easy}$ | $71.04_{0.14}$ | $+1.02$ | $58.96_{0.20}$ | $+1.85$ | $65.48_{0.15}$ | $+0.13$ |
| Cartography$_{Ambiguous}$ | $70.46_{0.28}$ | $+0.46$ | $57.30_{0.32}$ | $+0.19$ | $65.08_{0.27}$ | $-0.28$ |
| Cartography$_{Hard}$ | $70.02_{0.16}$ | $+0.01$ | $54.08_{0.28}$ | $-3.04$ | $64.58_{0.19}$ | $-0.77$ |
| Cartography$_{Easy+Ambig.}$ | $70.28_{0.28}$ | $+0.27$ | $57.52_{0.32}$ | $+0.41$ | $65.34_{0.34}$ | $-0.01$ |
| ACSESS$_{Uniform}$ | $71.04_{0.18}$ | $+1.03$ | $58.81_{0.11}$ | $+1.70$ | $66.25_{0.14}$ | $+0.90$ |
| ACSESS$_{Weighted}$ | $71.15_{0.09}$ | $+1.14$ | $59.60_{0.11}$ | $+2.48$ | $65.87_{0.05}$ | $+0.51$ |
| ACSESS$_{With Random}$ | $71.05_{0.16}$ | $+1.04$ | $58.89_{0.26}$ | $+1.78$ | $65.55_{0.16}$ | $+0.19$ |

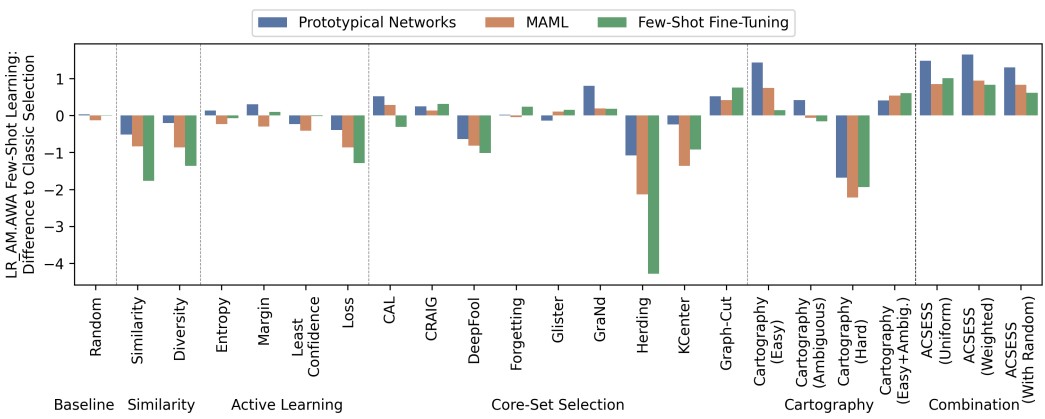

Figure 15: Benefit of the different selection strategies calculated as the difference to the classic few-shot selection strategy for the LR_AM.AWA dataset. The performance of the classic selection is represented as the zero value.

Table 5: Benefit of the different selection strategies calculated as the difference in accuracy to the classic few-shot selection strategy for the LR_AM.AWA dataset. The subscript represents the standard deviation in the accuracy from running the selection strategy multiple times. The difference to the classic selection baseline (first row) is also included.

| LR_AM.AWA | PROTONET | | MAML | | FINE-TUNING | |
|---|---|---|---|---|---|---|
| | Acc. + Std | Diff. | Acc. + Std | Diff. | Acc. + Std | Diff. |
| Classic | $76.32_{0.30}$ | $+0.00$ | $71.32_{0.27}$ | $+0.00$ | $79.16_{0.40}$ | $+0.00$ |
| Random | $76.35_{0.25}$ | $+0.03$ | $71.19_{0.16}$ | $-0.13$ | $79.17_{0.08}$ | $+0.01$ |
| Similarity | $75.80_{0.21}$ | $-0.52$ | $70.49_{0.23}$ | $-0.84$ | $77.40_{0.34}$ | $-1.77$ |
| Diversity | $76.11_{0.16}$ | $-0.21$ | $70.46_{0.13}$ | $-0.87$ | $77.80_{0.14}$ | $-1.37$ |
| Entropy | $76.46_{0.27}$ | $+0.14$ | $71.10_{0.08}$ | $-0.23$ | $79.09_{0.23}$ | $-0.07$ |
| Margin | $76.63_{0.16}$ | $+0.31$ | $71.03_{0.22}$ | $-0.30$ | $79.26_{0.20}$ | $+0.09$ |
| Least Confidence | $76.08_{0.27}$ | $-0.24$ | $70.91_{0.21}$ | $-0.42$ | $79.15_{0.10}$ | $-0.02$ |
| Loss | $75.93_{0.14}$ | $-0.39$ | $70.46_{0.24}$ | $-0.86$ | $77.88_{0.00}$ | $-1.28$ |
| CAL | $76.84_{0.25}$ | $+0.52$ | $71.61_{0.29}$ | $+0.28$ | $78.85_{0.29}$ | $-0.31$ |
| CRAIG | $76.56_{0.00}$ | $+0.24$ | $71.46_{0.00}$ | $+0.13$ | $79.48_{0.00}$ | $+0.32$ |
| DeepFool | $75.68_{0.00}$ | $-0.64$ | $70.51_{0.01}$ | $-0.82$ | $78.15_{0.01}$ | $-1.02$ |
| Forgetting | $76.34_{0.00}$ | $+0.02$ | $71.28_{0.00}$ | $-0.04$ | $79.40_{0.00}$ | $+0.24$ |
| Glister | $76.18_{0.00}$ | $-0.14$ | $71.43_{0.00}$ | $+0.10$ | $79.32_{0.00}$ | $+0.16$ |
| GraNd | $77.13_{0.20}$ | $+0.81$ | $71.52_{0.05}$ | $+0.19$ | $79.34_{0.41}$ | $+0.18$ |
| Herding | $75.24_{0.00}$ | $-1.08$ | $69.19_{0.00}$ | $-2.13$ | $74.88_{0.00}$ | $-4.28$ |
| KCenter | $76.07_{0.00}$ | $-0.25$ | $69.96_{0.00}$ | $-1.36$ | $78.24_{0.00}$ | $-0.92$ |
| Graph-Cut | $76.84_{0.00}$ | $+0.52$ | $71.74_{0.00}$ | $+0.42$ | $79.92_{0.00}$ | $+0.76$ |
| Cartography$_{Easy}$ | $77.75_{0.15}$ | $+1.43$ | $72.07_{0.12}$ | $+0.75$ | $79.31_{0.23}$ | $+0.14$ |
| Cartography$_{Ambiguous}$ | $76.74_{0.43}$ | $+0.42$ | $71.26_{0.32}$ | $-0.07$ | $79.00_{0.49}$ | $-0.16$ |
| Cartography$_{Hard}$ | $74.64_{0.19}$ | $-1.68$ | $69.11_{0.16}$ | $-2.22$ | $77.22_{0.30}$ | $-1.94$ |
| Cartography$_{Easy+Ambig.}$ | $76.73_{0.28}$ | $+0.41$ | $71.86_{0.30}$ | $+0.54$ | $79.76_{0.34}$ | $+0.60$ |
| ACSESS$_{Uniform}$ | $77.80_{0.05}$ | $+1.48$ | $72.17_{0.14}$ | $+0.85$ | $80.17_{0.15}$ | $+1.00$ |
| ACSESS$_{Weighted}$ | $77.97_{0.09}$ | $+1.65$ | $72.27_{0.07}$ | $+0.95$ | $80.00_{0.10}$ | $+0.83$ |
| ACSESS$_{With Random}$ | $77.62_{0.15}$ | $+1.30$ | $72.16_{0.18}$ | $+0.84$ | $79.78_{0.15}$ | $+0.61$ |

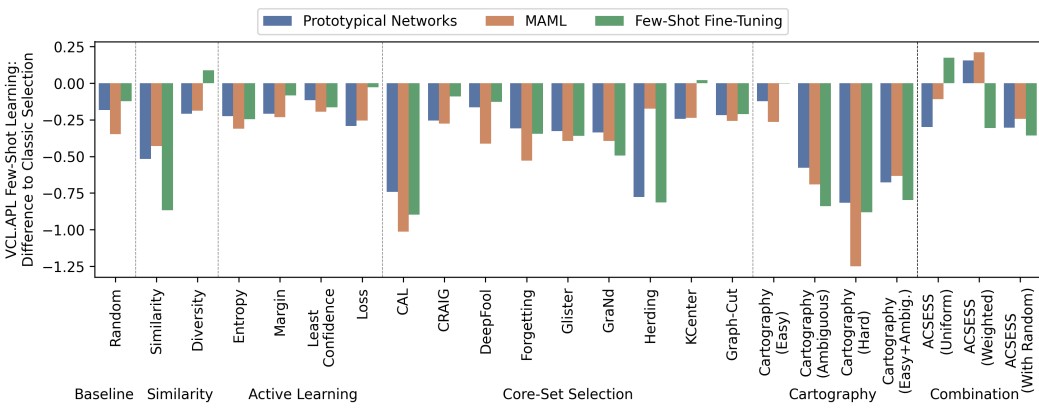

Figure 16: Benefit of the different selection strategies calculated as the difference to the classic few-shot selection strategy for the VCL.APL dataset. The performance of the classic selection is represented as the zero value.

Table 6: Benefit of the different selection strategies calculated as the difference in accuracy to the classic few-shot selection strategy for the VCL.APL dataset. The subscript represents the standard deviation in the accuracy from running the selection strategy multiple times. The difference to the classic selection baseline (first row) is also included.

| VCL.APL | PROTONET | | MAML | | FINE-TUNING | |
|---|---|---|---|---|---|---|
| | Acc. + Std | Diff. | Acc. + Std | Diff. | Acc. + Std | Diff. |
| Classic | $98.03_{0.09}$ | $+0.00$ | $96.95_{0.09}$ | $+0.00$ | $97.26_{0.09}$ | $+0.00$ |
| Random | $97.84_{0.08}$ | $-0.18$ | $96.60_{0.07}$ | $-0.35$ | $97.14_{0.09}$ | $-0.12$ |
| Similarity | $97.51_{0.20}$ | $-0.52$ | $96.52_{0.28}$ | $-0.43$ | $96.40_{0.54}$ | $-0.87$ |
| Diversity | $97.82_{0.03}$ | $-0.21$ | $96.76_{0.03}$ | $-0.19$ | $97.35_{0.05}$ | $+0.09$ |
| Entropy | $97.80_{0.03}$ | $-0.22$ | $96.64_{0.05}$ | $-0.31$ | $97.02_{0.05}$ | $-0.25$ |
| Margin | $97.82_{0.09}$ | $-0.21$ | $96.72_{0.05}$ | $-0.23$ | $97.18_{0.11}$ | $-0.08$ |
| Least Confidence | $97.91_{0.08}$ | $-0.12$ | $96.76_{0.05}$ | $-0.19$ | $97.10_{0.13}$ | $-0.16$ |
| Loss | $97.74_{0.02}$ | $-0.29$ | $96.69_{0.02}$ | $-0.26$ | $97.24_{0.12}$ | $-0.03$ |
| CAL | $97.28_{0.04}$ | $-0.74$ | $95.94_{0.07}$ | $-1.01$ | $96.37_{0.25}$ | $-0.90$ |
| CRAIG | $97.77_{0.00}$ | $-0.25$ | $96.68_{0.00}$ | $-0.28$ | $97.18_{0.00}$ | $-0.09$ |
| DeepFool | $97.86_{0.00}$ | $-0.16$ | $96.54_{0.00}$ | $-0.41$ | $97.14_{0.01}$ | $-0.13$ |
| Forgetting | $97.72_{0.00}$ | $-0.31$ | $96.42_{0.00}$ | $-0.53$ | $96.92_{0.00}$ | $-0.35$ |
| Glister | $97.70_{0.00}$ | $-0.33$ | $96.56_{0.00}$ | $-0.39$ | $96.91_{0.00}$ | $-0.36$ |
| GraNd | $97.69_{0.13}$ | $-0.34$ | $96.56_{0.32}$ | $-0.39$ | $96.77_{0.24}$ | $-0.49$ |
| Herding | $97.25_{0.00}$ | $-0.78$ | $96.78_{0.00}$ | $-0.17$ | $96.45_{0.00}$ | $-0.81$ |
| KCenter | $97.78_{0.00}$ | $-0.24$ | $96.71_{0.00}$ | $-0.24$ | $97.28_{0.00}$ | $+0.02$ |
| Graph-Cut | $97.81_{0.00}$ | $-0.22$ | $96.69_{0.00}$ | $-0.26$ | $97.05_{0.00}$ | $-0.21$ |
| Cartography$_{\text{Easy}}$ | $97.90_{0.12}$ | $-0.12$ | $96.68_{0.04}$ | $-0.26$ | $97.26_{0.24}$ | $-0.00$ |
| Cartography$_{\text{Ambiguous}}$ | $97.45_{0.18}$ | $-0.58$ | $96.26_{0.11}$ | $-0.69$ | $96.42_{0.42}$ | $-0.84$ |
| Cartography$_{\text{Hard}}$ | $97.21_{0.19}$ | $-0.82$ | $95.70_{0.08}$ | $-1.25$ | $96.38_{0.34}$ | $-0.88$ |
| Cartography$_{\text{Easy+Ambig.}}$ | $97.35_{0.21}$ | $-0.68$ | $96.32_{0.12}$ | $-0.63$ | $96.47_{0.32}$ | $-0.80$ |
| ACSESS$_{\text{Uniform}}$ | $97.73_{0.05}$ | $-0.30$ | $96.84_{0.09}$ | $-0.11$ | $97.44_{0.05}$ | $+0.17$ |
| ACSESS$_{\text{Weighted}}$ | $98.18_{0.06}$ | $+0.16$ | $97.16_{0.03}$ | $+0.21$ | $96.96_{0.18}$ | $-0.30$ |
| ACSESS$_{\text{With Random}}$ | $97.72_{0.08}$ | $-0.30$ | $96.71_{0.02}$ | $-0.24$ | $96.91_{0.16}$ | $-0.36$ |

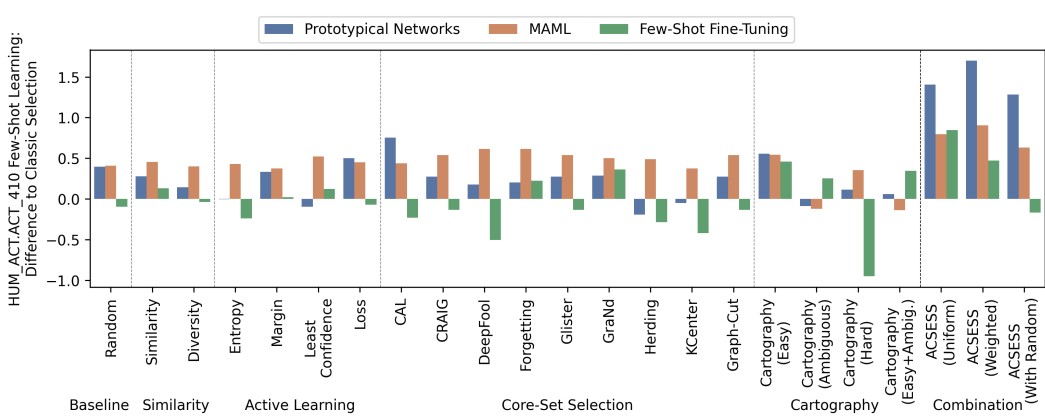

Figure 17: Benefit of the different selection strategies calculated as the difference to the classic few-shot selection strategy for the HUM_ACT.ACT_410 dataset. The performance of the classic selection is represented as the zero value.

Table 7: Benefit of the different selection strategies calculated as the difference in accuracy to the classic few-shot selection strategy for the HUM_ACT.ACT_410 dataset. The subscript represents the standard deviation in the accuracy from running the selection strategy multiple times. The difference to the classic selection baseline (first row) is also included.

| HUM_ACT.ACT_410 | PROTONET | | MAML | | FINE-TUNING | |
| --- | --- | --- | --- | --- | --- | --- |
| | Acc. + Std | Diff. | Acc. + Std | Diff. | Acc. + Std | Diff. |
| Classic | $71.38_{0.35}$ | $+0.00$ | $62.65_{0.38}$ | $+0.00$ | $69.43_{0.53}$ | $+0.00$ |
| Random | $71.77_{0.34}$ | $+0.40$ | $63.06_{0.09}$ | $+0.41$ | $69.33_{0.50}$ | $-0.10$ |
| Similarity | $71.66_{0.17}$ | $+0.28$ | $63.10_{0.08}$ | $+0.45$ | $69.56_{0.57}$ | $+0.13$ |
| Diversity | $71.52_{0.20}$ | $+0.14$ | $63.05_{0.08}$ | $+0.40$ | $69.39_{0.18}$ | $-0.04$ |
| Entropy | $71.37_{0.32}$ | $-0.00$ | $63.08_{0.07}$ | $+0.43$ | $69.19_{0.18}$ | $-0.24$ |
| Margin | $71.71_{0.29}$ | $+0.34$ | $63.02_{0.08}$ | $+0.38$ | $69.45_{0.31}$ | $+0.02$ |
| Least Confidence | $71.28_{0.16}$ | $-0.10$ | $63.17_{0.10}$ | $+0.52$ | $69.55_{0.33}$ | $+0.12$ |
| Loss | $71.88_{0.12}$ | $+0.50$ | $63.10_{0.15}$ | $+0.45$ | $69.36_{0.30}$ | $-0.07$ |
| CAL | $72.13_{0.20}$ | $+0.75$ | $63.09_{0.10}$ | $+0.44$ | $69.20_{0.28}$ | $-0.23$ |
| CRAIG | $71.65_{0.00}$ | $+0.27$ | $63.19_{0.00}$ | $+0.54$ | $69.30_{0.00}$ | $-0.13$ |
| DeepFool | $71.55_{0.20}$ | $+0.18$ | $63.26_{0.07}$ | $+0.62$ | $68.93_{0.30}$ | $-0.50$ |
| Forgetting | $71.58_{0.03}$ | $+0.20$ | $63.26_{0.03}$ | $+0.62$ | $69.66_{0.14}$ | $+0.23$ |
| Glister | $71.65_{0.00}$ | $+0.27$ | $63.19_{0.00}$ | $+0.54$ | $69.30_{0.00}$ | $-0.13$ |
| GraNd | $71.67_{0.07}$ | $+0.29$ | $63.15_{0.04}$ | $+0.50$ | $69.79_{0.40}$ | $+0.36$ |
| Herding | $71.18_{0.11}$ | $-0.19$ | $63.14_{0.04}$ | $+0.49$ | $69.15_{0.11}$ | $-0.28$ |
| KCenter | $71.32_{0.37}$ | $-0.05$ | $63.02_{0.07}$ | $+0.38$ | $69.01_{0.27}$ | $-0.42$ |
| Graph-Cut | $71.65_{0.00}$ | $+0.27$ | $63.19_{0.00}$ | $+0.54$ | $69.30_{0.00}$ | $-0.13$ |
| Cartography$_{Easy}$ | $71.93_{0.23}$ | $+0.56$ | $63.19_{0.06}$ | $+0.55$ | $69.89_{0.23}$ | $+0.46$ |
| Cartography$_{Ambiguous}$ | $71.29_{0.35}$ | $-0.09$ | $62.53_{0.40}$ | $-0.12$ | $69.68_{0.47}$ | $+0.25$ |
| Cartography$_{Hard}$ | $71.49_{0.34}$ | $+0.12$ | $63.00_{0.07}$ | $+0.35$ | $68.48_{0.32}$ | $-0.95$ |
| Cartography$_{Easy+Ambig.}$ | $71.44_{0.35}$ | $+0.06$ | $62.51_{0.32}$ | $-0.14$ | $69.78_{0.37}$ | $+0.35$ |
| ACSESS$_{Uniform}$ | $72.78_{0.12}$ | $+1.40$ | $63.45_{0.20}$ | $+0.80$ | $70.27_{0.22}$ | $+0.84$ |
| ACSESS$_{Weighted}$ | $73.08_{0.25}$ | $+1.70$ | $63.55_{0.26}$ | $+0.90$ | $69.90_{0.27}$ | $+0.47$ |
| ACSESS$_{With Random}$ | $72.66_{0.56}$ | $+1.28$ | $63.28_{0.54}$ | $+0.63$ | $69.26_{0.51}$ | $-0.17$ |

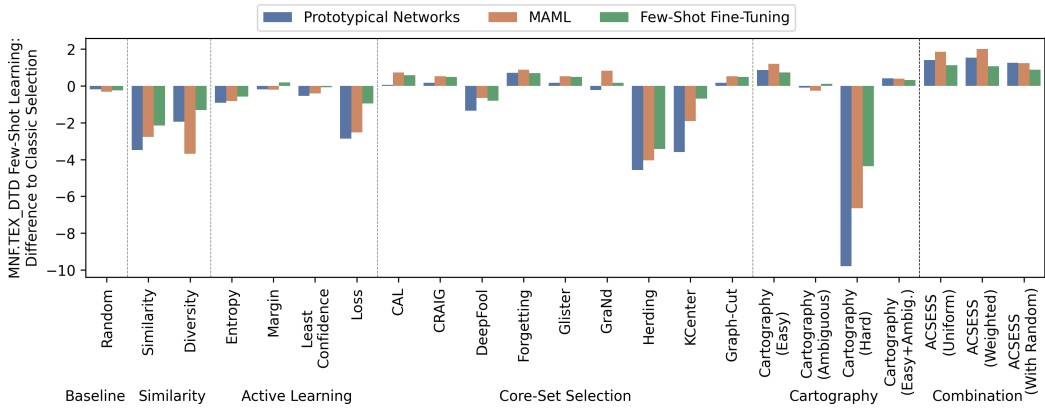

Figure 18: Benefit of the different selection strategies calculated as the difference to the classic few-shot selection strategy for the MNF.TEX_DTD dataset. The performance of the classic selection is represented as the zero value.

Table 8: Benefit of the different selection strategies calculated as the difference in accuracy to the classic few-shot selection strategy for the MNF.TEX_DTD dataset. The subscript represents the standard deviation in the accuracy from running the selection strategy multiple times. The difference to the classic selection baseline (first row) is also included.

| MNF.TEX_DTD | PROTONET | | MAML | | FINE-TUNING | |
| --- | --- | --- | --- | --- | --- | --- |
| | Acc. + Std | Diff. | Acc. + Std | Diff. | Acc. + Std | Diff. |
| Classic | $58.56_{0.42}$ | $+0.00$ | $55.32_{0.33}$ | $+0.00$ | $55.15_{0.24}$ | $+0.00$ |
| Random | $58.38_{0.61}$ | $-0.18$ | $55.00_{0.61}$ | $-0.32$ | $54.90_{0.38}$ | $-0.24$ |
| Similarity | $55.07_{0.86}$ | $-3.49$ | $52.56_{0.76}$ | $-2.77$ | $52.99_{0.56}$ | $-2.16$ |
| Diversity | $56.61_{0.23}$ | $-1.95$ | $51.63_{0.22}$ | $-3.69$ | $53.83_{0.12}$ | $-1.32$ |
| Entropy | $57.64_{0.20}$ | $-0.92$ | $54.50_{0.81}$ | $-0.82$ | $54.57_{0.40}$ | $-0.57$ |
| Margin | $58.37_{0.35}$ | $-0.19$ | $55.11_{0.20}$ | $-0.22$ | $55.34_{0.32}$ | $+0.19$ |
| Least Confidence | $58.02_{0.15}$ | $-0.54$ | $54.91_{0.38}$ | $-0.41$ | $55.07_{0.28}$ | $-0.08$ |
| Loss | $55.70_{0.30}$ | $-2.86$ | $52.79_{0.33}$ | $-2.53$ | $54.20_{0.28}$ | $-0.95$ |
| CAL | $58.62_{0.30}$ | $+0.06$ | $56.05_{0.30}$ | $+0.73$ | $55.73_{0.25}$ | $+0.58$ |
| CRAIG | $58.73_{0.00}$ | $+0.17$ | $55.85_{0.00}$ | $+0.52$ | $55.62_{0.00}$ | $+0.48$ |
| DeepFool | $57.21_{0.43}$ | $-1.36$ | $54.66_{0.09}$ | $-0.66$ | $54.34_{0.16}$ | $-0.81$ |
| Forgetting | $59.28_{0.09}$ | $+0.72$ | $56.21_{0.29}$ | $+0.89$ | $55.83_{0.10}$ | $+0.68$ |
| Glister | $58.73_{0.00}$ | $+0.17$ | $55.85_{0.00}$ | $+0.52$ | $55.62_{0.00}$ | $+0.48$ |
| GraNd | $58.33_{0.52}$ | $-0.23$ | $56.15_{0.37}$ | $+0.83$ | $55.31_{0.27}$ | $+0.16$ |
| Herding | $53.99_{0.50}$ | $-4.57$ | $51.27_{0.49}$ | $-4.06$ | $51.71_{0.19}$ | $-3.44$ |
| KCenter | $54.96_{0.30}$ | $-3.60$ | $53.41_{0.10}$ | $-1.92$ | $54.45_{0.12}$ | $-0.70$ |
| Graph-Cut | $58.73_{0.00}$ | $+0.17$ | $55.85_{0.00}$ | $+0.52$ | $55.62_{0.00}$ | $+0.48$ |
| Cartography$_{Easy}$ | $59.42_{0.30}$ | $+0.86$ | $56.52_{0.25}$ | $+1.20$ | $55.88_{0.28}$ | $+0.74$ |
| Cartography$_{Ambiguous}$ | $58.46_{0.38}$ | $-0.10$ | $55.07_{0.34}$ | $-0.26$ | $55.26_{0.34}$ | $+0.11$ |
| Cartography$_{Hard}$ | $48.76_{0.55}$ | $-9.80$ | $48.68_{0.42}$ | $-6.64$ | $50.78_{0.27}$ | $-4.37$ |
| Cartography$_{Easy+Ambig.}$ | $58.97_{0.14}$ | $+0.41$ | $55.71_{0.28}$ | $+0.38$ | $55.47_{0.31}$ | $+0.32$ |
| ACSESS$_{Uniform}$ | $59.96_{0.14}$ | $+1.40$ | $57.18_{0.13}$ | $+1.85$ | $56.28_{0.26}$ | $+1.13$ |
| ACSESS$_{Weighted}$ | $60.10_{0.30}$ | $+1.54$ | $57.33_{0.26}$ | $+2.00$ | $56.21_{0.27}$ | $+1.07$ |
| ACSESS$_{With Random}$ | $59.82_{0.60}$ | $+1.26$ | $56.57_{0.46}$ | $+1.24$ | $56.03_{0.50}$ | $+0.89$ |

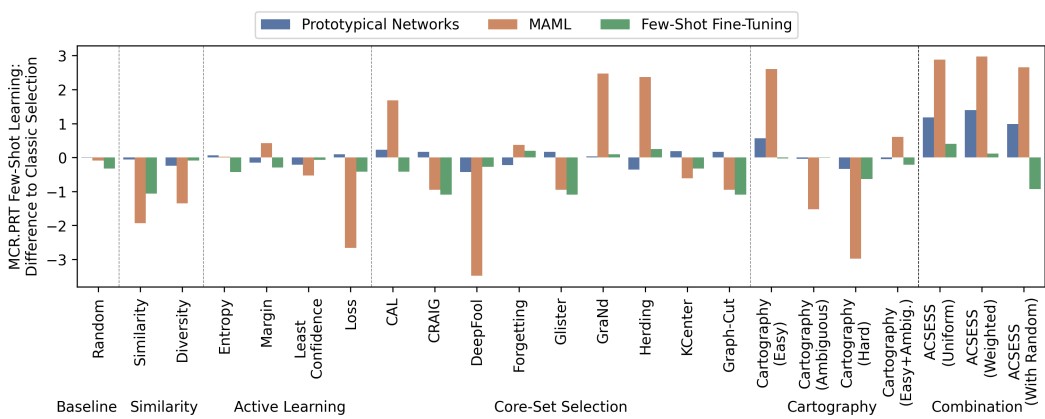

Figure 19: Benefit of the different selection strategies calculated as the difference to the classic few-shot selection strategy for the MCR.PRT dataset. The performance of the classic selection is represented as the zero value.

Table 9: Benefit of the different selection strategies calculated as the difference in accuracy to the classic few-shot selection strategy for the MCR.PRT dataset. The subscript represents the standard deviation in the accuracy from running the selection strategy multiple times. The difference to the classic selection baseline (first row) is also included.

| MCR.PRT | PROTONET | | MAML | | FINE-TUNING | |
|---|---|---|---|---|---|---|
| | Acc. + Std | Diff. | Acc. + Std | Diff. | Acc. + Std | Diff. |
| Classic | $39.85_{0.39}$ | $+0.00$ | $28.44_{0.29}$ | $+0.00$ | $37.57_{0.26}$ | $+0.00$ |
| Random | $39.86_{0.25}$ | $+0.01$ | $28.34_{1.41}$ | $-0.09$ | $37.24_{0.59}$ | $-0.33$ |
| Similarity | $39.79_{0.22}$ | $-0.06$ | $26.50_{2.06}$ | $-1.93$ | $36.51_{0.42}$ | $-1.06$ |
| Diversity | $39.61_{0.22}$ | $-0.24$ | $27.09_{0.48}$ | $-1.35$ | $37.48_{0.19}$ | $-0.09$ |
| Entropy | $39.92_{0.06}$ | $+0.07$ | $28.46_{1.05}$ | $+0.03$ | $37.14_{0.33}$ | $-0.43$ |
| Margin | $39.70_{0.29}$ | $-0.15$ | $28.86_{0.51}$ | $+0.42$ | $37.27_{0.28}$ | $-0.29$ |
| Least Confidence | $39.64_{0.10}$ | $-0.21$ | $27.91_{0.49}$ | $-0.53$ | $37.50_{0.15}$ | $-0.07$ |
| Loss | $39.94_{0.29}$ | $+0.09$ | $25.78_{1.39}$ | $-2.66$ | $37.15_{0.46}$ | $-0.42$ |
| CAL | $40.08_{0.38}$ | $+0.22$ | $30.12_{0.69}$ | $+1.68$ | $37.15_{0.13}$ | $-0.42$ |
| CRAIG | $40.02_{0.00}$ | $+0.17$ | $27.48_{0.00}$ | $-0.95$ | $36.47_{0.00}$ | $-1.10$ |
| DeepFool | $39.42_{0.00}$ | $-0.43$ | $24.95_{0.00}$ | $-3.48$ | $37.29_{0.00}$ | $-0.28$ |
| Forgetting | $39.63_{0.00}$ | $-0.22$ | $28.81_{0.00}$ | $+0.37$ | $37.76_{0.00}$ | $+0.19$ |
| Glister | $40.02_{0.00}$ | $+0.17$ | $27.48_{0.00}$ | $-0.95$ | $36.47_{0.00}$ | $-1.10$ |
| GraNd | $39.89_{0.14}$ | $+0.04$ | $30.90_{0.75}$ | $+2.47$ | $37.66_{0.30}$ | $+0.09$ |
| Herding | $39.50_{0.00}$ | $-0.36$ | $30.80_{0.00}$ | $+2.37$ | $37.82_{0.00}$ | $+0.25$ |
| KCenter | $40.04_{0.00}$ | $+0.18$ | $27.82_{0.00}$ | $-0.61$ | $37.24_{0.00}$ | $-0.33$ |
| Graph-Cut | $40.02_{0.00}$ | $+0.17$ | $27.48_{0.00}$ | $-0.95$ | $36.47_{0.00}$ | $-1.10$ |
| Cartography$_{\text{Easy}}$ | $40.42_{0.20}$ | $+0.57$ | $31.04_{0.62}$ | $+2.60$ | $37.54_{0.20}$ | $-0.03$ |
| Cartography$_{\text{Ambiguous}}$ | $39.82_{0.37}$ | $-0.04$ | $26.91_{0.84}$ | $-1.52$ | $37.56_{0.43}$ | $-0.00$ |
| Cartography$_{\text{Hard}}$ | $39.52_{0.16}$ | $-0.34$ | $25.46_{1.18}$ | $-2.97$ | $36.93_{0.17}$ | $-0.63$ |
| Cartography$_{\text{Easy+Ambig.}}$ | $39.80_{0.24}$ | $-0.05$ | $29.04_{0.57}$ | $+0.60$ | $37.35_{0.36}$ | $-0.22$ |
| ACSESS$_{\text{Uniform}}$ | $41.03_{0.19}$ | $+1.18$ | $31.31_{0.26}$ | $+2.88$ | $37.97_{0.26}$ | $+0.40$ |
| ACSESS$_{\text{Weighted}}$ | $41.25_{0.24}$ | $+1.40$ | $31.41_{0.39}$ | $+2.97$ | $37.68_{0.36}$ | $+0.11$ |
| ACSESS$_{\text{With Random}}$ | $40.84_{0.47}$ | $+0.99$ | $31.09_{0.69}$ | $+2.65$ | $36.63_{0.86}$ | $-0.93$ |

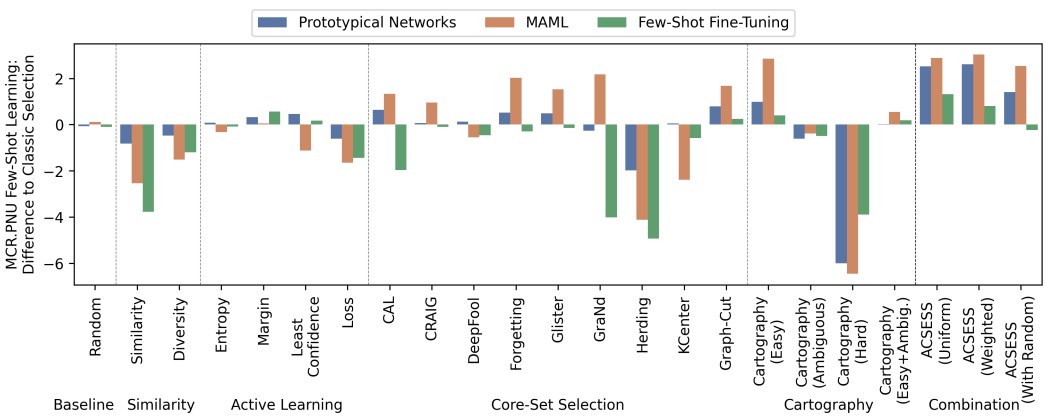

Figure 20: Benefit of the different selection strategies calculated as the difference to the classic few-shot selection strategy for the MCR.PNU dataset. The performance of the classic selection is represented as the zero value.

Table 10: Benefit of the different selection strategies calculated as the difference in accuracy to the classic few-shot selection strategy for the MCR.PNU dataset. The subscript represents the standard deviation in the accuracy from running the selection strategy multiple times. The difference to the classic selection baseline (first row) is also included.

| MCR.PNU | PROTONET | | MAML | | FINE-TUNING | |
|---|---|---|---|---|---|---|
| | Acc. + Std | Diff. | Acc. + Std | Diff. | Acc. + Std | Diff. |
| Classic | $37.34_{0.31}$ | $+0.00$ | $27.32_{0.34}$ | $+0.00$ | $51.25_{0.26}$ | $+0.00$ |
| Random | $37.27_{0.41}$ | $-0.07$ | $27.43_{0.82}$ | $+0.11$ | $51.15_{0.55}$ | $-0.10$ |
| Similarity | $36.52_{1.17}$ | $-0.82$ | $24.78_{1.33}$ | $-2.54$ | $47.48_{0.87}$ | $-3.77$ |
| Diversity | $36.86_{0.17}$ | $-0.47$ | $25.82_{0.28}$ | $-1.51$ | $50.05_{0.31}$ | $-1.20$ |
| Entropy | $37.42_{0.22}$ | $+0.08$ | $26.99_{1.29}$ | $-0.33$ | $51.17_{0.43}$ | $-0.08$ |
| Margin | $37.66_{0.46}$ | $+0.33$ | $27.38_{0.48}$ | $+0.05$ | $51.81_{0.79}$ | $+0.56$ |
| Least Confidence | $37.80_{0.09}$ | $+0.46$ | $26.20_{0.63}$ | $-1.12$ | $51.43_{0.38}$ | $+0.18$ |
| Loss | $36.73_{0.22}$ | $-0.60$ | $25.67_{0.42}$ | $-1.65$ | $49.82_{0.63}$ | $-1.44$ |
| CAL | $37.97_{0.20}$ | $+0.63$ | $28.65_{1.12}$ | $+1.32$ | $49.29_{0.51}$ | $-1.96$ |
| CRAIG | $37.40_{0.00}$ | $+0.06$ | $28.27_{0.00}$ | $+0.95$ | $51.15_{0.00}$ | $-0.10$ |
| DeepFool | $37.47_{0.00}$ | $+0.13$ | $26.78_{0.00}$ | $-0.55$ | $50.79_{0.00}$ | $-0.46$ |
| Forgetting | $37.86_{0.00}$ | $+0.52$ | $29.35_{0.00}$ | $+2.03$ | $50.96_{0.00}$ | $-0.29$ |
| Glister | $37.83_{0.00}$ | $+0.49$ | $28.86_{0.00}$ | $+1.53$ | $51.12_{0.00}$ | $-0.14$ |
| GraNd | $37.07_{0.24}$ | $-0.27$ | $29.50_{0.39}$ | $+2.17$ | $47.25_{0.44}$ | $-4.00$ |
| Herding | $35.35_{0.00}$ | $-1.98$ | $23.21_{0.00}$ | $-4.11$ | $46.32_{0.00}$ | $-4.93$ |
| KCenter | $37.40_{0.00}$ | $+0.06$ | $24.93_{0.00}$ | $-2.39$ | $50.68_{0.00}$ | $-0.57$ |
| Graph-Cut | $38.13_{0.00}$ | $+0.79$ | $29.01_{0.00}$ | $+1.68$ | $51.51_{0.00}$ | $+0.26$ |
| Cartography$_{Easy}$ | $38.32_{0.30}$ | $+0.98$ | $30.18_{0.60}$ | $+2.86$ | $51.65_{0.34}$ | $+0.40$ |
| Cartography$_{Ambiguous}$ | $36.73_{0.23}$ | $-0.60$ | $26.94_{0.24}$ | $-0.38$ | $50.77_{0.37}$ | $-0.48$ |
| Cartography$_{Hard}$ | $31.34_{0.64}$ | $-5.99$ | $20.87_{0.53}$ | $-6.45$ | $47.36_{0.73}$ | $-3.89$ |
| Cartography$_{Easy+Ambig.}$ | $37.35_{0.12}$ | $+0.02$ | $27.87_{0.26}$ | $+0.54$ | $51.44_{0.26}$ | $+0.19$ |
| ACSESS$_{Uniform}$ | $39.86_{0.20}$ | $+2.52$ | $30.20_{0.17}$ | $+2.88$ | $52.56_{0.20}$ | $+1.31$ |
| ACSESS$_{Weighted}$ | $39.95_{0.21}$ | $+2.61$ | $30.35_{0.26}$ | $+3.03$ | $52.05_{0.40}$ | $+0.80$ |
| ACSESS$_{With Random}$ | $38.75_{0.40}$ | $+1.41$ | $29.86_{0.58}$ | $+2.54$ | $51.02_{0.92}$ | $-0.23$ |

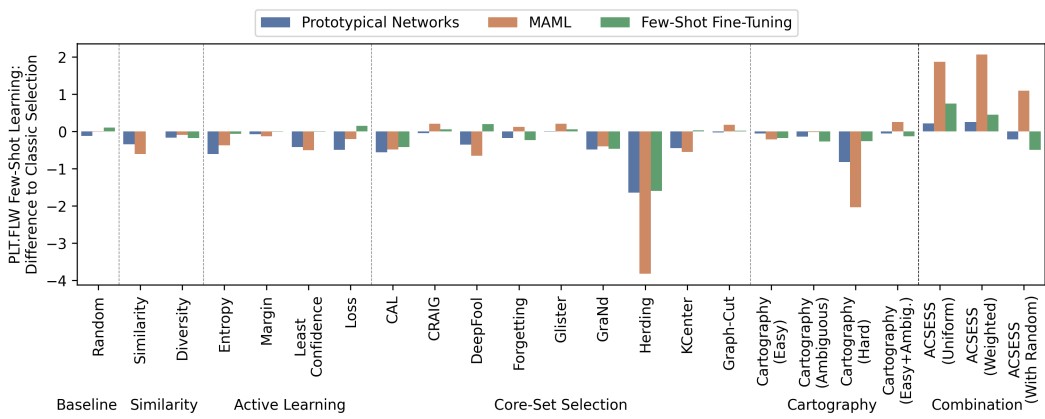

Figure 21: Benefit of the different selection strategies calculated as the difference to the classic few-shot selection strategy for the PLT.FLW dataset. The performance of the classic selection is represented as the zero value.

Table 11: Benefit of the different selection strategies calculated as the difference in accuracy to the classic few-shot selection strategy for the PLT.FLW dataset. The subscript represents the standard deviation in the accuracy from running the selection strategy multiple times. The difference to the classic selection baseline (first row) is also included.

| PLT.FLW | PROTONET | | MAML | | FINE-TUNING | |
|---|---|---|---|---|---|---|
| | Acc. + Std | Diff. | Acc. + Std | Diff. | Acc. + Std | Diff. |
| Classic | $91.41_{0.25}$ | $+0.00$ | $85.47_{0.31}$ | $+0.00$ | $91.72_{0.22}$ | $+0.00$ |
| Random | $91.29_{0.12}$ | $-0.12$ | $85.45_{0.16}$ | $-0.01$ | $91.82_{0.12}$ | $+0.10$ |
| Similarity | $91.07_{0.13}$ | $-0.34$ | $84.86_{0.29}$ | $-0.61$ | $91.72_{0.18}$ | $-0.00$ |
| Diversity | $91.25_{0.07}$ | $-0.16$ | $85.37_{0.15}$ | $-0.10$ | $91.54_{0.09}$ | $-0.18$ |
| Entropy | $90.80_{0.26}$ | $-0.61$ | $85.09_{0.19}$ | $-0.38$ | $91.66_{0.10}$ | $-0.07$ |
| Margin | $91.33_{0.22}$ | $-0.08$ | $85.33_{0.14}$ | $-0.14$ | $91.71_{0.07}$ | $-0.01$ |
| Least Confidence | $90.99_{0.08}$ | $-0.42$ | $84.96_{0.33}$ | $-0.50$ | $91.71_{0.06}$ | $-0.01$ |
| Loss | $90.91_{0.16}$ | $-0.50$ | $85.26_{0.16}$ | $-0.20$ | $91.87_{0.04}$ | $+0.14$ |
| CAL | $90.85_{0.02}$ | $-0.56$ | $84.98_{0.33}$ | $-0.49$ | $91.30_{0.10}$ | $-0.43$ |
| CRAIG | $91.37_{0.01}$ | $-0.04$ | $85.67_{0.00}$ | $+0.20$ | $91.78_{0.00}$ | $+0.06$ |
| DeepFool | $91.06_{0.20}$ | $-0.35$ | $84.81_{0.15}$ | $-0.65$ | $91.92_{0.12}$ | $+0.19$ |
| Forgetting | $91.24_{0.15}$ | $-0.18$ | $85.59_{0.14}$ | $+0.12$ | $91.49_{0.08}$ | $-0.24$ |
| Glister | $91.42_{0.01}$ | $+0.01$ | $85.67_{0.00}$ | $+0.20$ | $91.78_{0.00}$ | $+0.06$ |
| GraNd | $90.92_{0.19}$ | $-0.49$ | $85.06_{0.08}$ | $-0.40$ | $91.26_{0.08}$ | $-0.46$ |
| Herding | $89.76_{0.08}$ | $-1.65$ | $81.64_{0.34}$ | $-3.82$ | $90.12_{0.10}$ | $-1.60$ |
| KCenter | $90.96_{0.01}$ | $-0.45$ | $84.91_{0.13}$ | $-0.56$ | $91.75_{0.14}$ | $+0.02$ |
| Graph-Cut | $91.38_{0.00}$ | $-0.03$ | $85.64_{0.04}$ | $+0.17$ | $91.75_{0.01}$ | $+0.02$ |
| Cartography$_{Easy}$ | $91.35_{0.08}$ | $-0.06$ | $85.25_{0.15}$ | $-0.22$ | $91.55_{0.10}$ | $-0.18$ |
| Cartography$_{Ambiguous}$ | $91.27_{0.27}$ | $-0.14$ | $85.45_{0.35}$ | $-0.02$ | $91.45_{0.16}$ | $-0.28$ |
| Cartography$_{Hard}$ | $90.59_{0.16}$ | $-0.82$ | $83.43_{0.20}$ | $-2.04$ | $91.46_{0.11}$ | $-0.26$ |
| Cartography$_{Easy+Ambig.}$ | $91.35_{0.21}$ | $-0.06$ | $85.72_{0.15}$ | $+0.25$ | $91.59_{0.18}$ | $-0.13$ |
| ACSESS$_{Uniform}$ | $91.63_{0.10}$ | $+0.22$ | $87.34_{0.29}$ | $+1.87$ | $92.47_{0.13}$ | $+0.75$ |
| ACSESS$_{Weighted}$ | $91.66_{0.16}$ | $+0.25$ | $87.53_{0.37}$ | $+2.06$ | $92.17_{0.22}$ | $+0.44$ |
| ACSESS$_{With Random}$ | $91.20_{0.69}$ | $-0.21$ | $86.56_{0.46}$ | $+1.09$ | $91.22_{0.64}$ | $-0.50$ |

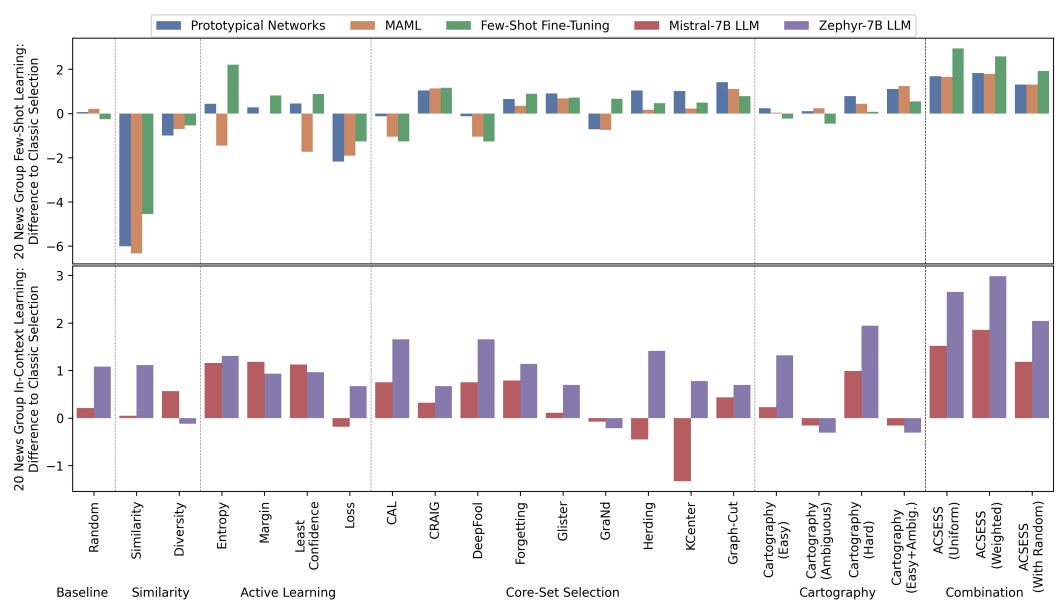

Figure 22: Benefit of the different selection strategies calculated as the difference to the classic few-shot selection strategy for the 20 News Group dataset. The performance of the classic selection is represented as the zero value.

Table 12: Benefit of the different selection strategies calculated as the difference in accuracy to the classic few-shot selection strategy for the 20 News Group dataset. The subscript represents the standard deviation in the accuracy from running the selection strategy multiple times. The difference to the classic selection baseline (first row) is also included.

| 20 News Group | PROTONET | | MAML | | FINE-TUNING | | MISTRAL | | ZEPHYR | |
|---|---|---|---|---|---|---|---|---|---|---|
| | Acc. + Std | Diff. | Acc. + Std | Diff. | Acc. + Std | Diff. | Acc. + Std | Diff. | Acc. + Std | Diff. |
| Classic | $59.41_{0.26}$ | +0.00 | $56.03_{0.29}$ | +0.00 | $43.88_{0.26}$ | +0.00 | $75.81_{1.24}$ | +0.00 | $74.58_{1.24}$ | +0.00 |
| Random | $59.47_{0.98}$ | +0.06 | $56.23_{0.84}$ | +0.21 | $43.62_{1.04}$ | −0.26 | $76.02_{0.93}$ | +0.21 | $75.66_{0.66}$ | +1.08 |
| LENS | N/A | N/A | N/A | N/A | N/A | N/A | $77.30_{0.43}$ | +1.49 | $76.42_{0.33}$ | +1.84 |
| Similarity | $53.40_{1.56}$ | −6.01 | $49.69_{1.35}$ | −6.34 | $39.33_{1.42}$ | −4.55 | $75.86_{0.91}$ | +0.05 | $75.70_{0.46}$ | +1.11 |
| Diversity | $58.41_{0.31}$ | −1.00 | $55.33_{0.27}$ | −0.70 | $43.34_{0.33}$ | −0.54 | $76.38_{0.34}$ | +0.57 | $74.46_{0.44}$ | −0.12 |
| Entropy | $59.85_{0.21}$ | +0.44 | $54.57_{0.54}$ | −1.46 | $46.09_{0.83}$ | +2.21 | $76.97_{1.02}$ | +1.16 | $75.89_{0.49}$ | +1.31 |
| Margin | $59.68_{1.34}$ | +0.27 | $56.02_{0.35}$ | −0.01 | $44.70_{0.36}$ | +0.82 | $76.99_{0.62}$ | +1.18 | $75.52_{0.34}$ | +0.94 |
| Least Confidence | $59.86_{0.65}$ | +0.45 | $54.28_{0.72}$ | −1.74 | $44.76_{2.45}$ | +0.88 | $76.93_{0.59}$ | +1.12 | $75.55_{0.56}$ | +0.96 |
| Loss | $57.23_{0.95}$ | −2.18 | $54.12_{1.72}$ | −1.91 | $42.62_{1.53}$ | −1.27 | $75.63_{1.18}$ | −0.18 | $75.25_{1.00}$ | +0.67 |
| CAL | $59.28_{0.00}$ | −0.13 | $54.98_{0.00}$ | −1.05 | $42.61_{0.00}$ | −1.27 | $76.56_{0.00}$ | +0.75 | $76.24_{0.00}$ | +1.66 |
| CRAIG | $60.45_{0.01}$ | +1.04 | $57.16_{0.00}$ | +1.13 | $45.04_{0.17}$ | +1.16 | $76.13_{0.04}$ | +0.32 | $75.25_{0.30}$ | +0.67 |
| DeepFool | $59.28_{0.00}$ | −0.13 | $54.98_{0.00}$ | −1.05 | $42.61_{0.00}$ | −1.27 | $76.56_{0.00}$ | +0.75 | $76.24_{0.00}$ | +1.66 |
| Forgetting | $60.06_{0.00}$ | +0.65 | $56.38_{0.00}$ | +0.35 | $44.78_{0.00}$ | +0.90 | $76.60_{0.00}$ | +0.79 | $75.72_{0.00}$ | +1.14 |
| Glister | $60.32_{0.00}$ | +0.91 | $56.71_{0.00}$ | +0.68 | $44.60_{0.00}$ | +0.72 | $75.92_{0.00}$ | +0.11 | $75.28_{0.00}$ | +0.70 |
| GraNd | $58.70_{0.74}$ | −0.71 | $55.28_{1.03}$ | −0.74 | $44.55_{0.73}$ | +0.67 | $75.73_{0.87}$ | −0.08 | $74.37_{0.77}$ | −0.21 |
| Herding | $60.44_{0.00}$ | +1.03 | $56.19_{0.00}$ | +0.17 | $44.35_{0.00}$ | +0.47 | $75.36_{0.00}$ | −0.45 | $76.00_{0.00}$ | +1.42 |
| KCenter | $60.42_{0.00}$ | +1.01 | $56.25_{0.00}$ | +0.22 | $44.38_{0.00}$ | +0.50 | $74.48_{0.00}$ | −1.33 | $75.36_{0.00}$ | +0.78 |
| Graph-Cut | $60.82_{0.00}$ | +1.41 | $57.13_{0.00}$ | +1.11 | $44.67_{0.00}$ | +0.79 | $76.24_{0.00}$ | +0.43 | $75.28_{0.00}$ | +0.70 |
| Cartography$_{Easy}$ | $59.65_{0.50}$ | +0.24 | $56.06_{0.36}$ | +0.03 | $43.65_{0.61}$ | −0.24 | $76.04_{0.86}$ | +0.23 | $75.90_{0.76}$ | +1.32 |
| Cartography$_{Ambiguous}$ | $59.51_{0.31}$ | +0.10 | $56.26_{0.31}$ | +0.24 | $43.42_{0.30}$ | −0.46 | $75.65_{1.90}$ | −0.16 | $74.28_{1.52}$ | −0.30 |
| Cartography$_{Hard}$ | $60.20_{0.47}$ | +0.79 | $56.46_{0.35}$ | +0.43 | $43.96_{0.57}$ | +0.08 | $76.80_{0.85}$ | +0.99 | $76.53_{0.29}$ | +1.94 |
| Cartography$_{Easy+Ambig.}$ | $60.51_{0.31}$ | +1.10 | $57.26_{0.31}$ | +1.24 | $44.42_{0.30}$ | +0.54 | $75.65_{1.90}$ | −0.16 | $74.28_{1.52}$ | −0.30 |
| ACSESS$_{Uniform}$ | $61.09_{0.23}$ | +1.68 | $57.68_{0.24}$ | +1.66 | $46.82_{0.28}$ | +2.94 | $77.33_{0.33}$ | +1.52 | $77.24_{0.22}$ | +2.65 |
| ACSESS$_{Weighted}$ | $61.24_{0.30}$ | +1.83 | $57.82_{0.35}$ | +1.79 | $46.46_{0.30}$ | +2.58 | $77.66_{0.56}$ | +1.86 | $77.57_{0.33}$ | +2.98 |
| ACSESS$_{With\ Random}$ | $60.72_{0.37}$ | +1.31 | $57.34_{0.46}$ | +1.31 | $45.80_{0.89}$ | +1.92 | $76.99_{1.68}$ | +1.18 | $76.63_{1.25}$ | +2.04 |

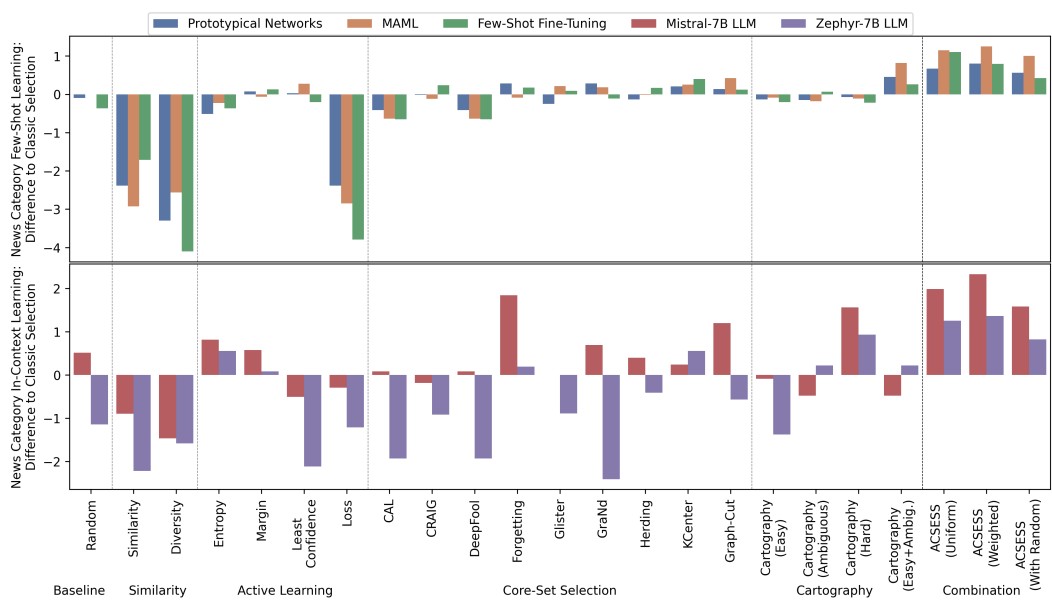

Figure 23: Benefit of the different selection strategies calculated as the difference to the classic few-shot selection strategy for the News Category dataset. The performance of the classic selection is represented as the zero value.

Table 13: Benefit of the different selection strategies calculated as the difference in accuracy to the classic few-shot selection strategy for the News Category dataset. The subscript represents the standard deviation in the accuracy from running the selection strategy multiple times. The difference to the classic selection baseline (first row) is also included.

| News Category | PROTONET | | MAML | | FINE-TUNING | | MISTRAL | | ZEPHYR | |
|---|---|---|---|---|---|---|---|---|---|---|
| | Acc. + Std | Diff. | Acc. + Std | Diff. | Acc. + Std | Diff. | Acc. + Std | Diff. | Acc. + Std | Diff. |
| Classic | $58.39_{0.25}$ | $+0.00$ | $55.00_{0.24}$ | $+0.00$ | $45.56_{0.30}$ | $+0.00$ | $70.64_{1.66}$ | $+0.00$ | $70.73_{1.56}$ | $+0.00$ |
| Random | $58.29_{0.53}$ | $-0.10$ | $55.00_{0.43}$ | $+0.00$ | $45.20_{0.35}$ | $-0.37$ | $71.15_{1.02}$ | $+0.51$ | $69.58_{0.52}$ | $-1.14$ |
| LENS | N/A | N/A | N/A | N/A | N/A | N/A | $72.10_{0.44}$ | $+1.46$ | $71.64_{0.48}$ | $+0.91$ |
| Similarity | $56.00_{0.74}$ | $-2.39$ | $52.07_{1.20}$ | $-2.92$ | $43.85_{0.89}$ | $-1.71$ | $69.74_{1.24}$ | $-0.90$ | $68.51_{1.09}$ | $-2.22$ |
| Diversity | $55.09_{0.28}$ | $-3.30$ | $52.43_{0.32}$ | $-2.56$ | $41.46_{0.42}$ | $-4.10$ | $69.18_{0.79}$ | $-1.46$ | $69.14_{0.75}$ | $-1.58$ |
| Entropy | $57.87_{0.34}$ | $-0.51$ | $54.77_{0.09}$ | $-0.22$ | $45.20_{0.30}$ | $-0.36$ | $71.45_{0.89}$ | $+0.81$ | $71.28_{0.64}$ | $+0.55$ |
| Margin | $58.46_{0.68}$ | $+0.08$ | $54.93_{0.66}$ | $-0.06$ | $45.69_{0.61}$ | $+0.12$ | $71.21_{0.15}$ | $+0.57$ | $70.81_{1.10}$ | $+0.08$ |
| Least Confidence | $58.42_{0.48}$ | $+0.03$ | $55.27_{0.59}$ | $+0.28$ | $45.36_{0.45}$ | $-0.20$ | $70.13_{0.19}$ | $-0.51$ | $68.61_{0.31}$ | $-2.12$ |
| Loss | $56.00_{0.61}$ | $-2.39$ | $52.14_{0.39}$ | $-2.85$ | $41.77_{0.30}$ | $-3.79$ | $70.35_{0.51}$ | $-0.29$ | $69.52_{0.23}$ | $-1.21$ |
| CAL | $57.97_{0.00}$ | $-0.42$ | $54.36_{0.00}$ | $-0.64$ | $44.91_{0.00}$ | $-0.66$ | $70.72_{0.00}$ | $+0.08$ | $68.80_{0.00}$ | $-1.93$ |
| CRAIG | $58.36_{0.04}$ | $-0.02$ | $54.87_{0.07}$ | $-0.12$ | $45.80_{0.01}$ | $+0.24$ | $70.45_{0.42}$ | $-0.19$ | $69.81_{0.19}$ | $-0.92$ |
| DeepFool | $57.97_{0.00}$ | $-0.42$ | $54.36_{0.00}$ | $-0.64$ | $44.91_{0.00}$ | $-0.66$ | $70.72_{0.00}$ | $+0.08$ | $68.80_{0.00}$ | $-1.93$ |
| Forgetting | $58.67_{0.00}$ | $+0.29$ | $54.91_{0.00}$ | $-0.09$ | $45.74_{0.00}$ | $+0.18$ | $72.48_{0.00}$ | $+1.84$ | $70.92_{0.00}$ | $+0.19$ |
| Glister | $58.13_{0.00}$ | $-0.25$ | $55.21_{0.00}$ | $+0.22$ | $45.65_{0.00}$ | $+0.09$ | $70.64_{0.00}$ | $+0.00$ | $69.84_{0.00}$ | $-0.89$ |
| GraNd | $58.67_{0.14}$ | $+0.28$ | $55.18_{0.08}$ | $+0.19$ | $45.45_{0.21}$ | $-0.11$ | $71.33_{0.72}$ | $+0.69$ | $68.32_{0.17}$ | $-2.41$ |
| Herding | $58.25_{0.00}$ | $-0.13$ | $54.97_{0.00}$ | $-0.02$ | $45.73_{0.00}$ | $+0.17$ | $71.04_{0.00}$ | $+0.40$ | $70.32_{0.00}$ | $-0.41$ |
| KCenter | $58.60_{0.00}$ | $+0.21$ | $55.25_{0.00}$ | $+0.26$ | $45.96_{0.00}$ | $+0.40$ | $70.88_{0.00}$ | $+0.24$ | $71.28_{0.00}$ | $+0.55$ |
| Graph-Cut | $58.52_{0.00}$ | $+0.14$ | $55.42_{0.00}$ | $+0.43$ | $45.68_{0.00}$ | $+0.12$ | $71.84_{0.00}$ | $+1.20$ | $70.16_{0.00}$ | $-0.57$ |
| Cartography$_{Easy}$ | $58.26_{0.43}$ | $-0.13$ | $54.90_{0.61}$ | $-0.09$ | $45.36_{0.27}$ | $-0.20$ | $70.55_{0.82}$ | $-0.09$ | $69.35_{0.84}$ | $-1.38$ |
| Cartography$_{Ambiguous}$ | $58.24_{0.29}$ | $-0.15$ | $54.82_{0.31}$ | $-0.18$ | $45.63_{0.30}$ | $+0.06$ | $70.16_{1.64}$ | $-0.48$ | $70.94_{1.39}$ | $+0.22$ |
| Cartography$_{Hard}$ | $58.32_{0.36}$ | $-0.07$ | $54.89_{0.40}$ | $-0.11$ | $45.34_{0.54}$ | $-0.22$ | $72.20_{1.04}$ | $+1.56$ | $71.66_{0.70}$ | $+0.93$ |
| Cartography$_{Easy+Ambig.}$ | $58.84_{0.29}$ | $+0.45$ | $55.82_{0.31}$ | $+0.82$ | $45.83_{0.30}$ | $+0.26$ | $70.16_{1.64}$ | $-0.48$ | $70.94_{1.39}$ | $+0.22$ |
| ACSESS$_{Uniform}$ | $59.06_{0.10}$ | $+0.67$ | $56.14_{0.10}$ | $+1.15$ | $46.67_{0.07}$ | $+1.11$ | $72.62_{0.39}$ | $+1.98$ | $71.98_{0.35}$ | $+1.25$ |
| ACSESS$_{Weighted}$ | $59.19_{0.17}$ | $+0.80$ | $56.24_{0.15}$ | $+1.25$ | $46.35_{0.28}$ | $+0.79$ | $72.97_{0.52}$ | $+2.33$ | $72.09_{0.48}$ | $+1.36$ |
| ACSESS$_{With Random}$ | $58.95_{0.34}$ | $+0.56$ | $56.00_{0.34}$ | $+1.00$ | $45.99_{0.73}$ | $+0.42$ | $72.22_{0.87}$ | $+1.58$ | $71.55_{0.86}$ | $+0.82$ |

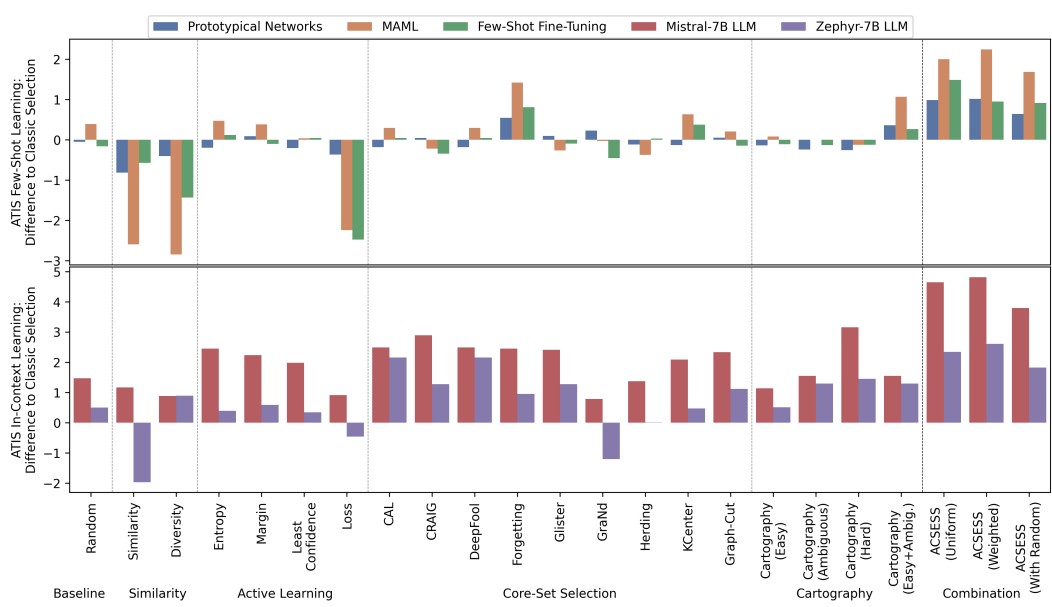

Figure 24: Benefit of the different selection strategies calculated as the difference to the classic few-shot selection strategy for the ATIS dataset. The performance of the classic selection is represented as the zero value.

Table 14: Benefit of the different selection strategies calculated as the difference in accuracy to the classic few-shot selection strategy for the ATIS dataset. The subscript represents the standard deviation in the accuracy from running the selection strategy multiple times. The difference to the classic selection baseline (first row) is also included.

| ATIS | PROTONET | | MAML | | FINE-TUNING | | MISTRAL | | ZEPHYR | |
|---|---|---|---|---|---|---|---|---|---|---|
| | Acc. + Std | Diff. | Acc. + Std | Diff. | Acc. + Std | Diff. | Acc. + Std | Diff. | Acc. + Std | Diff. |
| Classic | $89.90_{0.18}$ | +0.00 | $73.29_{0.26}$ | +0.00 | $73.09_{0.42}$ | +0.00 | $91.19_{1.04}$ | +0.00 | $86.09_{1.39}$ | +0.00 |
| Random | $89.85_{0.31}$ | −0.05 | $73.68_{0.44}$ | +0.39 | $72.93_{0.48}$ | −0.16 | $92.66_{0.58}$ | +1.47 | $86.58_{0.50}$ | +0.50 |
| LENS | N/A | N/A | N/A | N/A | N/A | N/A | $94.36_{0.45}$ | +3.17 | $87.84_{0.32}$ | +1.75 |
| Similarity | $89.08_{0.92}$ | −0.82 | $70.69_{1.83}$ | −2.60 | $72.52_{0.85}$ | −0.57 | $92.36_{1.07}$ | +1.17 | $84.12_{1.28}$ | −1.97 |
| Diversity | $89.50_{0.14}$ | −0.40 | $70.44_{0.29}$ | −2.85 | $71.65_{0.23}$ | −1.44 | $92.07_{0.37}$ | +0.88 | $86.98_{0.76}$ | +0.89 |
| Entropy | $89.70_{0.03}$ | −0.20 | $73.76_{0.41}$ | +0.47 | $73.21_{0.48}$ | +0.12 | $93.64_{0.77}$ | +2.45 | $86.48_{0.99}$ | +0.39 |
| Margin | $89.99_{0.21}$ | +0.09 | $73.67_{0.35}$ | +0.38 | $72.98_{0.58}$ | −0.11 | $93.43_{0.42}$ | +2.24 | $86.68_{0.75}$ | +0.59 |
| Least Confidence | $89.69_{0.30}$ | −0.21 | $73.32_{0.50}$ | +0.04 | $73.13_{0.48}$ | +0.04 | $93.17_{0.20}$ | +1.98 | $86.43_{0.44}$ | +0.34 |
| Loss | $89.54_{0.23}$ | −0.36 | $71.05_{0.66}$ | −2.24 | $70.61_{0.31}$ | −2.48 | $92.11_{0.27}$ | +0.92 | $85.63_{0.78}$ | −0.46 |
| CAL | $89.72_{0.00}$ | −0.18 | $73.58_{0.00}$ | +0.29 | $73.13_{0.00}$ | +0.05 | $93.68_{0.00}$ | +2.49 | $88.24_{0.00}$ | +2.15 |
| CRAIG | $89.94_{0.00}$ | +0.04 | $73.06_{0.00}$ | −0.22 | $72.74_{0.00}$ | −0.34 | $94.08_{0.00}$ | +2.89 | $87.36_{0.00}$ | +1.27 |
| DeepFool | $89.72_{0.00}$ | −0.18 | $73.58_{0.00}$ | +0.29 | $73.13_{0.00}$ | +0.05 | $93.68_{0.00}$ | +2.49 | $88.24_{0.00}$ | +2.15 |
| Forgetting | $90.45_{0.00}$ | +0.55 | $74.71_{0.00}$ | +1.42 | $73.90_{0.00}$ | +0.81 | $93.64_{0.00}$ | +2.45 | $87.04_{0.00}$ | +0.95 |
| Glister | $89.99_{0.00}$ | +0.09 | $73.03_{0.00}$ | −0.26 | $72.99_{0.00}$ | −0.10 | $93.60_{0.00}$ | +2.41 | $87.36_{0.00}$ | +1.27 |
| GraNd | $90.12_{0.57}$ | +0.22 | $73.26_{2.17}$ | −0.03 | $72.63_{1.33}$ | −0.46 | $91.97_{0.72}$ | +0.78 | $84.88_{1.57}$ | −1.21 |
| Herding | $89.78_{0.00}$ | −0.12 | $72.92_{0.00}$ | −0.37 | $73.12_{0.00}$ | +0.03 | $92.56_{0.00}$ | +1.37 | $86.08_{0.00}$ | −0.01 |
| KCenter | $89.77_{0.00}$ | −0.13 | $73.92_{0.00}$ | +0.63 | $73.46_{0.00}$ | +0.38 | $93.28_{0.00}$ | +2.09 | $86.56_{0.00}$ | +0.47 |
| Graph-Cut | $89.95_{0.00}$ | +0.05 | $73.49_{0.00}$ | +0.20 | $72.94_{0.00}$ | −0.14 | $93.52_{0.00}$ | +2.33 | $87.20_{0.00}$ | +1.11 |
| Cartography$_{Easy}$ | $89.76_{0.29}$ | −0.14 | $73.37_{0.67}$ | +0.08 | $72.98_{0.58}$ | −0.11 | $92.33_{0.59}$ | +1.14 | $86.60_{0.82}$ | +0.51 |
| Cartography$_{Ambiguous}$ | $89.66_{0.33}$ | −0.24 | $73.29_{0.40}$ | +0.00 | $72.95_{0.44}$ | −0.13 | $92.74_{1.20}$ | +1.55 | $87.38_{1.02}$ | +1.30 |
| Cartography$_{Hard}$ | $89.64_{0.34}$ | −0.26 | $73.16_{0.69}$ | −0.12 | $72.96_{0.60}$ | −0.12 | $94.34_{0.74}$ | +3.15 | $87.54_{0.95}$ | +1.45 |
| Cartography$_{Easy+Ambig.}$ | $90.26_{0.33}$ | +0.36 | $74.36_{0.40}$ | +1.07 | $73.35_{0.44}$ | +0.27 | $92.74_{1.20}$ | +1.55 | $87.38_{1.02}$ | +1.30 |
| ACSESS$_{Uniform}$ | $90.88_{0.25}$ | +0.98 | $75.29_{0.31}$ | +2.00 | $74.57_{0.22}$ | +1.48 | $95.84_{0.25}$ | +4.65 | $88.43_{0.22}$ | +2.34 |
| ACSESS$_{Weighted}$ | $90.92_{0.30}$ | +1.02 | $75.53_{0.39}$ | +2.24 | $74.04_{0.36}$ | +0.95 | $96.00_{0.44}$ | +4.81 | $88.70_{0.37}$ | +2.61 |
| ACSESS$_{With Random}$ | $90.54_{0.33}$ | +0.64 | $74.97_{0.66}$ | +1.68 | $74.00_{0.69}$ | +0.91 | $94.98_{1.59}$ | +3.79 | $87.91_{1.13}$ | +1.83 |

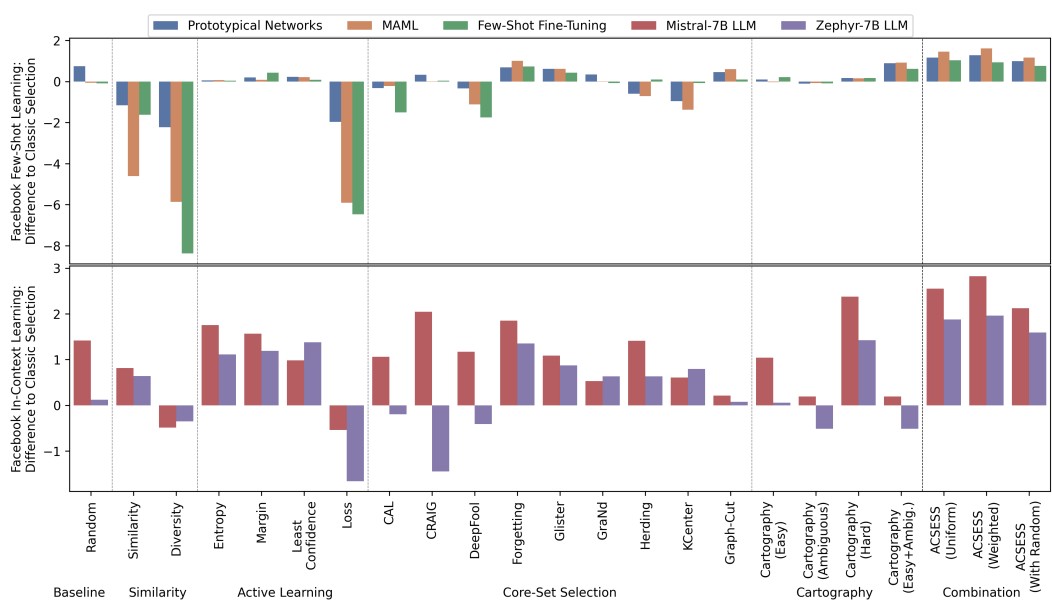

Figure 25: Benefit of the different selection strategies calculated as the difference to the classic few-shot selection strategy for the Facebook dataset. The performance of the classic selection is represented as the zero value.

Table 15: Benefit of the different selection strategies calculated as the difference in accuracy to the classic few-shot selection strategy for the Facebook dataset. The subscript represents the standard deviation in the accuracy from running the selection strategy multiple times. The difference to the classic selection baseline (first row) is also included.

| Facebook | PROTONET | | MAML | | FINE-TUNING | | MISTRAL | | ZEPHYR | |
|---|---|---|---|---|---|---|---|---|---|---|
| | Acc. + Std | Diff. | Acc. + Std | Diff. | Acc. + Std | Diff. | Acc. + Std | Diff. | Acc. + Std | Diff. |
| Classic | $86.74_{0.28}$ | $+0.00$ | $69.39_{0.48}$ | $+0.00$ | $68.57_{0.41}$ | $+0.00$ | $81.87_{2.14}$ | $+0.00$ | $78.17_{2.76}$ | $+0.00$ |
| Random | $87.48_{0.41}$ | $+0.74$ | $69.32_{1.23}$ | $-0.06$ | $68.48_{0.66}$ | $-0.09$ | $83.29_{0.94}$ | $+1.42$ | $78.29_{1.02}$ | $+0.12$ |
| LENS | N/A | N/A | N/A | N/A | N/A | N/A | $83.86_{0.37}$ | $+1.99$ | $79.58_{0.39}$ | $+1.41$ |
| Similarity | $85.57_{0.92}$ | $-1.16$ | $64.78_{2.56}$ | $-4.60$ | $66.94_{1.35}$ | $-1.62$ | $82.69_{1.24}$ | $+0.82$ | $78.81_{1.57}$ | $+0.64$ |
| Diversity | $84.51_{0.40}$ | $-2.23$ | $63.52_{0.54}$ | $-5.86$ | $60.19_{0.37}$ | $-8.38$ | $81.38_{0.67}$ | $-0.49$ | $77.82_{0.84}$ | $-0.35$ |
| Entropy | $86.78_{0.20}$ | $+0.05$ | $69.45_{0.23}$ | $+0.06$ | $68.60_{0.45}$ | $+0.03$ | $83.63_{0.54}$ | $+1.76$ | $79.28_{0.73}$ | $+1.11$ |
| Margin | $86.93_{0.47}$ | $+0.20$ | $69.47_{0.48}$ | $+0.08$ | $69.00_{0.32}$ | $+0.43$ | $83.44_{0.88}$ | $+1.57$ | $79.36_{0.54}$ | $+1.19$ |
| Least Confidence | $86.96_{0.46}$ | $+0.22$ | $69.60_{0.47}$ | $+0.21$ | $68.65_{0.49}$ | $+0.08$ | $82.85_{0.64}$ | $+0.98$ | $79.55_{1.56}$ | $+1.38$ |
| Loss | $84.76_{0.93}$ | $-1.97$ | $63.48_{1.06}$ | $-5.91$ | $62.10_{1.06}$ | $-6.46$ | $81.33_{1.21}$ | $-0.54$ | $76.51_{0.76}$ | $-1.66$ |
| CAL | $86.41_{0.11}$ | $-0.32$ | $69.16_{0.43}$ | $-0.23$ | $67.06_{0.25}$ | $-1.50$ | $82.93_{0.87}$ | $+1.06$ | $77.97_{0.04}$ | $-0.20$ |
| CRAIG | $87.06_{0.00}$ | $+0.32$ | $69.37_{0.00}$ | $-0.02$ | $68.61_{0.00}$ | $+0.04$ | $83.92_{0.00}$ | $+2.05$ | $76.72_{0.00}$ | $-1.45$ |
| DeepFool | $86.39_{0.00}$ | $-0.34$ | $68.27_{0.00}$ | $-1.12$ | $66.82_{0.00}$ | $-1.75$ | $83.04_{0.00}$ | $+1.17$ | $77.76_{0.00}$ | $-0.41$ |
| Forgetting | $87.42_{0.00}$ | $+0.68$ | $70.39_{0.00}$ | $+1.00$ | $69.30_{0.00}$ | $+0.73$ | $83.72_{0.00}$ | $+1.85$ | $79.52_{0.00}$ | $+1.35$ |
| Glister | $87.36_{0.00}$ | $+0.62$ | $70.00_{0.00}$ | $+0.62$ | $69.00_{0.00}$ | $+0.43$ | $82.96_{0.00}$ | $+1.09$ | $79.04_{0.00}$ | $+0.87$ |
| GraNd | $87.08_{0.00}$ | $+0.34$ | $69.40_{0.00}$ | $+0.02$ | $68.50_{0.00}$ | $-0.07$ | $82.40_{0.00}$ | $+0.53$ | $78.80_{0.00}$ | $+0.63$ |
| Herding | $86.13_{0.00}$ | $-0.60$ | $68.68_{0.00}$ | $-0.71$ | $68.67_{0.00}$ | $+0.10$ | $83.28_{0.00}$ | $+1.41$ | $78.80_{0.00}$ | $+0.63$ |
| KCenter | $85.78_{0.00}$ | $-0.95$ | $68.02_{0.00}$ | $-1.37$ | $68.48_{0.00}$ | $-0.08$ | $82.48_{0.00}$ | $+0.61$ | $78.96_{0.00}$ | $+0.79$ |
| Graph-Cut | $87.20_{0.00}$ | $+0.46$ | $69.99_{0.00}$ | $+0.60$ | $68.66_{0.00}$ | $+0.09$ | $82.08_{0.00}$ | $+0.21$ | $78.24_{0.00}$ | $+0.07$ |
| Cartography$_{Easy}$ | $86.83_{0.48}$ | $+0.10$ | $69.35_{1.21}$ | $-0.04$ | $68.78_{0.82}$ | $+0.21$ | $82.91_{0.82}$ | $+1.04$ | $78.22_{0.97}$ | $+0.06$ |
| Cartography$_{Ambiguous}$ | $86.62_{0.25}$ | $-0.11$ | $69.31_{0.42}$ | $-0.08$ | $68.47_{0.60}$ | $-0.09$ | $82.06_{1.84}$ | $+0.19$ | $77.66_{1.36}$ | $-0.51$ |
| Cartography$_{Hard}$ | $86.90_{0.70}$ | $+0.16$ | $69.54_{0.81}$ | $+0.15$ | $68.74_{0.54}$ | $+0.17$ | $84.25_{0.54}$ | $+2.38$ | $79.59_{1.19}$ | $+1.42$ |
| Cartography$_{Easy+Ambig.}$ | $87.62_{0.25}$ | $+0.89$ | $70.31_{0.42}$ | $+0.92$ | $69.18_{0.60}$ | $+0.61$ | $82.06_{1.84}$ | $+0.19$ | $77.66_{1.36}$ | $-0.51$ |
| ACSESS$_{Uniform}$ | $87.90_{0.10}$ | $+1.16$ | $70.83_{0.20}$ | $+1.44$ | $69.60_{0.38}$ | $+1.03$ | $84.42_{0.29}$ | $+2.55$ | $80.04_{0.33}$ | $+1.88$ |
| ACSESS$_{Weighted}$ | $88.01_{0.10}$ | $+1.27$ | $71.00_{0.31}$ | $+1.61$ | $69.50_{0.48}$ | $+0.93$ | $84.70_{0.36}$ | $+2.82$ | $80.13_{0.42}$ | $+1.96$ |
| ACSESS$_{With Random}$ | $87.72_{0.37}$ | $+0.99$ | $70.56_{0.58}$ | $+1.17$ | $69.32_{0.83}$ | $+0.75$ | $84.00_{1.30}$ | $+2.13$ | $79.76_{1.00}$ | $+1.59$ |

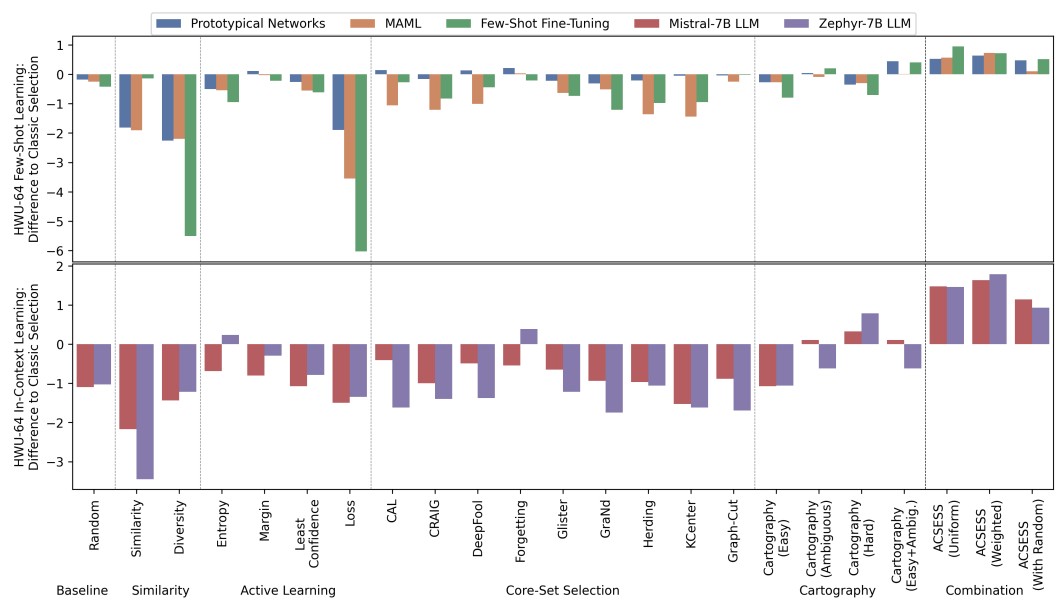

Figure 26: Benefit of the different selection strategies calculated as the difference to the classic few-shot selection strategy for the HWU-64 dataset. The performance of the classic selection is represented as the zero value.

Table 16: Benefit of the different selection strategies calculated as the difference in accuracy to the classic few-shot selection strategy for the HWU-64 dataset. The subscript represents the standard deviation in the accuracy from running the selection strategy multiple times. The difference to the classic selection baseline (first row) is also included.

| HWU-64 | PROTONET | | MAML | | FINE-TUNING | | MISTRAL | | ZEPHYR | |
|---|---|---|---|---|---|---|---|---|---|---|
| | Acc. + Std | Diff. | Acc. + Std | Diff. | Acc. + Std | Diff. | Acc. + Std | Diff. | Acc. + Std | Diff. |
| Classic | $88.61_{0.31}$ | $+0.00$ | $65.90_{0.46}$ | $+0.00$ | $66.01_{0.78}$ | $+0.00$ | $93.53_{0.68}$ | $+0.00$ | $91.62_{1.21}$ | $+0.00$ |
| Random | $88.43_{0.26}$ | $-0.18$ | $65.64_{0.26}$ | $-0.26$ | $65.59_{0.30}$ | $-0.42$ | $92.43_{0.44}$ | $-1.10$ | $90.58_{0.64}$ | $-1.03$ |
| LENS | N/A | N/A | N/A | N/A | N/A | N/A | $94.66_{0.14}$ | $+1.13$ | $92.80_{0.30}$ | $+1.18$ |
| Similarity | $86.79_{0.43}$ | $-1.82$ | $63.99_{0.62}$ | $-1.91$ | $65.87_{0.45}$ | $-0.14$ | $91.36_{0.45}$ | $-2.17$ | $88.17_{0.54}$ | $-3.45$ |
| Diversity | $86.34_{0.20}$ | $-2.26$ | $63.71_{0.27}$ | $-2.19$ | $60.50_{0.19}$ | $-5.51$ | $92.09_{0.30}$ | $-1.44$ | $90.40_{0.46}$ | $-1.22$ |
| Entropy | $88.11_{0.08}$ | $-0.50$ | $65.36_{0.30}$ | $-0.54$ | $65.06_{0.44}$ | $-0.95$ | $92.84_{0.56}$ | $-0.69$ | $91.85_{0.63}$ | $+0.24$ |
| Margin | $88.72_{0.01}$ | $+0.11$ | $65.87_{0.54}$ | $-0.03$ | $65.78_{0.10}$ | $-0.22$ | $92.73_{0.10}$ | $-0.80$ | $91.32_{0.28}$ | $-0.30$ |
| Least Confidence | $88.34_{0.23}$ | $-0.26$ | $65.35_{0.41}$ | $-0.55$ | $65.39_{0.06}$ | $-0.62$ | $92.45_{0.27}$ | $-1.08$ | $90.83_{0.10}$ | $-0.79$ |
| Loss | $86.71_{0.41}$ | $-1.90$ | $62.35_{0.05}$ | $-3.55$ | $59.98_{0.14}$ | $-6.03$ | $92.03_{0.46}$ | $-1.50$ | $90.27_{0.42}$ | $-1.35$ |
| CAL | $88.75_{0.00}$ | $+0.15$ | $64.84_{0.00}$ | $-1.06$ | $65.74_{0.00}$ | $-0.27$ | $93.12_{0.00}$ | $-0.41$ | $90.00_{0.00}$ | $-1.62$ |
| CRAIG | $88.44_{0.07}$ | $-0.16$ | $64.69_{0.04}$ | $-1.21$ | $65.18_{0.03}$ | $-0.83$ | $92.53_{0.04}$ | $-1.00$ | $90.21_{0.19}$ | $-1.40$ |
| DeepFool | $88.74_{0.00}$ | $+0.14$ | $64.89_{0.00}$ | $-1.01$ | $65.57_{0.00}$ | $-0.44$ | $93.04_{0.00}$ | $-0.49$ | $90.24_{0.00}$ | $-1.38$ |
| Forgetting | $88.82_{0.00}$ | $+0.21$ | $65.93_{0.00}$ | $+0.03$ | $65.80_{0.00}$ | $-0.21$ | $92.98_{0.00}$ | $-0.55$ | $92.00_{0.00}$ | $+0.38$ |
| Glister | $88.39_{0.00}$ | $-0.22$ | $65.26_{0.00}$ | $-0.64$ | $65.28_{0.00}$ | $-0.73$ | $92.88_{0.00}$ | $-0.65$ | $90.40_{0.00}$ | $-1.22$ |
| GraNd | $88.30_{0.18}$ | $-0.31$ | $65.39_{0.12}$ | $-0.52$ | $64.81_{0.20}$ | $-1.20$ | $92.59_{0.39}$ | $-0.94$ | $89.87_{0.25}$ | $-1.75$ |
| Herding | $88.40_{0.00}$ | $-0.21$ | $64.54_{0.00}$ | $-1.36$ | $65.03_{0.00}$ | $-0.98$ | $92.56_{0.00}$ | $-0.97$ | $90.56_{0.00}$ | $-1.06$ |
| KCenter | $88.55_{0.00}$ | $-0.05$ | $64.46_{0.00}$ | $-1.44$ | $65.07_{0.00}$ | $-0.94$ | $92.00_{0.00}$ | $-1.53$ | $90.00_{0.00}$ | $-1.62$ |
| Graph-Cut | $88.57_{0.00}$ | $-0.04$ | $65.65_{0.00}$ | $-0.25$ | $65.99_{0.00}$ | $-0.02$ | $92.64_{0.00}$ | $-0.89$ | $89.92_{0.00}$ | $-1.70$ |
| Cartography$_{Easy}$ | $88.34_{0.26}$ | $-0.27$ | $65.63_{0.37}$ | $-0.27$ | $65.22_{0.44}$ | $-0.80$ | $92.46_{0.38}$ | $-1.07$ | $90.56_{0.42}$ | $-1.06$ |
| Cartography$_{Ambiguous}$ | $88.65_{0.23}$ | $+0.04$ | $65.81_{0.45}$ | $-0.09$ | $66.21_{0.55}$ | $+0.20$ | $93.63_{1.00}$ | $+0.10$ | $90.99_{0.94}$ | $-0.62$ |
| Cartography$_{Hard}$ | $88.25_{0.32}$ | $-0.35$ | $65.60_{0.39}$ | $-0.30$ | $65.30_{0.52}$ | $-0.71$ | $93.85_{0.49}$ | $+0.32$ | $92.40_{0.43}$ | $+0.78$ |
| Cartography$_{Easy+Ambig.}$ | $89.05_{0.23}$ | $+0.44$ | $65.91_{0.45}$ | $+0.01$ | $66.41_{0.55}$ | $+0.40$ | $93.63_{1.00}$ | $+0.10$ | $90.99_{0.94}$ | $-0.62$ |
| ACSESS$_{Uniform}$ | $89.13_{0.10}$ | $+0.53$ | $66.47_{0.14}$ | $+0.57$ | $66.96_{0.26}$ | $+0.95$ | $95.00_{0.07}$ | $+1.48$ | $93.08_{0.26}$ | $+1.46$ |
| ACSESS$_{Weighted}$ | $89.25_{0.13}$ | $+0.64$ | $66.62_{0.19}$ | $+0.72$ | $66.73_{0.37}$ | $+0.72$ | $95.16_{0.16}$ | $+1.63$ | $93.40_{0.32}$ | $+1.79$ |
| ACSESS$_{With Random}$ | $89.09_{0.26}$ | $+0.48$ | $66.01_{0.31}$ | $+0.11$ | $66.52_{0.73}$ | $+0.51$ | $94.67_{0.85}$ | $+1.14$ | $92.55_{0.88}$ | $+0.93$ |

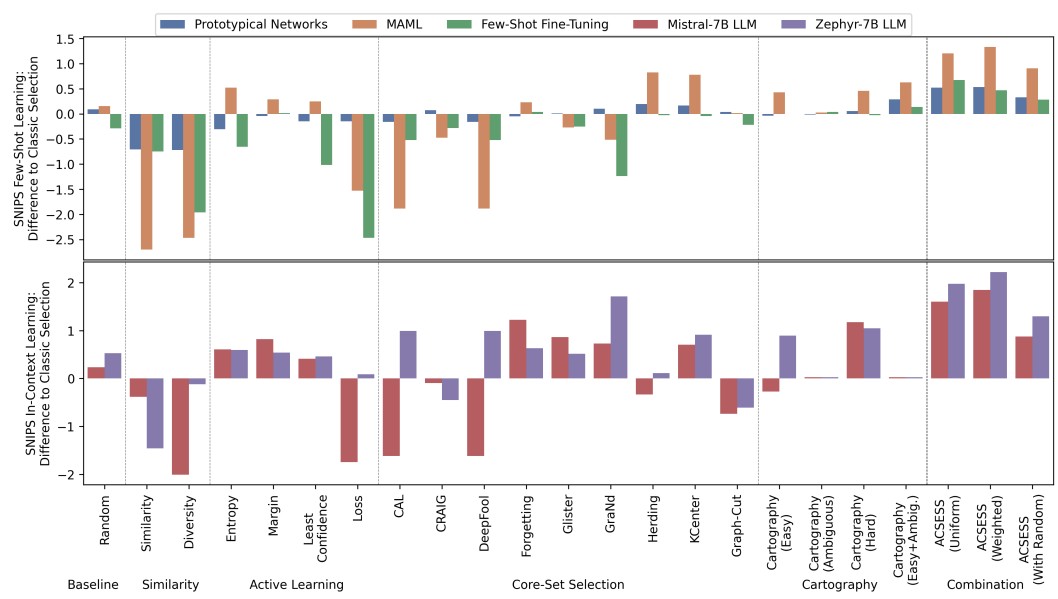

Figure 27: Benefit of the different selection strategies calculated as the difference to the classic few-shot selection strategy for the SNIPS dataset. The performance of the classic selection is represented as the zero value.

Table 17: Benefit of the different selection strategies calculated as the difference in accuracy to the classic few-shot selection strategy for the SNIPS dataset. The subscript represents the standard deviation in the accuracy from running the selection strategy multiple times. The difference to the classic selection baseline (first row) is also included.

| SNIPS | PROTONET | | MAML | | FINE-TUNING | | MISTRAL | | ZEPHYR | |
|---|---|---|---|---|---|---|---|---|---|---|
| | Acc. + Std | Diff. | Acc. + Std | Diff. | Acc. + Std | Diff. | Acc. + Std | Diff. | Acc. + Std | Diff. |
| Classic | $96.98_{0.09}$ | +0.00 | $84.10_{0.24}$ | +0.00 | $89.86_{0.26}$ | +0.00 | $92.02_{1.23}$ | +0.00 | $85.65_{1.16}$ | +0.00 |
| Random | $97.07_{0.11}$ | +0.09 | $84.26_{1.50}$ | +0.16 | $89.58_{0.46}$ | −0.29 | $92.25_{0.52}$ | +0.23 | $86.18_{1.39}$ | +0.53 |
| LENS | N/A | N/A | N/A | N/A | N/A | N/A | $93.16_{0.41}$ | +1.14 | $87.06_{0.66}$ | +1.41 |
| Similarity | $96.27_{0.61}$ | −0.70 | $81.40_{2.64}$ | −2.70 | $89.12_{1.15}$ | −0.75 | $91.63_{1.38}$ | −0.38 | $84.19_{2.06}$ | −1.46 |
| Diversity | $96.26_{0.17}$ | −0.72 | $81.64_{0.57}$ | −2.46 | $87.90_{0.16}$ | −1.96 | $90.01_{0.85}$ | −2.01 | $85.53_{0.78}$ | −0.12 |
| Entropy | $96.67_{0.03}$ | −0.30 | $84.63_{0.46}$ | +0.52 | $89.21_{0.87}$ | −0.65 | $92.62_{0.62}$ | +0.60 | $86.24_{0.73}$ | +0.59 |
| Margin | $96.93_{0.19}$ | −0.04 | $84.40_{0.41}$ | +0.29 | $89.88_{0.45}$ | +0.01 | $92.83_{0.83}$ | +0.82 | $86.19_{0.21}$ | +0.54 |
| Least Confidence | $96.83_{0.07}$ | −0.14 | $84.35_{0.40}$ | +0.25 | $88.85_{0.90}$ | −1.01 | $92.43_{0.58}$ | +0.41 | $86.11_{0.36}$ | +0.46 |
| Loss | $96.83_{0.27}$ | −0.15 | $82.57_{1.03}$ | −1.53 | $87.40_{0.29}$ | −2.46 | $90.27_{0.89}$ | −1.75 | $85.73_{1.16}$ | +0.08 |
| CAL | $96.82_{0.00}$ | −0.16 | $82.22_{0.00}$ | −1.88 | $89.35_{0.00}$ | −0.52 | $90.40_{0.00}$ | −1.62 | $86.64_{0.00}$ | +0.99 |
| CRAIG | $97.05_{0.00}$ | +0.08 | $83.63_{0.00}$ | −0.47 | $89.58_{0.00}$ | −0.28 | $91.92_{0.00}$ | −0.10 | $85.20_{0.00}$ | −0.45 |
| DeepFool | $96.82_{0.00}$ | −0.16 | $82.22_{0.00}$ | −1.88 | $89.35_{0.00}$ | −0.52 | $90.40_{0.00}$ | −1.62 | $86.64_{0.00}$ | +0.99 |
| Forgetting | $96.93_{0.00}$ | −0.05 | $84.34_{0.00}$ | +0.23 | $89.91_{0.00}$ | +0.04 | $93.24_{0.00}$ | +1.22 | $86.28_{0.00}$ | +0.63 |
| Glister | $96.98_{0.00}$ | +0.01 | $83.83_{0.00}$ | −0.27 | $89.61_{0.00}$ | −0.25 | $92.88_{0.00}$ | +0.86 | $86.16_{0.00}$ | +0.51 |
| GraNd | $97.08_{0.01}$ | +0.10 | $83.59_{0.02}$ | −0.52 | $88.63_{0.01}$ | −1.24 | $92.75_{0.04}$ | +0.73 | $87.36_{0.34}$ | +1.71 |
| Herding | $97.17_{0.00}$ | +0.20 | $84.93_{0.00}$ | +0.83 | $89.84_{0.00}$ | −0.02 | $91.68_{0.00}$ | −0.34 | $85.76_{0.00}$ | +0.11 |
| KCenter | $97.14_{0.00}$ | +0.17 | $84.88_{0.00}$ | +0.78 | $89.82_{0.00}$ | −0.04 | $92.72_{0.00}$ | +0.70 | $86.56_{0.00}$ | +0.91 |
| Graph-Cut | $97.01_{0.00}$ | +0.04 | $84.12_{0.00}$ | +0.02 | $89.65_{0.00}$ | −0.22 | $91.28_{0.00}$ | −0.74 | $85.04_{0.00}$ | −0.61 |
| Cartography$_{Easy}$ | $96.94_{0.21}$ | −0.04 | $84.54_{0.95}$ | +0.43 | $89.86_{0.54}$ | −0.00 | $91.74_{1.56}$ | −0.27 | $86.54_{1.21}$ | +0.90 |
| Cartography$_{Ambiguous}$ | $96.96_{0.10}$ | −0.01 | $84.13_{0.25}$ | +0.03 | $89.91_{0.20}$ | +0.04 | $92.04_{1.02}$ | +0.02 | $85.67_{1.51}$ | +0.02 |
| Cartography$_{Hard}$ | $97.04_{0.18}$ | +0.06 | $84.56_{0.64}$ | +0.46 | $89.84_{0.55}$ | −0.02 | $93.19_{1.01}$ | +1.18 | $86.70_{1.29}$ | +1.05 |
| Cartography$_{Easy+Ambig.}$ | $97.26_{0.10}$ | +0.29 | $84.73_{0.25}$ | +0.63 | $90.01_{0.20}$ | +0.14 | $92.04_{1.02}$ | +0.02 | $85.67_{1.51}$ | +0.02 |
| ACSESS$_{Uniform}$ | $97.50_{0.02}$ | +0.52 | $85.31_{0.22}$ | +1.20 | $90.54_{0.08}$ | +0.67 | $93.62_{0.36}$ | +1.60 | $87.62_{0.58}$ | +1.98 |
| ACSESS$_{Weighted}$ | $97.51_{0.09}$ | +0.53 | $85.44_{0.27}$ | +1.33 | $90.34_{0.13}$ | +0.47 | $93.86_{0.42}$ | +1.85 | $87.87_{0.68}$ | +2.22 |
| ACSESS$_{With Random}$ | $97.30_{0.27}$ | +0.33 | $85.01_{0.59}$ | +0.90 | $90.15_{0.39}$ | +0.29 | $92.89_{1.02}$ | +0.88 | $86.94_{2.25}$ | +1.30 |

