# OpenReview forum: "Automatic Combination of Sample Selection Strategies for Few-Shot Learning"
_ICLR.cc/2025/Conference — Submitted to ICLR 2025_

### Official Review · Reviewer_tzJ3 · 2024-10-30

**Soundness:** 3
**Presentation:** 3
**Contribution:** 3
**Rating:** 6
**Confidence:** 4

**Summary:**

This thesis focuses on proposing a method that identifies a subset of relevant strategies that can improve the overall success rate of few-shot learning. The identified strategies are then weighted according to their expected contribution, and the identified strategies are combined to identify the most informative, highest quality samples that provide the greatest benefit.

**Strengths:**

The conclusions obtained through a large number of experimental validations provide a valuable reference for research teams with limited resources, which can effectively reduce the cost of trial and error and accelerate the research process.

**Weaknesses:**

1. What model is used for the backbone of this article (e.g., whether it is a CNN or a Transformer), and the characteristics of the different architectures can have an impact on the model performance. Most classical methods are usually proposed without pre-training, but if pre-trained models are used, it is necessary to verify whether the differences between the pre-trained and target tasks affect the applicability of the conclusions of this paper. It is recommended that the authors perform specific backbone architecture comparisons, such as testing on ResNet and Vision Transformer, and perform ablation experiments, using pre-trained and randomly initialized models, to directly assess the impact of pre-training on the findings.

2. In few-shot learning tasks, the domain gap exists in data distribution between an upstream task (pre-training task) and a downstream task (fine-tuning or application task). This may result in the knowledge learned by the model in the upstream task not being effectively migrated to the downstream task. In this case, methods need to be evaluated on cross-domain undersampled learning benchmark datasets such as DomainNet or Office-Home. Testing on these datasets enables a more specific assessment of the impact of domain gaps on the performance of its sample selection strategy and a clearer understanding of the applicability of its strategy to cross-domain few-shot learning.

**Questions:**

Please see the weaknesses.

---

> ### Author Response · Authors · 2024-11-20
>
> **Thank you for your review, and highlighting the paper’s strengths**, especially the **usefulness of the research** to other researcher teams working with limited resources.
>
> Below, we address the raised weaknesses and the clarifying questions.
>
> > W1: What model is used for the backbone of this article…ablation experiments, using pre-trained and randomly initialized models
>
> **The information about the backbone models, training details and how they are used is provided in Section 4 and (for the most part) in Appendix E.**
>
> In short, we use ResNet-18 for image data and BERT for text data (and Mistral-7B and Zephyr-7B for ICL).
>
> To keep the setting as similar as possible across the modalities, we use pre-trained models in all cases – ResNet-18 pre-trained on Imagenet and pre-trained BERT, Mistral and Zephyr models. The models (except for the ICL ones) are further trained on the training split of each dataset and then evaluated on the unseen classes from the test set (where we also apply the sample selection).
>
> We opted to use the pre-trained models as training them from scratch would not be feasible and lead to significantly lower performance. As the focus of the paper is on the benefit of sample selection for few-shot learning (and adaptation to unseen classes) we do not believe that comparing pre-trained and non-pre-trained models would provide any meaningful contribution to our findings. Especially as **there is already a lot of research that investigates the impact of pre-training on few-shot learning models.**
>
> > W2: Evaluation on cross-domain setting
>
> Thank you for the suggestion. However, as discussed in Section 1 and Limitations (Appendix B), the **scope of the paper is on the within-domain few-shot learning** – i.e., looking at the performance benefit when adapting to unseen classes from the same dataset. **Focusing on the cross-domain setting would introduce an additional compounding factor** that would need to be considered and worked with, and would potentially make the interpretation of the results more complicated.
>
> However, we **agree that evaluating the impact of sample selection in a cross-domain setting could be interesting** (and, as discussed in Limitations in Appendix B, may lead to higher benefits of sample selection). We **publicly release the code** of our experiments (using an anonymised link in a footnote on the second page) to allow other researchers to extend our work to settings, tasks and approaches we do not consider in the paper.
>
> **We hope this answers all your clarifying questions and the raised concerns. If there are any further questions, we will be glad to continue the discussion. Thank you very much.**

---

### Official Review · Reviewer_2vH8 · 2024-11-01

**Soundness:** 3
**Presentation:** 2
**Contribution:** 4
**Rating:** 5
**Confidence:** 4

**Summary:**

Samples are important in few-shot learning (FSL), meta-learning and in-context learning (ICL), especially in noisy environment. Sample selection picks informative, easily-learned or representative examples, probably improve performance of FSL or meta-learning or ICL when compared to typical supervised settings. However, the impact of sample selection is not well-studied in few-shot scenario.

This work managed to justify the impact of different sample selection strategies. It showcases that the effectiveness of each sample selection strategies depends on data modality, dataset characteristics and models. To facilitate sample selection for FSL, in this work authors propose a method to identify a subset of well-performing strategies and combine those chosen ones for learning.

Three types of  sample selection strategy identification are investigated, including forward selection, backward selection and "datamodels" selection. The obtained three subsets of sample selection strategies are intersected to get the final set of strategies. Then, a weighting scheme is utilized to assign different weight to each strategy for sample selection.

With the proposed sample selection method by automatically combining different sample selection strategies, this work witnesses promising results on different tasks.

**Strengths:**

The work is well-motivated, the literature is well reviewed, and a reasonable and promising method for combing different sample selection strategies for learning is proposed.

The idea of learning a regression model to derive weights for each strategy, by learning upon a small number of random combinations of strategies, is interesting.

**Weaknesses:**

My concerns are as follows:
1) **Novelty**: The work uses off-the-shelf sample selection strategies, proposing a strategy to leverage these sample selection strategies for model learning. The integration of existing strategies into a unified pipeline easily distracts my attention away from its main contribution (Note I know it is the automatic combination of sample selection strategies).
2) **Clarity**: Though I know the overall framework of the proposed ACSESS, it remains ambiguous for me how sample selection works within FSL, for instance in the classic 5w-5s few-shot image classification. And, it is unclear how to efficiently measure the contribution of each strategy, especially when several strategies are combined. How exactly does the sample selection operate in the meta-learning procedure? How do we generalize or use ACSESS in a new task from a new dataset?
3) **Efficiency**: As said, the optimal combination of sample selection strategies depends on data modality, dataset characteristic (eg. size, #categories) and model. It means we have no idea of which sample selection is better for a new dataset, model, nor do we know which combination of sample selection strategies boost the performance most unless searching for the optimal one. This is computational inefficient.  How can we intuitively and quickly identify a subset of sample selection strategies when encountering a new dataset or model?

**Questions:**

The studied topic is useful and meaningful. But the details can hardly be elaborated in limited pages. Illustrative examples are missing which impedes understanding. I think this work would be a great fit for journals like TPAMI after meticulous reorganization.

For questions, plz refer to the weaknesses.

---

> ### Author Response · Authors · 2024-11-20
>
> **Thank you for your comprehensive and detailed review, and highlighting the paper’s strengths**, especially the **contribution** it brings and the motivation for performing sample selection in FSL setting.
>
> Below we address the raised concerns and answer the questions.
>
> > W1: Novelty
>
> Although the proposed ACSESS strategy uses a combination of existing selection strategies, it **allows us to draw novel insights and findings regarding the benefit of sample selection in a few-shot setting.**
>
> One such finding is **identifying that the learnability of samples plays a critical role** in the success of few-shot learning, and thus identifying a subset of strategies that consistently work well.
>
> Another significant finding is that the **sample selection provides benefit even for the FSL setting** (where we already work with small sample sizes) and that the combination of the existing strategies enhances the overall effectiveness of sample selection and outperforms the existing state-of-the-art solutions, especially in imbalanced datasets with noisy samples – thanks to identifying samples with complementary properties.
>
> Furthermore, we observe that sample selection has a **higher impact on a lower number of shots**, while the boost of using more samples is negligible after a certain point.
>
> As such, we **believe the paper provides sufficient novelty through these findings** (which are not observed in previous works), showing that a simple combination of existing, well-known sample selection strategies is enough to provide significant benefit (that even outperforms other, more complicated state-of-the-art methods) and allow us to draw novel insights into sample selection in few-shot learning.
>
> > W2: Clarity
>
> To **improve the clarity and allow for better understanding**, we submitted **a revised version of the paper in which we provide an Algorithm** – the steps of the ACSESS strategy are compiled in Algorithm 1 in Section 3.2, and further low-level details of the forward, backward and datamodels selection are in Appendix C.
>
> **The details of how the sample selection works and of the experimental setup are already provided in Appendix D and E.**
>
> In short, the ACSESS method is designed to be independent of the setup – therefore, there is no difference to the typical 5w-5s few-shot setting.
>
> We split the dataset into training classes and unseen classes. The approaches are then trained using all samples from the training classes. Afterwards, we apply the sample selection to select a set of high-quality samples from the unseen classes and use these to adapt the trained model to the new task and evaluate the performance.
>
> The performance and efficiency of the individual strategies are determined in the same setup (using validation samples/classes) by evaluating the performance when using the set of selected samples for adaptation to previously unseen classes. The contribution of individual strategies from the combination can be determined mainly by an ablation study (similar to the backward selection) – observing how the performance benefit changes when the strategy is removed from the combination.
>
> Overall, the **ACSESS strategy is applied only to the new task/classes** (in our in-domain setting) or new dataset/domain in an out-of-domain setting (which we do not consider as discussed in Limitations in Appendix B).
>
> > W3: Efficiency
>
> As discussed in Section 4.2 and Appendix C, the highest performance benefit can indeed be achieved by using the strategy in a model/dataset/modality-dependent setting. However, one of our **contributions is exploring the benefit on a more aggregated level** (e.g., across multiple datasets and models) and **identifying which selection strategies lead to consistently high benefits.**
>
> Based on our results, we provide a **discussion in Section 4.2 and Appendix C** (part “Choosing the best configuration for the ACSESS method) on quickly identifying a well-performing subset of strategies and their combination. We observed that a **uniform combination of the Cartography and Margin selection strategies provides good enough and consistent performance benefits.**
>
> **We hope this answers all your clarifying questions and the raised concerns. If there are any further questions, we will be glad to continue the discussion. Thank you very much.**

---

> > ### Comment · Reviewer_2vH8 · 2024-11-22
> >
> > Thank you for the discussion. However, I do think that my concerns are not properly addressed yet.
> >
> > >  Novelty
> >
> > The novelty is limited as there is a tacit understanding  in machine learning that high-quality samples benefit performance.  This work contributes a complex strategy but upon several existing sample selection strategies for the purpose of sample selection.
> >
> > Agarwal etal [1] studied the sensitivity of FSL methods to support data which identified the importance of support samples. They also showcased that different methods suffered differently on different datasets in different shots.
> >
> > The paper complements to [1] by leveraging sample selection to improve performance. I refer authors to [1] for more details.
> >
> > [1] Agarwal, Mayank, Mikhail Yurochkin, and Yuekai Sun. "On sensitivity of meta-learning to support data." Advances in Neural Information Processing Systems 34 (2021): 20447-20460.
> >
> > > Clarity
> >
> > I understand the sample selection illustrated in Alg-1 clearly.
> > I have thought that the sample selection gave each sample a score in the dataset and prefers to select those with higher scores for model training, ie pre-training and meta-training. However, authors explain that they utilize the sample selection pipeline in evaluation to "select a set of high-quality samples from the unseen classes and use these to adapt the trained model to the new task ". A new question is, if the new task has so many examples, what if we directly use all the samples for adaptation, say we use 5-way 20-shot (rather than 5-way 5-shot after sample selection)? If we do have so many samples in a new task, I would like to use them all. According to previous work in the literature, 20 samples can greatly improve FSL performance over 5- or 10-shot settings.
> >
> > A sample selection strategy that applies in the pre-training and meta-training stage is much more useful than one in the evaluation stage, because we donot have that many samples to cherry-pick the best one.
> >
> > > Efficiency
> >
> > As "the contribution of individual strategies from the combination can be determined mainly by an ablation study",
> > the propose sample selection has to estimate the importance of each sample to the performance of the specific model. This could be done efficiently in evaluation due to small number of samples in each novel category.
> >
> > However, as discussed above, I donot think it is a good idea to sample a subset of samples from limited support data. And, if the number of support data is large enough so that we have to do sample selection, this is not a FSL problem but a date efficiency problem as in the conventional supervised learning.
> >
> > Hope the discussion help.

---

> > > ### Author Response · Authors · 2024-11-22
> > >
> > > Thank you for your answer and the additional discussion points
> > >
> > > > Novelty
> > >
> > > We are aware of the work by Agarwal et al [1], as it is **one of a few papers that actually deal with the sensitivity of meta-learning** to the support data (same as we do) and is **one of the papers that motivated us to perform this study.** The authors have indeed shown the importance of support samples, by focusing on the difference between worst and best-performing samples – finding that the bad-performing samples are not artefacts and/or noisy. However, the authors also notice that to achieve good robustness **it is important to have well-separated class embeddings, but this cannot be achieved by adversarial training** (as their results suggest).
> > >
> > > We indeed **complement the work** by Agarwal et al, by **exploring which sample characteristics are important for achieving good few-shot performance** – finding that it can be consistently achieved by employing a combination of Margin and Cartography sample selection, which further reinforces the findings of [1] (i.e., easy to learn samples on the decision boundary are the best).
> > >
> > > Following your comment, we have **realised that this work was unintentionally removed from the paper during its preparation – we have just submitted another revised version of the paper where we cite this work again** (in the Introduction and Related Work sections). Thank you very much for pointing out this mistake.
> > >
> > >
> > > > Clarity
> > >
> > > Although we evaluate the proposed ACSESS strategy by selecting the adaptation data (i.e., the support samples from unseen classes that represent a “new task” for us), **there is nothing preventing it from being used for selecting the support data in the pre-training and meta-training** phases. However, we opted to explore **only the adaptation phase** to keep the setting across all approaches we use consistent, as well as because we consider it to be the part that is **more affected by the sensitivity** (based on our analysis of related work).
> > >
> > > > A new question is, if the new task has so many examples, what if we directly use all the samples for adaptation, say we use 5-way 20-shot (rather than 5-way 5-shot after sample selection)?
> > >
> > > We **already explore this question in the ablation studies provided in Section 4.3** (with detailed results in Appendix H and I) – we increase the number of selected shots and observe the change in performance. As noted in the findings (also listed in the Introduction), increasing the number of shots does help, but only to a certain point after which the performance plateaus or even decreases.
> > >
> > > As such, we **find that subsampling even the limited support set we have can often lead to higher performance than using all of its samples** (especially when there are noisy support samples).
> > >
> > > > Efficiency; This could be done efficiently in evaluation due to small number of samples in each novel category.
> > >
> > > We **discuss this exact thing in Appendix C.** Similarly, as discussed above, we showed in our ablation studies (Section 4.3, Appendix H and I) that subsampling even a limited support set (or selecting from a smaller dataset subset) can often lead to improved performance on the new task. Therefore, we have indeed shown that **it makes sense to sample a subset of samples even from limited support data.**
> > >
> > >
> > > We hope this clarifies the paper further and addresses your concerns. If there are any additional points, we are happy to discuss them further. Thank you very much.
> > >
> > > References:
> > > 1. Agarwal, Mayank, Mikhail Yurochkin, and Yuekai Sun. "On sensitivity of meta-learning to support data." Advances in Neural Information Processing Systems 34 (2021): 20447-20460.

---

### Official Review · Reviewer_UDbF · 2024-11-02

**Soundness:** 3
**Presentation:** 3
**Contribution:** 3
**Rating:** 6
**Confidence:** 5

**Summary:**

The article addresses the uncertainty of the impact of sample selection on few-shot learning by:
(1) Broadly investigating the effects of single-property sample selection strategies used in previous work across three approaches: image classification and text classification in few-shot learning, and context-based text classification on extensive datasets.
(2) Proposing Automatic Combination of Sample Selection Strategies (ACSESS) method, which combines multiple complementary attributes in sample selection. The method employs the intersection of Forward Selection, Backward Selection, and Datamodels Selection schemes, assigning scores to samples through a weighted approach. Experimental results show that ACSESS has a sustained impact on performance, indicating that learnability is the highest-priority metric in few-shot learning.
(3) Further exploring the effects of sample quantity, sample quality (including noisy samples), dataset size, and sample selection strategies.
Overall, this paper provides a comprehensive analysis of the influence of sample selection strategies on few-shot learning and creatively introduces the ACSESS method, a strategic combination to overcome the limitations of single-attribute selection. The paper is supported by extensive experimental data and conclusive findings, making it a thorough and innovative contribution to sample selection strategies.

**Strengths:**

1. The article systematically evaluates the performance of single-property sample selection strategies and introduces a novel sample selection strategy—ACSESS. This strategy aims to combine multiple single-property strategies, optimizing the selected samples through a balanced weighting scheme to overcome the limitations of single-attribute selection, significantly enhancing the performance of few-shot learning under extreme sample conditions.
2. The paper provides extensive experimental data, covering nearly all existing sample selection strategies, and evaluates their effectiveness in few-shot learning tasks across text classification, image classification, and context-based text classification.
3. The ACSESS method innovatively proposes a scheme for autonomously combining strategy sets, building on previous research and addressing prior strategy limitations, with potential to advance future few-shot learning studies.
4. The article’s structure is clear, with numerous bolded sections to enhance readability and summary for researchers. However, the frequent use of bolding may sometimes detract from the clarity of key points, and it would be beneficial to use bolding more selectively.

**Weaknesses:**

1.Considering that the paper finalizes the relevant selection strategies by taking the intersection of three selection methods (Forward Selection, Backward Selection, Datamodels Selection), does this combined approach demonstrate a performance advantage over using single-strategy methods?
2.The paper evaluates three independent weighting schemes. Are there other feasible weighting schemes beyond these three? Alternatively, could an adaptive mechanism be introduced to adjust these weights based on the characteristics of the dataset? Further explanation on this aspect would aid in understanding the flexibility and effectiveness of the ACSESS approach.
3.While ACSESS demonstrates excellent performance in text and image classification tasks, could this method also be applied to other few-shot learning scenarios, such as more challenging tasks like object detection, semantic segmentation, or depth estimation?

**Questions:**

see the weaknesses

---

> ### Author Response · Authors · 2024-11-20
>
> **Thank you for your comprehensive and detailed review, and for highlighting the papers’s strengths**, especially the **extensive evaluation, thorough and innovative contributions and good readability.**
>
> Below we address the individual questions that were raised as part of the weaknesses.
>
> > Q1/W1 Advantage of ACSESS over using only a single selection method
>
> **Yes, we would like to confirm that the combined approach indeed demonstrated a performance advantage over using single-strategy methods**. In our preliminary experiments, we observed that the individual selection methods (forward, backward, datamodels) often selected strategies that, in combination with other strategies, could lead to lower performance than those obtained when using the ACSESS method. **We believe this is mainly due to the interactions between the individual selection strategies and how the different selection methods can handle the interactions** – for example, the forward method often selects as the first strategy the best performing one from the list (as presented in the results in Section 4), but the combination of this strategy with others often led to inferior performance.
>
> **To a certain extent, this can be observed with the $ACSESS_{WithRandom}$ in Section 4** (or the results in Appendix L), as the random selection was one of the strategies that could have been selected, but its combination with other strategies leads to lower performance.
>
> Using only the single-selection methods with ProtoNets on LR_AM.DOG dataset as an example, we observe:
> - Forward selects Cartography, Margin and Glister, where this combination leads to a performance of 70.91
> - Backward selects Cartography, Margin, GraNd, CRAIG, Forgetting and Glister and leads to a performance of 70.97
> - Datamodels selects Cartography, Margin, KCenter and Random and leads to a performacne of 71.01
> - Combination of Forward and Backward results in the same setting as Forward
> - Combination of all (or any of the methods with Datamodels) leads to Cartography and Margin with performance of 71.04
>
> As such, there are small differences in performance, but also significant differences in a number of sample selection strategies that are used.
>
> > Q2/W2 Other possible weighting schemes
>
> As we **designed the method to be as flexible as possible, any weighting scheme can be used.** However, the ones we explore in the paper should be the most straightforward while providing good performance.
>
> **The $ACSESS_{Weighted}$ can already be considered as the adaptive weighting scheme** that can adjust the weights based on the characteristics of the dataset – as **discussed in Section 3 and Appendix C,** creating a separate set of weights for each dataset leads to the highest increase in performance but makes the strategy dataset dependent. However, one of the goals of our work was to provide more general findings, so we opted to reduce the model/dataset dependence as much as possible. However, the method is designed to allow for working on a per-model/per-dataset basis.
>
> > Q3/W3 ACSESS on other tasks
>
> **Yes, the ACSESS can be applied to other few-shot learning scenarios.** The method is designed to be flexible, so we do not see anything that would prevent the method from being applied to these scenarios – any task where we can determine the performance of the model for sample selection (accuracy/F1/MSE or other metric) and where meta-learning of few-shot learning approaches can be applied (which may include object detection, segmentation or depth estimation, although we are not that well versed in the specifics of these domains).
>
> **We hope this answers all your clarifying questions raised as part of the weaknesses. If there are any further questions, we will be glad to continue the discussion. Thank you very much.**

---

### Official Review · Reviewer_hLgQ · 2024-11-04

**Soundness:** 2
**Presentation:** 3
**Contribution:** 2
**Rating:** 5
**Confidence:** 4

**Summary:**

This paper introduces ACSESS, an Automatic Combination of Sample Selection Strategies aimed at enhancing few-shot learning by combining diverse sample selection strategies. The paper evaluates 20 sample selection strategies across multiple few-shot learning frameworks, including meta-learning and in-context learning, using datasets from both image and text domains. ACSESS leverages several selection properties (informativeness, representativeness, and learnability) and proposes a scoring mechanism that improves the quality of selected samples by assigning weights to different strategies. This approach demonstrates consistent performance gains, especially in settings with imbalanced or noisy datasets.

**Strengths:**

1.	This paper presents an extensive comparison of 20 sample selection strategies, highlighting each strategy’s strengths and limitations across different few-shot learning contexts. This analysis supports the validation of ACSESS’s effectiveness over traditional single-property strategies.
2.	The approach incorporates a diverse set of datasets, spanning both image and text domains, and evaluates multiple few-shot learning methods, including meta-learning and in-context learning.

**Weaknesses:**

1. **Unclear differentiation from existing mechanisms**: The proposed forward and backward selection mechanisms lack a clear distinction from existing methods. The paper does not sufficiently emphasize how these mechanisms differ from conventional techniques or why they are particularly well-suited to the ACSESS framework. The paper could benefit from providing clear formulas or pseudocode to highlight the differences from recent related work (e.g., L149, L151), which would help readers better understand the distinction.

2. **Lack of detailed methodological steps**: Although the approach resembles an ensemble method that combines multiple strategies, the methodology section does not clearly outline the specific steps required to implement this combination. This lack of detail may challenge readers attempting to reproduce or apply the method. For example, providing clear algorithmic steps or expressing them through formulas would help avoid ambiguities and facilitate verification of the method on recent related works. Additionally, if relevant experiments are available, presenting them would enhance the manuscript’s credibility.

3. **Limited explanation of integration across strategies and datasets**: While the paper examines the impact of combining various strategies across datasets, it does not provide a structured explanation of how these combinations are optimized. A clearer flowchart for integrating diverse strategies with datasets would enhance the presentation. The manuscript could provide a specific example to illustrate the clear operational flowchart and pseudocode, further clarifying the execution process of ACSESS.

4. **Insufficient theoretical support for strategy selection**: Although extensive experiments empirically validate the method, the paper lacks a theoretical basis for favoring certain strategies. Greater theoretical justification could strengthen the feasibility and applicability of ACSESS, as relying solely on experimental results may limit the method’s perceived robustness.

**Questions:**

1. How does the proposed forward and backward selection in ACSESS differ fundamentally from existing selection mechanisms, and what distinct advantages does it bring?
2. Could the paper provide a clearer step-by-step explanation of how different strategies are combined within ACSESS?
3. What specific methods or frameworks were considered to determine the optimal combination of strategies and datasets, and how was this integration logically structured?
4. What theoretical considerations underpin the strategy selection in ACSESS, and how might the absence of theoretical support affect the method’s generalizability across diverse datasets and tasks?

---

> ### Author Response · Authors · 2024-11-20
>
> **Thank you for your review, and highlighting the extensive comparison and diverse set of datasets and few-shot learning methods.**
>
> To **address the majority of the raised concerns** (W1, W2, W3), we submitted **a revised version of the paper in which we provide an Algorithm that summarises the steps of the ACSESS strategy** (Algorithm 1 in Section 3.2 and further low-level details of the forward, backward and datamodels selection in Appendix C).
>
> Below, we address the individual weaknesses (W) and questions (Q) not addressed by this algorithm.
>
> > Q1: difference of ACSESS to related work
>
> The related work (from lines 149-150) introduces a two-step search process for identifying samples with specific properties and defines an objective to achieve this. On the other hand, we use the existing strategies and provide an automatic selection of sample selection strategies. As such, this **makes our method more flexible**, even on datasets and settings where different properties are more important.
>
> > Q3: combination of strategies
>
> As discussed in Section 3.2, we utilise 3 methods to identify the best-performing sample selection strategies in the few-shot learning setting. The methods all use a validation set to determine the overall performance when using the set of selected samples.
> Afterwards, we find the best-performing combination as an intersection between the strategies identified from each method.
>
> As already mentioned, the steps are now compiled in Algorithm 1 in Section 3.2, and further low-level details of the forward, backward and datamodels selection are in Appendix C.
>
> > W4, Q4: theoretical support for why certain strategies are favoured
>
> By default, the **ACSESS strategy is not designed to favour any specific strategy but instead to identify the samples with properties that provide the most benefit for the given setting.** To achieve this and identify the properties, we use the different objectives as defined by the sample selection strategies that we consider in the combination – as described in Table 1 in Section 3.2 and Appendix D. As such, there is no theoretical consideration that would underpin the strategy and affect its generalisability.
>
> In addition, we **provide further analysis of why we believe that the learnability property is the preferred property in few-shot learning – included in Section 4.2.** On all of the datasets and models, the Cartography and Margin strategies are always part of the combination. As such, for few-shot learning, the samples that are easy to learn and are located on the decision boundary (and so provide the most information in this setting) provide the most benefit.
>
> **We hope this addresses all your concerns and questions. If there are any further questions, we will be glad to continue the discussion. Thank you very much.**

---

> > ### Comment · Reviewer_hLgQ · 2024-11-29
> > **Official Comment by Reviewer hLgQ**
> >
> > Thanks for the authors' response. However, I agree with Reviewer aHzt and think the novelty and contributions of this work are incremental. Therefore, I will maintain my original score.

---

> > > ### Author Response · Authors · 2024-11-29
> > >
> > > Thank you for your answer.
> > >
> > > Please, **could you be more specific regarding the novelty and which contributions are only incremental, by pointing us to relevant existing works?** Analysing and incorporating these existing studies would surely help us to improve the paper and allow us to **address this concern more precisely** (e.g., clarifying the research gap we are addressing).
> > >
> > > Just to reiterate, as discussed in the Introduction, our focus is on **few-shot learning** and the **novel contributions** (which, as far as we are aware, were not observed in previous works) are: 1) showing that even in the few-shot learning setting that is characteristic with a limited number of available samples, **curating a set of high-quality samples** and using only these for training leads to **better performance than using all available samples** for adaptation; 2) identifying that the **learnability of samples plays a critical role** in few-shot learning; and 3) showing that a **sophisticated combination** of the existing sample selection approaches enhances the effectiveness of transfer and even **outperforms the existing state-of-the-art approaches.**
> > >
> > > Finally, as you mention only the concern with novelty (which can be loosely mapped to W1), **we hope we addressed all the remaining raised concerns sufficiently (W2-W4) and answered all clarifying questions (Q1-Q4).** If so, we would like to kindly ask you to **consider reflecting it in the review and review score. Thank you very much.**

---

### Official Review · Reviewer_aHzt · 2024-11-06

**Soundness:** 3
**Presentation:** 2
**Contribution:** 2
**Rating:** 5
**Confidence:** 3

**Summary:**

This paper explores 20 sample selection strategies in few-shot learning, introducing the Automatic Combination of Sample Selection Strategies (ACSESS) method. ACSESS consistently outperforms individual strategies across five approaches using 8 image and 6 text datasets. The findings highlight the importance of modality and dataset characteristics, demonstrating that effective selection enhances performance, particularly in imbalanced and noisy datasets. Notably, the strategies are more beneficial with fewer shots, often reverting to random selection with larger numbers of shots.

**Strengths:**

1. Performance Improvement through Strategic Selection: The proposed ACSESS method effectively combines various sample selection strategies, leading to consistently better performance in few-shot learning tasks. This combination enhances the overall effectiveness of sample selection, particularly in imbalanced datasets with noisy samples.
2. Emphasis on Learnability: The study reveals that the learnability of samples is more critical than their informativeness or representativeness for the success of few-shot learning.
3. Cost Reduction and Efficiency: Selecting a subset of samples rather than using full-shot training reduces computational costs while improving performance, especially on imbalanced and noisy datasets. The findings indicate that even in scenarios with limited available samples per class, curated selections can still yield benefits in performance and efficiency.

**Weaknesses:**

1. Although this paper conducts thorough experiments, it appears to lack significant insights into sample selection strategies and seems to be merely a combination of existing approaches.
2. Many key points in the paper are not clearly articulated, such as how the proposed ACSESS utilizes weighted strategies and active learning, which require further clarification.
3. Table 1 categorizes the comparison strategies into three major groups. However, the classification of strategies presented in Table 2 appears inconsistent with that in Table 1, which may lead to confusion. Furthermore, additional updated comparison strategies are needed to demonstrate the effectiveness of the proposed method.

**Questions:**

Please refer to weakness.

---

> ### Author Response · Authors · 2024-11-20
>
> **Thank you for your detailed review, and for highlighting the paper's strengths**, especially the **critical role of learnability** and the **benefit the curated selection of samples provides** for few-shot learning in terms of performance and efficiency.
>
> Below, we address the raised concerns.
>
> > W1: Lack of insights into sample selection strategies
>
> Although the ACSESS strategy is “merely a combination of existing approaches”, it **allows us to draw novel insights and findings** regarding the benefit of sample selection in a few-shot setting. One such finding is, as highlighted as a part of strengths, **identifying that the learnability of samples plays a critical role** in the success of few-shot learning. Another significant finding (also highlighted in the strengths) is that the **combination enhances the overall effectiveness of sample selection and outperforms the existing state-of-the-art solutions**, especially in imbalanced datasets with noisy samples – thanks to identifying samples with complementary properties.
>
> Further findings, which were revealed by our study and were not thoroughly investigated in previous works (as discussed in Section 1), include:
> - Sample selection has a higher impact on a lower number of shots, while the boost of using more samples is negligible after a certain point
> - Sample selection is beneficial even on small dataset sizes
>
> As such, **we believe the paper provides sufficient novelty through these findings** (which are not observed in previous works), showing that a simple combination of existing, well-known sample selection strategies is enough to provide significant benefit (that even outperforms other, more complicated state-of-the-art methods) and allow us to draw novel insights into sample selection in few-shot learning.
>
>
> > W2: Better articulation of the key points in the paper
>
> To **provide an easier and clearer understanding** of the key points of the ACSESS strategy, we submitted **a revised version of the paper in which we provide an Algorithm of its steps** (Algorithm 1 in Section 3.2 and further low-level details of the forward, backward and datamodels selection in Appendix C).
>
> To directly answer the raised concerns for clarification:
> The sample scores of the “weighted strategies” are multiplied by the weight in order for ACSESS to prefer those samples more.
> Active learning strategies are part of the 20 sample selection strategies we utilise and are potentially combined together.
>
> > W3: Inconsistency in Tables 1 and 2
>
> Thank you for pointing out the inconsistencies. **In the revised manuscript, we have added one strategy to overview Table 1 that was left out by mistake** (DeepFool, the strategy is still referenced in all the result Tables and Figures and also in the Appendices, mainly Appendix D).
>
> In addition, we have **modified Table 1 to allow for better readability** by explicitly mentioning the group of selection strategies it belongs to (in addition to the property it selects based on) and separated the paper references into a separate column.
>
> > Additional updated comparison strategies are needed to demonstrate
>
> In the study, we **already focus on the 20 sample selection strategies.** These were selected as they represent the **best-performing strategies** and those **not specific for only a single approach** (e.g., in-context learning specific) based on our analysis of related work. As such, we believe that they are sufficient for demonstrating the effectiveness of our proposed ACSESS strategy – **if we are missing any specific ones, which you have in mind and would provide a more comprehensive comparison, please, could you point them out specifically?** It would help us to address your concern directly.
>
> **We hope that we have addressed all your concerns in our response. If there are any further questions, we will be glad to continue the discussion. Thank you very much.**

---

> > ### Comment · Reviewer_aHzt · 2024-11-27
> >
> > The authors mentioned two findings: "Sample selection has a higher impact on a lower number of shots, while the boost of using more samples is negligible after a certain point" and "Sample selection is beneficial even on small dataset sizes." However, these findings are essentially reflected in many existing studies, differing primarily in terms of presentation and articulation. They do not offer particularly novel insights.
> >
> > Thank you for the authors' contributions. While this work has accomplished a considerable amount of effort overall, it indeed lacks novelty and insightful contributions. Although the authors provided responses, these explanations failed to offer convincing answers.

---

> > > ### Author Response · Authors · 2024-11-27
> > >
> > > **Thank you very much for your answer, and willingness to discuss the paper further.** As specified in the rebuttal answer as well as in the Introduction section, the **two mentioned findings are only one part of the overall findings** (they are a result of ablation studies). Other main findings include:
> > > - Identifying that the **learnability of samples plays a critical role** in few-shot learning
> > > - The combination **enhances the overall effectiveness** of sample selection and outperforms the existing state-of-the-art solutions
> > >
> > > > these findings are essentially reflected in many existing studies, differing primarily in terms of presentation and articulation
> > >
> > > Please, **could you be more specific and point us to such existing studies that already observed in few-shot learning what was shown in our work**, namely that: 1) subsampling the few samples available in few-shot learning further leads to better performance; 2) sample selection has a higher impact with a lower number of samples; and 3) using more samples has a negligible impact after a certain point? Analysing and incorporating these many existing studies **would surely help us to improve** the paper and allow us to **address this concern more precisely** (e.g., clarifying the research gap we are addressing).
> > >
> > > As you mentioned only this concern, **we hope we addressed the remaining raised concerns sufficiently.** If so, we would like to kindly ask you to **consider reflecting it in the review and review score. Thank you very much.**

---

### Meta-Review · Area_Chair_kwcv · 2024-12-22

**Metareview:**

The paper empirically explores the question of the impact of sample selection on the performance of few-shot learning and presents a scheme: ACSESS, that can automatically combine strategies for selecting data samples towards improved performance. The key idea is to identify a subset of strategies that are scored well and which are combined based on their expected performance improvement. The strategies are built over the informativeness, representativeness, and learnability properties of the samples. Experiments are presented on a number of image and text datasets and various conclusions are drawn. A key result is that ACSESS leads to a consistent 1-2% improvement in performance across the datasets.

**Additional Comments On Reviewer Discussion:**

The paper received mixed borderline reviews. The reviewers generally appreciated the performance improvements demonstrated in the paper using the ACSESS method as well as some of the interesting findings such as the learnability of the samples being more critical, subset selection leading to a reduction in the compute cost, among others. However, all the reviewers appear less convinced on the magnitude of the contributions with regards to prior works or the general understanding in machine learning on sample selection and ensemble approaches. Some of the key concerns raised by the reviewers are summarized below.

1. Lack of novelty or technical insights (aHzt, aHzt, 2vH8)
2. Lack of clarity in many sections of the paper on how precisely various modules are implemented (aHzt, hLgQ, 2vH8, tzJ3)
3. Lack of any theoretical or practical insights into how efficiently can the method be used on a new dataset or task (hLgQ, 2vH8, UDbF)
4. Missing experiments on other tasks and neural architectures (2vH8, tzJ3)

These questions were thoroughly discussed with the authors, however there has not been any change in the stand of the reviewers even after the authors' responses. AC agrees with these concerns raised by the reviewers and believes while the proposed approach demonstrates empirical promise, the paper lacks new theoretical or practical insights that could make it useful for new datasets or tasks.  This is further exacerbated by the lack of a general study over varied neural architectures and tasks towards answering if the proposed ideas of scoring samples could be applicable. Thus, AC recommends reject.

---

### Decision · Program_Chairs · 2025-01-22

Reject